# CRISPR-RfxCas13d screening uncovers Bckdk as a post-translational regulator of maternal-to-zygotic transition in teleosts

Luis Hernández-Huertas[1,2], Ismael Moreno-Sánchez [ID][1,2], Jesús Crespo-Cuadrado[1], Ana Vargas-Baco [ID][1,2], Gabriel da Silva Pescador[3], Ying Zhang[3], Zhihui Wen[3], Laurence Florens [ID][3], José M Santos-Pereira [ID][4,5], Ariel A Bazzini [ID][3,6 ✉] & Miguel A Moreno-Mateos [ID][1,2 ✉]

## Abstract

The maternal-to-zygotic transition (MZT) is a reprograming process encompassing zygotic genome activation (ZGA) and the clearance of maternally-provided mRNAs. While some factors regulating MZT have been identified, there are thousands of maternal RNAs whose function has not been ascribed yet. Here, we have performed a proof-of-principle CRISPR-RfxCas13d maternal screen, in which we targeted mRNAs encoding kinases and phosphatases or proteins regulating them in zebrafish. This screen identified branched-chain ketoacid dehydrogenase kinase, Bckdk, as a novel post-translational regulator of MZT. *Bckdk* mRNA knockdown caused epiboly defects, ZGA deregulation, H3K27ac reduction and a partial impairment of miR-430 processing. Phospho-proteomic analysis revealed that Phf10/Baf45a, a chromatin remodeling factor, is less phosphorylated upon Bckdk depletion. Further, *phf10* mRNA knockdown also altered ZGA, and expression of a phospho-mimetic mutant of Phf10 rescued the developmental defects observed after *bckdk* mRNA depletion, as well as restored H3K27ac levels. Altogether, our results demonstrate the competence of CRISPR-RfxCas13d screenings to uncover new regulators of early vertebrate development and shed light on the post-translational control of MZT mediated by protein phosphorylation.

**Keywords** CRISPR-RfxCas13d; MZT; Kinases; Bckdk; Zebrafish
**Subject Categories** Development; Methods & Resources

## Introduction

The first stages of metazoan development are essentially controlled by instructions deposited in the oocyte in the form of proteins and mRNA. This maternal contribution directs the first molecular events in the zygote and triggers an embryonic reprogramming where cells transition from pluripotent to differentiated cellular states (Heyn et al, 2014; Vastenhouw et al, 2019), a process named the maternal-to-zygotic transition (MZT) (Vastenhouw et al, 2019; Lee et al, 2014). The MZT is associated with two major events. First, zygotic genome needs to be activated, and is associated with widespread chromatin remodeling within the genome (Wike et al, 2021). Second, maternal mRNAs are selectively and actively eliminated from the embryo in a programmed manner (Vastenhouw et al, 2019; Lee et al, 2014).

One of the most used animal models for the study of early embryonic development and MZT is zebrafish (*Danio rerio*) where the onset of zygotic genome activation (ZGA) occurs at 64-cell stage (2 h post fertilization, hpf), leading to a major wave of transcriptional activity over the subsequent hours as the embryo progresses towards epiboly (Chan et al, 2019; Baia Amaral et al, 2024). Some regulators of the MZT have been described and characterized in zebrafish at different levels such as transcriptional pioneer factors (i.e., Nanog, SoxB1, and Pou5f3) or chromatin remodelers controlling ZGA, and the microRNA, miR-430, and codon optimality with a major role in the clearance of the maternal RNA after ZGA (Murphy et al, 2018; Wike et al, 2021; Miao et al, 2022; Pálfy et al, 2020; Riesle et al, 2023; Vejnar et al, 2019; Bazzini et al, 2012; Giraldez et al, 2006; Bazzini et al, 2016; Vastenhouw et al, 2010). In addition, translational control of maternal mRNAs and complete nuclear pore complexes maturation are other mechanisms controlling ZGA (Leesch et al, 2023; Bazzini et al, 2012; Chan et al, 2019; Shen et al, 2022; Ugolini et al, 2024). Despite these advances, the maternally-instructed post-translational regulatory mechanisms such as protein phosphorylation or ubiquitination modulating MZT, are much less understood, especially in

[1]Andalusian Center for Developmental Biology (CABD), Pablo de Olavide University/CSIC/Junta de Andalucía, Ctra. Utrera Km.1, 41013 Seville, Spain. [2]Department of Molecular Biology and Biochemical Engineering, Pablo de Olavide University, Ctra. Utrera Km.1, 41013 Seville, Spain. [3]Stowers Institute for Medical Research, 1000 E 50th St, Kansas City, MO 64110, USA. [4]Departamento de Biología Celular, Facultad de Biología, Universidad de Sevilla, 41012 Sevilla, Spain. [5]Instituto de Biomedicina de Sevilla (IBiS), Hospital Universitario Virgen del Rocío/CSIC/Universidad de Sevilla, 41013 Sevilla, Spain. [6]Department of Molecular and Integrative Physiology, University of Kansas Medical Center, 3901 Rainbow Blvd, Kansas City, KS 66160, USA. ✉E-mail: arb@stowers.org; mamormat@upo.es

vertebrates (Vastenhouw et al, 2019; Liu et al, 2018) and a systematic analysis is needed.

Although maternal screenings have been performed in non-vertebrate systems (Luschnig et al, 2004), maternal gene functions have remained evasive in vertebrate models largely due to the lack of suitable approaches to systematically perturb oocyte-provided RNAs that can drive early development and MZT. This is particularly important in teleosts and other aquatic vertebrate models, as RNAi technology is not effective (Chen et al, 2017; Kelly and Hurlstone, 2011; Lund et al, 2011) and the use of morpholinos to knockdown mRNAs can trigger toxicity, off-targeting, and undesired effects such as the activation of innate immunity and cellular stress responses (Gentsch et al, 2018; Joris et al, 2017; Robu et al, 2007). To circumvent this, we recently optimized CRISPR-RfxCas13d to induce the specific, efficient, and cost-effective degradation of RNAs in different animal embryos such as zebrafish, medaka, killifish and mouse, recapitulating well-known maternal and/or zygotic embryonic phenotypes (Kushawah et al, 2020; Hernandez-Huertas et al, 2022; da Silva Pescador et al, 2024; Moreno-Sánchez et al, 2025; Kushawah et al, 2024). This opened an unprecedented opportunity to systematically study the maternal RNA contribution that can control MZT in an early vertebrate embryo model.

Here, as a proof-of-principle, we applied our optimized CRISPR-RfxCas13d system (Kushawah et al, 2020; Hernandez-Huertas et al, 2022) to deplete 49 maternal mRNAs encoding proteins with a potential role in the regulation of protein phosphorylation during MZT in zebrafish. We identified seven candidates whose knockdown (KD) triggers epiboly defects, a phenotype consistent with an alteration in MZT and ZGA (Lee et al, 2013; Chan et al, 2019; Kushawah et al, 2020; Riesle et al, 2023; Pálfy et al, 2020). By analyzing the transcriptomic profile at the onset of the ZGA major wave, we demonstrated that the specific mRNA depletion of two kinases, Bckdk and Mknk2a, triggered a downregulation of the pure zygotic genes (PZG); i.e., those with absent or low maternal contribution (Lee et al, 2013). Particularly, the KD of *bckdk* mRNA, coding a kinase with a mitochondrial and cytosolic location (Suryawan et al, 1998; Tian et al, 2020; White et al, 2018; Xue et al, 2017; Xu et al, 2023), led to the most severe developmental phenotype and transcriptional perturbation. Using phospho-proteomics and metabolic assays, we showed that the Bckdk non-mitochondrial role was responsible for the MZT regulation that we further investigated. By performing RNA-seq during early development, as well as SLAM-seq and ATAC-seq, we showed that ZGA and MZT were globally altered. Notably, while chromatin accessibility slightly changed upon *bckdk* mRNA KD, Histone 3 K27 acetylation (H3K27ac), an epigenetic mark crucial to trigger ZGA (Sato et al, 2019; Chan et al, 2019; Miao et al, 2022) that we measured through CUT&RUN and western blot assays, was significantly reduced. We also observed an impairment of the maternal RNA decay mediated by miR-430, whose biogenesis was affected. Through a phospho-proteomic analysis, we demonstrated that *bckdk* mRNA depletion reduced the phosphorylation of different proteins during MZT, and we focused on Phf10/Baf45a, a protein related to chromatin remodeling that is part of the Polybromo-associated BAF (pBAF) complex belonging to the SWI/SNF family (Yuan et al, 2022; Brechalov et al, 2014; Kadoch and Crabtree, 2015). CRISPR-RfxCas13d KD of *phf10* maternal mRNA also triggered a ZGA deficiency together with epiboly defects.

Importantly, while the phospho-mimetic mutant version of Phf10 rescued the developmental defects observed in the absence of maternal *bckdk* mRNA along with wild-type H3K27ac levels, neither the wild-type nor the non-phosphorylatable mutant versions had this effect. Finally, *bckdk* mRNA depletion also induced an early development perturbation and downregulation of PZG in medaka (*Oryzias latipes*), indicating a conservation of Bckdk's role in MZT among teleosts. Altogether, our results (i) demonstrate the competence of CRISPR-RfxCas13d as an RNA KD tool to perform maternal screenings in vertebrates, (ii) uncover Bckdk as a novel post-translational modulator of MZT through the regulation of both ZGA and maternal RNA clearance, and (iii) demonstrate that the early developmental alterations seen after *bckdk* mRNA KD is, at least in part, due to changes in the phosphorylation state and activity of Phf10.

## Results

### A CRISPR-RfxCas13d screening for maternally provided mRNA encoding protein phosphorylation regulators

To systematically determine whether the phosphorylation state of proteins plays a role during MZT, we selected 49 genes annotated as kinases, phosphatases, and factors directly related to their activity that had not been previously characterized in zebrafish. We particularly chose these candidates based on their expression regulation during early zebrafish development. Thus, these candidates showed a specific transcriptomic and translational pattern described for known regulatory factors of MZT such as Nanog or Pou5f3 (Oct4): highly abundant mRNA in the oocyte as well as highly translated at the onset of ZGA with subsequent decreases in translation afterwards (Bazzini et al, 2014; Medina-Muñoz et al, 2021; Chan et al, 2019; Lee et al, 2013) (Fig. 1A). Next, we designed three chemically synthesized and optimized guide RNAs (gRNAs) per mRNA to maximize targeting efficacy (Wessels et al, 2020; Hernandez-Huertas et al, 2022) (see Methods). These gRNAs, together with non-targeting negative control gRNAs (designed to target *gfp* and *rfp* mRNAs absent in the zebrafish transcriptome) and three gRNAs targeting *nanog* mRNA as a positive control (Kushawah et al, 2020), were co-injected with a purified RfxCas13d protein with a cytosolic location (Kushawah et al, 2020) into one-cell stage zebrafish embryos (Fig. 1B; Dataset EV1). Then, the development of these embryos was monitored for 24 h, as well as control embryos (uninjected or injected with RfxCas13d only). While most KD candidates did not affect early development compared to the control embryos, seven KD induced epiboly defects between 4 and 6 hpf in, at least, one third of the injected embryos, a phenotype potentially compatible with an alteration in MZT and ZGA as observed for Nanog KD, a well-known transcriptional factor regulating ZGA (Kushawah et al, 2020; Miao et al, 2022; Chan et al, 2019) (Figs. 1C and EV1A). These seven candidates included (i) proteins related to calcium signaling that have been associated with the modulation of kinase activity, such as the Calmodulin paralogs Calm1a and Calm2a (Sahu et al, 2017) and Cab39l (Li et al, 2018), (ii) MAPK interacting serine/threonine kinases, such as Mknk1 and Mknk2a (Pinto-Díez et al, 2020; Hu et al, 2012), and (iii) a regulatory subunit part of the serine/threonine phospho-protein phosphatase 4 complex, Ppp4r2a

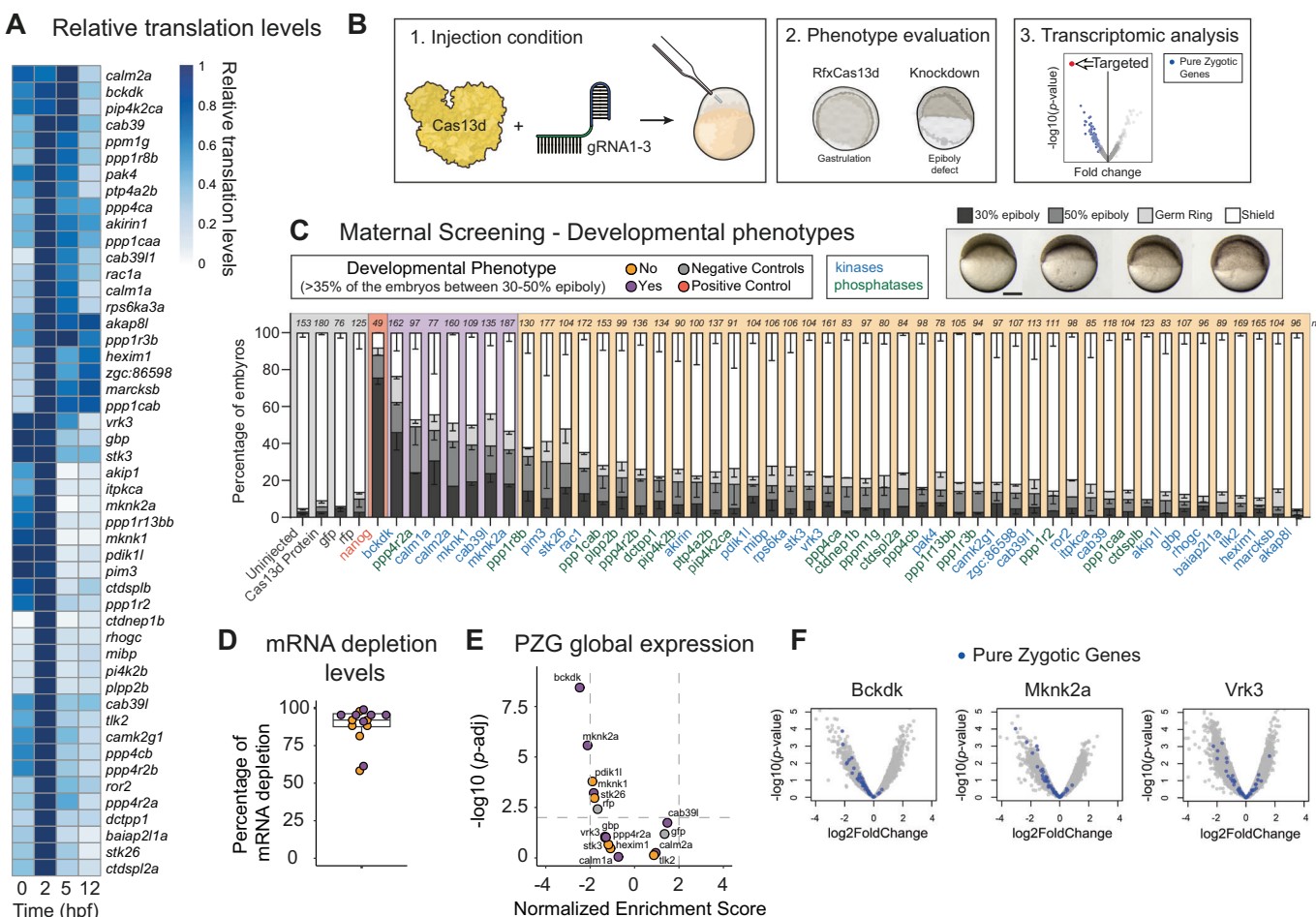

**Figure 1.	Maternal screening using CRISPR-RfxCas13d system reveals Bckdk as a regulator of ZGA in zebrafish.**

(A) Heatmap plot of relative translation data during early developmental stages (0, 2, 5, and 12 hpf) of the kinases and phosphatases selected for the maternal screening. Data from Bazzini et al, 2014 and Chan et al, 2019. (B) Schematic illustration of the experimental setup used for the maternal screening. (1) RfxCas13d protein (Cas13d, 3 ng per embryo) was mixed with a pool of 3 gRNAs (1000 pg per embryo) and injected into one-cell-stage zebrafish embryos. (2) Knockdown conditions were evaluated and considered positive when epiboly defects were observed at 6 hpf. Control embryos injected only with RfxCas13d protein were at a normal gastrulation stage. (3) Transcriptomic analysis was performed at 4 hpf to evaluate mRNA depletion and the global analysis of pure zygotic genes (PZG). Knockdown conditions with an epiboly defect and global downregulation of PZG were considered ZGA candidates. (C) Stacked barplots showing the percentage of phenotypes observed at 6 hpf under maternal screening conditions. Control conditions, either uninjected embryos, injected with RfxCas13d protein alone or with RfxCas13d plus non-targeting gRNAs (designed for *gfp* or *rfp* mRNAs), are indicated with a gray background. The positive control, targeting *nanog* mRNA, is indicated with a red background. Positive candidates (purple background) were considered when more than 35% of the embryos showed a developmental delay (30–50% epiboly at 6 hpf where control embryos were at shield stage). Negative candidates are indicated with an orange background. mRNAs encoding for kinases are labeled in blue and mRNAs encoding for phosphatases are labeled in green. The results are shown as the averages ± standard error of the mean of each developmental stage from at least two independent experiments. Number of embryos evaluated (*n*) is shown for each condition. Representative pictures of different zebrafish epiboly-stages from *bckdk* mRNA knockdown are shown in the upper panel. 30% epiboly, 50% epiboly, germ ring, and shield stages correspond to 4.6, 5.3, 5.7, and 6 hpf in uninjected embryos growing in standard conditions, respectively (scale bar, 0.05 mm). (D) Boxplot with individual values showing the percentage of mRNA depletion using the RfxCas13d approach for seven positive candidates (purple dots) and seven negative candidates (orange dots) from maternal screening measured by Bulk-RNA-Seq at 4 hpf. The mean of mRNA depletion is represented together with the first and third quartile. Vertical lines indicate the variability outside the upper and lower quartiles. (E) Scatter plot of Normalized Enrichment Score and adjusted *p* value associated with gene set enrichment analysis of pure zygotic genes (PZG) defined by Lee et al, 2013 from positive (purple dots), negative candidates (orange dots) and negative controls (gray dots) (see Methods for details). Bulk-RNA-Seq data from panel (D). Vertical and horizontal dashed lines indicate −2 and 2 normalized enrichment score (NES) and adjusted *p* value = 0.01, respectively. NES and adjusted *p* value were calculated using gene set enrichment analysis (GSEA) (Subramanian et al, 2005; Mootha et al, 2003) (see Methods for details). (F) Scatter plots representing the log2 fold change in mRNA levels and *p* value from two biological RNA-seq replicates (*n* = 20 embryos/biological replicate) at 4 hpf in zebrafish embryos injected with Cas13 protein and three gRNAs targeting *bckdk* (left), *mknk2a* (middle) or *vrk3* (right) mRNA. *p* values were calculated using the Wald test. Pure zygotic genes mRNAs (PZG) defined by Lee et al, 2013 are indicated in blue. Source data are available online for this figure.

(Liao et al, 2020). Interestingly, the most severe delay was observed upon the KD of the branched-chain ketoacid dehydrogenase kinase (*bckdk*) mRNA. Bckdk is a well-conserved kinase (80.8 and 78.7% amino acid similarity between human and zebrafish and between mouse and zebrafish, respectively) located in the mitochondria with a known role in controlling branched-chain essential amino acid (valine, leucine, and isoleucine, BCAA) catabolism (Suryawan et al, 1998; Murashige et al, 2022). In addition, Bckdk has been recently detected in the cytosol where, for example, it is able to (i) upregulate the MEK-ERK signaling pathway promoting cell proliferation, invasion, and metastasis and/or (ii) stimulate lipogenesis by phosphorylating ATP-citrate lyase (Tian et al, 2020; Xue et al, 2017; White et al, 2018).

At 24 hpf, the seven KDs that showed an early developmental phenotype exhibited a lower viability correlating with the percentage of embryos with epiboly defects (Fig. EV1B) while the rest of the KDs did not reveal any remarkable alteration (Fig. EV1C) except for Mibp, a ribosylnicotinamide kinase orthologous to human NMRK2 (Li et al, 1999). *Mibp* mRNA KD embryos showed a defect in brain development, causing microcephaly in 60% of embryos (Fig. EV1D–F), consistent with its neural expression in zebrafish (Tapial et al, 2017). However, since we were interested in uncovering novel regulators of MZT, we focused on the seven candidates whose KD caused an early developmental defect during the first hours after fertilization.

Epiboly failure during early zebrafish development does not always correlate with an alteration in ZGA and MZT (Cheng et al, 2004). To test whether these phenotypes were associated with a perturbation of ZGA, we performed a transcriptomic analysis at 4 hpf of the KD of 14 candidates, including the seven that showed an epiboly alteration and seven with a normal development as well as embryos injected with RfxCas13d only or together with non-targeting gRNAs (*gfp* or *rfp* mRNAs) as negative controls (Fig. 1B). Since zebrafish embryos treated with transcription inhibitors like triptolide or alpha-amanitin can develop normally up to the sphere stage (4 hpf) (Chan et al, 2019), we specifically chose this timepoint to avoid potential confounding developmental and transcriptional effects that could be potentially observed in embryos with delayed gastrulation at 6 hpf. First, RNA-seq data showed high target efficiency for the 14 targets from 58% to 99% mRNA reduction with a median of 92% (Fig. 1D). Additionally, we analyzed potential off-targets effects (up to three mismatches) (Wessels et al, 2024; Wei et al, 2023; Shembrey et al, 2024) and found no substantial off-target activity (Fig. EV2). Indeed, each targeted mRNA showed the highest level of depletion with the lowest $p$ value in its corresponding transcriptome compared to its potential off-targets (Fig. EV2). However, when targeting Calmodulins *calm1a* and *calm2a* mRNAs, we observed depletion of two closely related paralogs mRNAs, *calm3a* and *calm2b*, which shared 100% protein identity. For these KDs, three out of six employed gRNAs could potentially trigger this off-target based on the defined criteria (Wessels et al, 2024; Wei et al, 2023; Shembrey et al, 2024) (Fig. EV2). However, although significant, the depletion of these paralogs occurred with lower efficiency than the on-target KD. Next, to address whether ZGA was affected, we compared the mRNA level of genes previously defined as pure zygotic genes (Lee et al, 2013) (PZG), in the KD embryos to the embryos injected with Cas13d alone (Fig. 1B). Interestingly, *bckdk* and *mknk2a* KD negatively affected the expression of the PZG (Fig. 1E,F,

normalized enrichment score <−2 and EV3) suggesting a role in the ZGA while other KDs such as *vrk3* or the non-targeting controls did not (Figs. 1E,F, normalized enrichment score between −2 and 2 and EV3). However, other KD candidates that showed an epiboly defect and did not alter ZGA could affect other molecular pathways (Figs. 1C,E and EV3). Indeed, analyzing downregulated genes from the seven candidate KDs revealed that the depletion of mRNA coding proteins related to calcium signaling and kinase activity shared the highest number of depleted genes among all possible comparisons, suggesting a potential common role for these proteins (Fig. EV1G and Methods for details).

It has been recently shown that RNA-targeting mediated by CRISPR-RfxCas13d may trigger collateral activity in mammalian cells (Shi et al, 2023; Li et al, 2023; Tong et al, 2022). This is an uncontrolled and gRNA-independent RNA-targeting after a previous and specific gRNA-dependent activity of RfxCas13d (Shi et al, 2023; Tong et al, 2022; Li et al, 2023). One of the hallmarks of the collateral activity triggered by RfxCas13d is ribosomal fragmentation and a decrease in the RNA integrity number (Shi et al, 2023; Li et al, 2023). Based on these molecular markers, we did not observe evidence of collateral effects mediated by CRISPR-RfxCas13d in zebrafish embryos when targeting these 14 endogenous mRNAs (Fig. EV4) as we previously showed for other transcripts (Kushawah et al, 2020; Moreno-Sánchez et al, 2025; da Silva Pescador et al, 2024; Kushawah et al, 2024). Altogether, these results demonstrate that (i) CRISPR-RfxCas13d is a powerful KD approach to screen for dozens of maternal genes affecting early vertebrate development and (ii) specifically, the KDs of two maternal mRNA encoded kinases (*bckdk* and *mknk2a*) exhibit epiboly defects together with a significant ZGA perturbation.

### *Bckdk* mRNA knockdown is rescued with wild-type Bckdk but not with a loss-of-kinase-activity mutant version, does not affect mitochondrial function and has a conserved developmental effect in medaka

Considering that *bckdk* mRNA KD resulted in the most drastic depletion of PZG expression and severe phenotype from the maternal screening, we focused on this kinase to further understand its role in regulating MZT. First, to validate the *bckdk* mRNA KD phenotype and further rule out off-target effects, four different gRNAs were individually injected with Cas13d protein. These four gRNAs efficiently decrease *bckdk* mRNA levels (Dataset EV1; Fig. EV5A) and recapitulate the early developmental phenotype previously observed (Fig. EV5B). Then, to perform a phenotype rescue experiment, we used a gRNA targeting the 3′ UTR of *bckdk* mRNA, which recapitulated the 6 hpf epiboly defects (Dataset EV1; Fig. EV5C,E). Despite the intrinsic mosaic nature caused by the embryo microinjection, the developmental phenotype was rescued by the coinjection of an mRNA encoding for Bckdk tagged with an HA containing a different, and gRNA-resistant, 3′ UTR (Fig. 2A,B). As expected, the phenotype was not rescued by an ectopic *gfp* mRNA (Fig. 2A,B). In addition, the overexpression of *gfp* or *bckdk* mRNAs did not cause any developmental delay in WT embryos and the cognate *bckdk* mRNA was detected at mRNA and protein levels (Fig. EV5D–F). Furthermore, GFP fluorescence was similar in *bckdk* mRNA KD compared to the control, reinforcing the absence of collateral activity evidence in our CRISPR-RfxCas13d targeting conditions (Fig. EV5G). These results consolidate and validate the

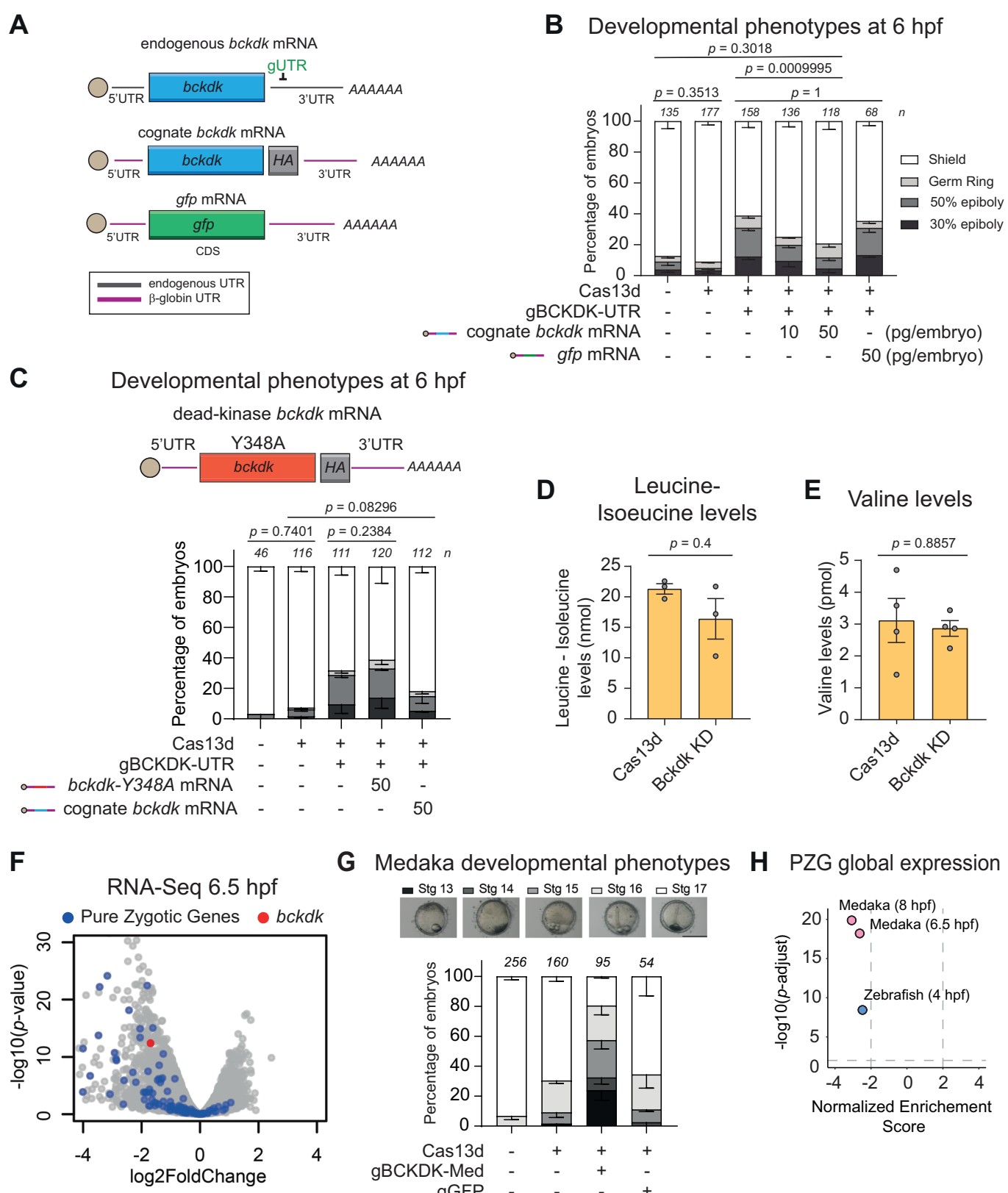

**A** endogenous *bckdk* mRNA

cognate *bckdk* mRNA

*gfp* mRNA

**B** Developmental phenotypes at 6 hpf

**C** Developmental phenotypes at 6 hpf

dead-kinase *bckdk* mRNA

**D** Leucine-Isoeucine levels

**E** Valine levels

**F** RNA-Seq 6.5 hpf

**G** Medaka developmental phenotypes

**H** PZG global expression

**Figure 2.** **_Bckdk_ mRNA knockdown is rescued with wild-type Bckdk but not with a loss-of-kinase activity mutant version, does not affect mitochondrial function and triggers a ZGA alteration in medaka.**

(A) Schematics of endogenous _bckdk_, cognate _bckdk_ with β-Globin 5′ and 3′ UTR from _Xenopus laevis_ and _gfp_ mRNAs co-injected at 10 or 50 pg/embryo with RfxCas13d protein and one gRNA targeting the 3′UTR of the endogenous _bckdk_ mRNA (gUTR). Endogenous and β-globin UTR are indicated as black or pink lines, respectively. (B) Stacked barplots showing the percentage of observed phenotypes at 6 hpf in zebrafish embryos injected with RfxCas13d protein (3 ng/embryo) together with one gRNA targeting the 3′UTR of the endogenous _bckdk_ mRNA (gBCKDK-UTR) (1000 pg/embryo) and co-injected with an exogenous _bckdk_ mRNA (10–50 pg/embryo) or with a _gfp_ mRNA (50 pg/embryo). The results are shown as the averages ± standard error of the mean of each developmental stage from at least two independent experiments. The number of embryos evaluated (_n_) is shown for each condition. The phenotype selection criteria were the same as in Fig. 1C. Exact _p_ values are indicated above, $\chi^2$-test. (C) Schematic representation of _bckdk_ mRNA with a specific point mutation in Tyrosinase at position 348, where an alanine substitution (Y348A) leads to a 95% reduction in kinase activity (Wynn et al, 2000; Singh et al, 2024) (dead-kinase _bckdk_ mRNA) (top). Stacked barplots showing the percentage of observed phenotypes in zebrafish embryos injected with RfxCas13d protein (3 ng/embryo) together with one gRNA targeting the 3′UTR of the endogenous _bckdk_ mRNA (gBCKDK-UTR) (1000 pg/embryo), and co-injected with either exogenous _bckdk_ mRNA (50 pg/embryo) or the dead-kinase _bckdk_ mRNA version (50 pg/embryo). The results are shown as the averages ± standard error of the mean of each developmental stage from three independent experiments. Exact _p_ values are indicated above, $\chi^2$-test. Leucine/isoleucine (D) and valine (E) normalized levels in zebrafish embryos injected with RfxCas13d alone (Cas13d) or together with a mix of two gRNAs targeting _bckdk_ mRNA (Bckdk KD). The results are shown as the averages ± standard error of the mean of at least three biological replicates (_n_ = 50 embryos/biological replicate). Exact _p_ values are indicated above, Mann–Whitney _U_-test. (F) Scatter plot representing the log2 fold change in mRNA level and the associated _p_ value from three biological RNA-seq replicates (_n_ = 10 embryos/biological replicate) at Stage 11 (Late Blastula Stage; 6.5 hpf) in medaka embryos in the conditions described in panel (G). _bckdk_ mRNA is represented in red. Pure zygotic genes mRNAs from medaka embryos, determined from Li et al, 2020 data were depicted in blue. _p_ values were calculated using the Wald test. (G) Stacked barplots showing the percentage of observed phenotypes in medaka embryos injected with RfxCas13d protein (6 ng/embryo) alone or together with one gRNA (gBCKDK-Med) targeting _bckdk_ mRNA (2000 pg/embryo) or with a mix of three non-targeting gRNAs (gGFP, 1000 pg/embryo) as a negative control. The results are shown as the averages ± standard error of the mean of each developmental stage from three independent experiments. The number of embryos evaluated (_n_) is shown for each condition. Representative pictures of medaka epiboly-stages from _bckdk_ mRNA knockdown are shown in the top panel. Epiboly-stages earlier than Stg 16 at 25 hpf indicate epiboly defects. Stages 13, 14, 15, 16, and 17 correspond to 13, 15, 17.5, 21, and 25 hpf in uninjected embryos growing in standard conditions, respectively (scale bar, 0.1 mm). (H) Scatter plot of normalized enrichment score (NES) and adjusted _p_ value associated with gene set enrichment analysis of pure zygotic genes (PZG) defined by Lee et al, 2013 (zebrafish) or by Li et al, 2020 (medaka) upon the depletion of _bckdk_ mRNA in zebrafish (blue dot, 4 hpf) or medaka embryos (pink dots, 6.5 or 8 hpf), respectively. NES and adjusted _p_ value were calculated using gene set enrichment analysis (GSEA) (Subramanian et al, 2005; Mootha et al, 2003) (see Methods for details). Source data are available online for this figure.

specific targeting induced by CRISPR-RfxCas13d and the role of Bckdk in regulating early development in zebrafish.

Second, to determine the onset of the observed developmental defects, cell proliferation (mitotic cells) and total number of cells were quantified in embryos depleted on Bckdk before the beginning of the epiboly at 2 and 4 hpf (Fig. EV5H–J). No significant differences were observed between embryos with KD of _bckdk_ mRNA and the control condition, suggesting that Bckdk depletion did not affect the proliferation rate nor cell division and demonstrating that the observed developmental phenotype takes place between 4 and 6 hpf (Fig. EV5H–J).

Third, to determine whether Bckdk regulates early development through its kinase activity, we generated a kinase-dead version. A specific point mutation in the Tyrosine 384 (Y348A) has been shown to reduce the kinase activity of Bckdk by 95% (Wynn et al, 2000; Singh et al, 2024). We introduced this same mutation into the zebrafish _bckdk_ ORF and used the resulting isoform in a rescue experiment, similar to the one previously performed with the wild-type version (Fig. EV5K). Remarkably, this kinase-dead version failed to rescue the developmental phenotype upon Bckdk KD (Fig. 2C). Consistent with previous reports and given that Bckdk autophosphorylation enhances its own stability (Wynn et al, 2000; Singh et al, 2024), this kinase-dead version was less stable due to the loss of kinase activity (Fig. EV5L). Importantly, increasing the concentration of this kinase-dead mRNA still did not rescue the developmental phenotype (Fig. EV5M), indicating that the failure to rescue was not simply due to reduced protein stability.

Fourth, the loss-of-function of Bckdk in mammalian cells causes a loss of phosphorylation and hyperactivation of the branched-chain ketoacid dehydrogenase Bckdh in the mitochondria (Suryawan et al, 1998), resulting in a decreased BCAA availability (García-Cazorla et al, 2014; Joshi et al, 2006; Novarino et al, 2012).

To elucidate whether the KD of _bckdk_ mRNA affects its mitochondrial activity during early development, we measured BCAA levels. We observed that the amount of isoleucine-leucine and valine did not significantly change in comparison to embryos injected with Cas13d alone (Fig. 2D,E). Together, these data suggest that the _bckdk_ mRNA KD only affects its non-mitochondrial function, ultimately triggering a ZGA alteration and epiboly defects.

Fifth, _bkcdk_ mRNA is also maternally provided in medaka (Li et al, 2020), another teleost model used to study MZT and that evolutionary diverged from zebrafish 115–200 millions years ago (Signore et al, 2008). To test whether the role of Bckdk is conserved among teleosts, the maternal _bckdk_ mRNA in medaka was depleted using CRISPR-RfxCas13d (Dataset EV1). As observed in zebrafish, CRISPR-RfxCas13d induced an efficient KD of _bckdk_ mRNA (3.2-fold change _p_ value <3.3e-13) at 6.5 hpf in medaka (Fig. 2F). Additionally, the downregulation of _bckdk_ mRNA caused (i) a delay during early development that was not observed when using a non-targeting gRNA (gGFP) (Fig. 2G), (ii) a decrease in viability later during the embryogenesis (Fig. EV5N), and (iii), based on RNA-seq, triggered a global downregulation of PZG observed at different time points along the ZGA in medaka (Li et al, 2020) (Figs. 2F,H and EV5O). Therefore, the role of Bckdk during embryogenesis is conserved between zebrafish and medaka.

## Bckdk modulates the transcription of zygotic and maternal-zygotic expressed genes

Upon determining that _bckdk_ mRNA KD impacts the PZG expression, we explored whether it also altered the transcription of the maternal-and-zygotic expressed genes, which represent more than half of the transcriptome (Baia Amaral et al, 2024; Bhat et al, 2023; Fishman et al, 2024). We employed SLAM-seq, a technique

that differentiates maternal and zygotic mRNA through nascent mRNA labeling with 4-thiouridine (S4U) (Fig. 3A). To accurately assess these effects, we first optimized SLAM-seq alongside CRISPR-RfxCas13d use in zebrafish. Thus, we titrated different amounts of S4U that, in combination with the RfxCas13 protein, did not affect early development. Previously, it has been shown that up to 75 pmol of S4U per embryo resulted in high nascent label signal without developmental alteration (Bhat et al, 2023; Baia Amaral et al, 2024). Although injection of *cas13d* mRNA with 75 pmol of S4U did not affect early embryogenesis, 75 or 50 pmol of S4U injected in combination with purified RfxCas13d protein caused epiboly defects (Fig. EV6A). For that reason, we injected 25 pmol of S4U per embryo to address whether, beyond PZG, Bckdk depletion affected the transcription of the maternal-and-zygotic genes (Figs. 3A and EV6B). As expected and similar to the previous RNA-seq experiment (Figs. 1E,F and EV3; *bckdk* panel), *bckdk* mRNA and PZG were downregulated in the KD embryos compared to the control embryos (Fig. EV6B,C). Beyond the previously described PZG (Lee et al, 2013), we observed a downregulation of an additional set of PZG (Figs. 3B,C and EV6D) recently defined through SLAM-Seq approaches (Baia Amaral et al, 2024). Interestingly, besides the PZG, 1734 genes with maternal mRNA contribution, displayed a reduction in labeled transcript levels in the KD compared to the control, indicating that Bckdk also affects the transcription of maternal-zygotic genes. In total, 1884 genes were less transcribed upon *bckdk* mRNA KD (2-fold change) denoting a global ZGA downregulation (Fig. 3B). Nevertheless, there were 2907 genes which zygotic expression (label reads) was not affected in the *bckdk* mRNA depletion, suggesting that their transcription was independent of Bckdk (Fig. 3B), and 1491 mRNAs were upregulated (Fig. 3B). The vast majority of these upregulated genes (96.4%) were normally transcribed between 4 and 7 hpf in WT embryos (Baia Amaral et al, 2024) (Fig. EV6E). Notably, 36.4% of the genes upregulated in *bckdk* mRNA KD at 4 hpf did not show any transcriptional signal, based on SLAM-seq analysis, in WT embryos at that timepoint (Baia Amaral et al, 2024), suggesting that Bckdk depletion could potentially trigger an acceleration of genome activation for a subset of genes (Fig. EV6F). Interestingly, 30% of upregulated transcripts were known targets of miR-430, an important MZT factor promoting the degradation of hundreds of maternal mRNAs in zebrafish (Baia Amaral et al, 2024). This suggests that, in addition to increased transcription, those elevated levels could also result from enhanced mRNA stability due to reduced or delayed degradation of these zygotically expressed genes during the MZT (Baia Amaral et al, 2024). Altogether, these data suggest that *bckdk* mRNA KD affects the expression dynamics of genes specifically transcribed during ZGA and not genes activated later in zebrafish development. Remarkably, downregulated genes showed enriched biological processes related to transcriptional regulation and RNA processing; meanwhile, upregulated genes revealed processes associated with translation and protein biogenesis (Fig. EV6G). In conclusion, our results show that SLAM-seq in combination with CRISPR-Cas13d KD can be used to dissect the impact of a maternal factor in transcription during the MZT and specifically suggest that *bckdk* mRNA depletion deregulates the expression dynamics of more than half (54%) of the genes expressed during ZGA in zebrafish.

## *bckdk* mRNA KD leads to a slight rewiring of chromatin accessibility but a strong reduction of H3K27ac during MZT

To address whether the transcriptional changes were associated with altered chromatin accessibility, we performed an assay for transposase-accessible Chromatin analyses coupled to high-throughput sequencing (ATAC-seq) in *bckdk* mRNA KD and WT (Cas13d injected embryos) at 4 hpf. Interestingly, we observed that *bckdk* mRNA depletion only induced a slight change in chromatin accessibility with hundreds of regions with differential openness (n = 675; ~3% of the total peaks), compared to embryos only injected with Cas13d (Figs. 3D and EV6H). Most of these regions (n = 498, 74% of differentially accessible regions) were less accessible upon *bckdk* KD and, to some degree, enriched in some particular transcription factor motifs, including Gata1/2/3, Foxo6, Six5, and CTCFL (Fig. EV6I). On the other hand, more accessible regions (n = 177, 26% of differentially accessible regions) showed a minor enrichment in different transcription factor motifs, such as Nf1, Tfcp2l1, Hltf, Nfkb1, and Tead4 (Fig. EV6J). Overall, the motif analyses showed relatively low enrichments compared with the loss-of-function of specific transcription factors (Franke et al, 2021; Adam et al, 2020; Miao et al, 2022; Riesle et al, 2023), suggesting a modest and less specific effect of *bckdk* mRNA KD on chromatin accessibility that likely affects more heterogeneous transcription factor binding sites. Differentially expressed genes (DEG) according to SLAM-seq labeled data (Fig. 3B) associated with regions altered in their openness did not reveal a strong correlation towards transcriptional misregulation in the same direction as genome accessibility changes. Nevertheless, we found that downregulated genes showed a minor but still significant difference towards less open chromatin regions (less accessible regions >3-fold change, *p* value = 0.0415, Mann–Whitney *U*-test) (Fig. EV6K).

Since this slight impact on chromatin accessibility, we hypothesized that another layer of transcriptional regulation could be affected upon Bckdk depletion. Indeed, it has been shown that chromatin accessibility is an initial step, induced by pioneer factors, to activate transcription after fertilization but is not the ultimate trigger of ZGA (Miao et al, 2022). Instead, the inhibition of histone acetylation caused a massive downregulation of ZGA (Chan et al, 2019) without altering the chromatin accessibility landscape (Miao et al, 2022; Marsh et al, 2024). Indeed, we observed a prominent and significant decrease in H3K27ac (*p* value = 0.0023, Welch's *t*-test), an epigenetic mark essential to promote ZGA, in *bckdk* mRNA KD embryos in comparison to control embryos without changes in total histone 3 amount measured by western blot (Fig. EV6L,M). These results were further confirmed by a CUT&RUN assay where 52% of H3K27ac peaks were significantly reduced (>1.5-fold change *p* value <0.05, Fig. 3E,F).

Altogether, these data indicate that *bckdk* mRNA KD leads to a slight alteration of chromatin accessibility and a strong reduction in H3K27ac during early development that might be due to the miss-regulation of global chromatin remodelers rather than specific transcriptional activators or pioneer factors.

## *Bckdk* mRNA depletion affects miR-430 processing and activity during MZT

The maternal RNA degradation during the MZT is primarily driven by both maternally and zygotically encoded mechanisms (Vejnar

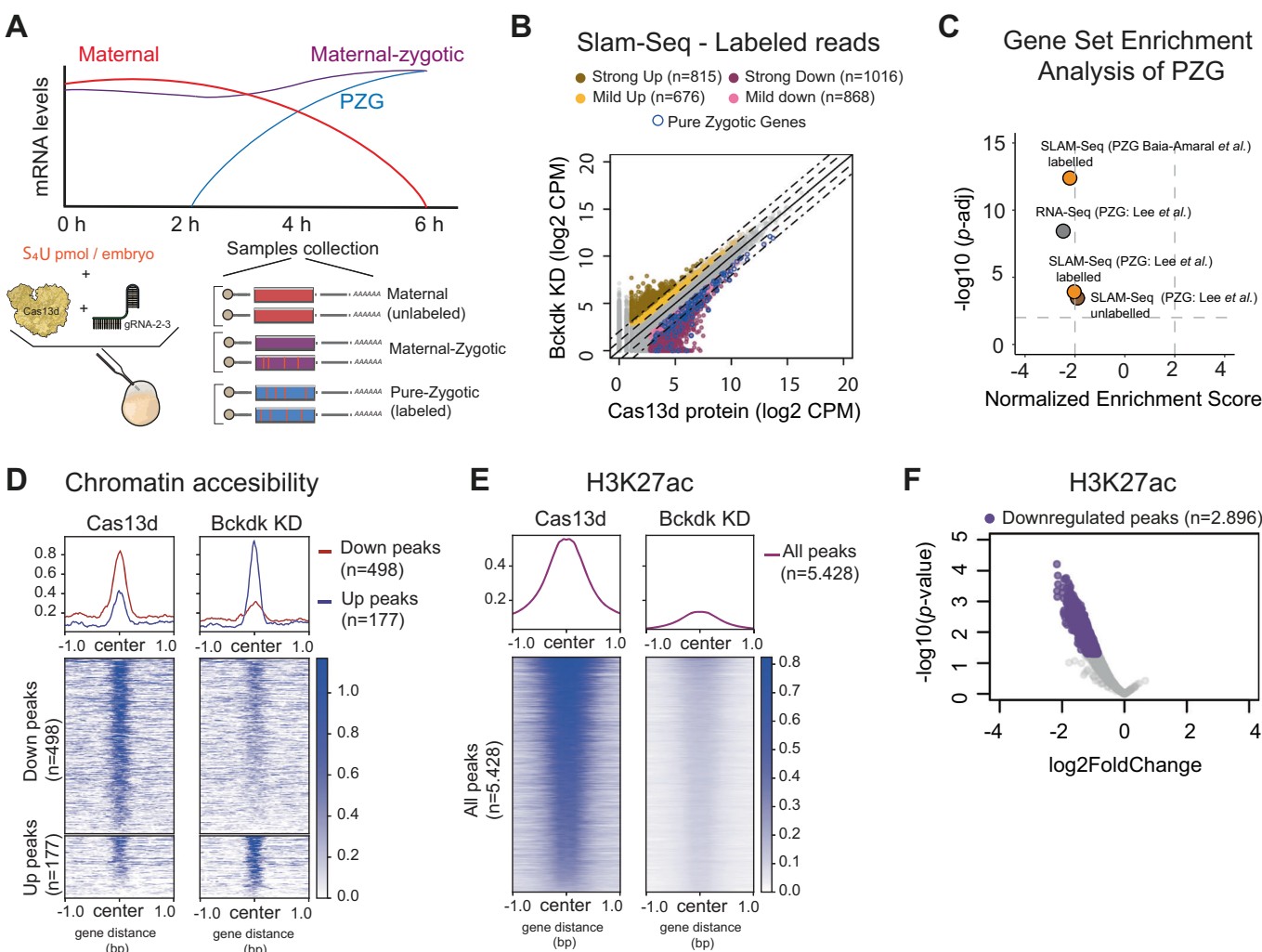

**Figure 3.** The knockdown of *bckdk* mRNA globally deregulates gene transcription inducing a slight chromatin accessibility alteration and strong H3K27ac reduction.

(**A**) Schematic representation of SLAM-Seq workflow. Zebrafish embryos were injected with 4-thiouridine ($S_4U$, 25 pmol/embryo) in one-cell-stage together with RfxCas13d protein alone or with a mix of two gRNAs targeting *bckdk* mRNA. Total RNA was extracted at 4 hpf from 20 embryos. Total RNA was chemically modified before library preparation. After library preparation T > C ratio conversion was used to differentiate between unlabeled (preexisting RNA; Maternal) and labeled (newly synthesized RNA; Zygotic) reads (see Methods for further details). (**B**) Scatter plot showing newly transcribed RNAs (SLAM-seq data in Counts per million reads; CPM) of zebrafish embryos at 4 hpf from two and four biological replicates ($n = 25$ embryos/biological replicate) from embryos injected with RfxCas13d protein alone (Cas13d protein) or with a mix of two gRNAs targeting *bckdk* mRNA (Bckdk KD), respectively. Dashed and dot-dash lines indicate a 2 and 5-fold difference between RNA levels, respectively. Up- and down-regulated genes are represented by yellow (between 2- and 5-fold change) and brown (>5-fold change) or pink (between 2- and 5-fold change) and purple (>5-fold change) dots, respectively. Pure zygotic genes (PZG) defined by Lee et al, 2013 are indicated by blue outline dots. The number of mRNAs in each category is shown (*n*). (**C**) Scatter plot of normalized enrichment score and adjusted *p* value associated of pure zygotic genes (PZG) defined by Lee et al, 2013 (PZG: Lee et al,) or Baia Amaral et al, 2024 (PZG: Baia Amaral et al,) from Bckdk KD data generated by RNA-Seq (gray dot) or by SLAM-Seq with labeled (orange dots) or unlabeled (brown dot) reads. NES and adjusted *p* value were calculated using gene set enrichment analysis (GSEA) (Subramanian et al, 2005; Mootha et al, 2003) (see Methods for details). (**D**) Plot profile (top) and heatmaps (bottom) representing normalized ATAC intensity signal for less accessible regions (red line; Down peaks) and more accessible regions (blue line; Up peaks) from the comparison between two biological replicates ($n = 80$ embryos/biological replicate) from zebrafish embryos injected with RfxCas13d protein (Cas13d) alone or together with a mix of two gRNAs targeting *bckdk* mRNA (Bckdk KD). Number of less accessible (Down peaks) or more accessible (Up peaks) regions are shown (*n*). (**E**) Plot profile (top) and heatmaps (bottom) representing normalized H3K27ac intensity signal for all detected peaks ($n = 5.428$) from CUT&RUN data at 4 hpf from two biological replicates per condition in zebrafish embryos ($n = 75$ embryos/biological replicate) injected with RfxCas13d protein alone (Cas13d) or together with a mix of two gRNAs targeting *bckdk* mRNA (Bckdk KD). (**F**) Differential analyses of H3K27ac levels between zebrafish embryos injected with RfxCas13d protein alone or co-injected with 2 gRNAs targeting *bckdk* mRNA from CUT&RUN data described in (**E**). The Log2 fold change (FC) and *p* value associated are represented. Peaks with a significant decrease in H3K27ac are represented as purple dots (FC <−1.5 and *p* value <0.05). The number of downregulated peaks is indicated (*n*). *p* values were calculated using the Wald test. Source data are available online for this figure.

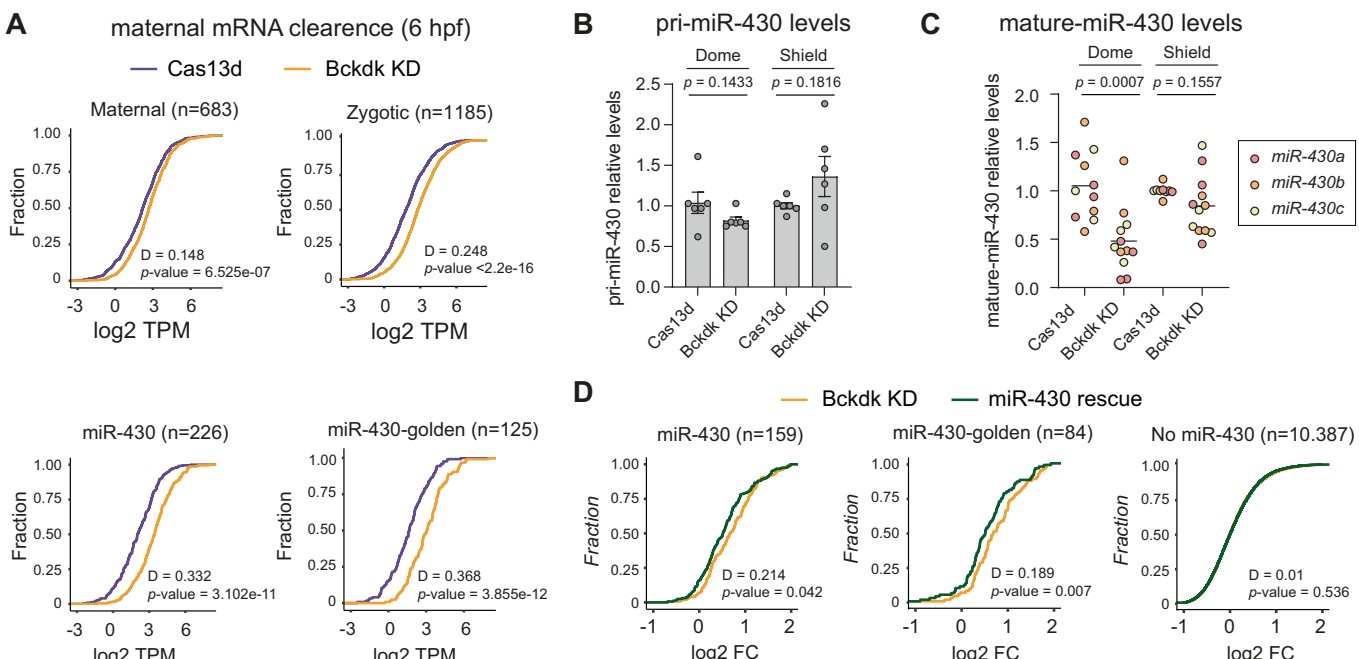

**Figure 4. miR-430 processing and activity is compromised upon *bckdk* mRNA depletion.**

(A) Cumulative distribution of mRNA levels (TPM; Transcript per million reads) in control (Cas13d; purple line) and *bckdk* mRNA knockdown embryos (Bckdk KD; orange line) that are degraded by four different pathways determined by Vejnar et al, 2019 (maternal, zygotic, miR-430) and described by Medina-Muñoz et al, 2021 (miR-430 golden candidates were defined as downregulated mRNAs, fourfold, from the comparison between WT and maternal-zygotic *dicer* mutant embryos at 6 hpf). Number of mRNA controlled by each pathway is shown (*n*). Distance (D) and *p* value calculated by Kolmogorov–Smirnov test are indicated. (B) RT-qPCR analysis showing relative primary miR-430 (pri-miR-430) transcript levels at 4.3 hpf (Dome) and 6 hpf (Shield). Results are shown as the averages ± standard error of the mean from three experiments with two biological replicates per experiment (*n* = 10 embryos/biological replicate) for RfxCas13d protein alone (Cas13d) and RfxCas13d plus two gRNAs targeting *bckdk* mRNA (Bckdk KD), respectively. *taf15* mRNA was used as normalization control. Exact *p* values are indicated above, unpaired *t*-test. (C) RT-qPCR analysis showing levels of mature miR-430 isoforms (miR430-a, red; miR430-b, orange; and miR430c, yellow) at 4.3 hpf (Dome) and 6 hpf (Shield). Results are shown as individual values and the mean from three experiments with two biological replicates per experiment (*n* = 10 embryos/biological replicate) for RfxCas13d protein alone (Cas13d) and RfxCas13d plus 2 gRNAs targeting *bckdk* mRNA (Bckdk KD). ncRNA *u4atac* was used as normalization control. Exact *p* values are indicated above, unpaired *t*-test). (D) Cumulative distribution of Log2 fold changes (Log2FC) of mRNAs degraded by miR-430 (miR-430 described by Vejnar et al, 2019, miR-430 golden described by Medina-Muñoz et al, 2021), or not dependent on the activity of this microRNA, from the comparison between zebrafish embryos injected with RfxCas13d protein and two gRNAs targeting *bckdk* mRNA, either alone (Bckdk KD, orange line) or co-injected with a mix of mature miR-430 isoforms (miR-430 rescue, green line) at 6 hpf. The number of mRNAs is shown (*n*). Distance (D) and *p* value calculated by the Kolmogorov–Smirnov test are indicated. Source data are available online for this figure.

et al, 2019). Within the zygotic mechanisms, miR-430 stands out as a critical microRNA (miRNA) facilitating the clearance of maternal RNAs (Giraldez et al, 2006; Vejnar et al, 2019; Bazzini et al, 2012). Since ZGA was notably affected upon *bckdk* mRNA KD and encouraged by our analysis at 4 hpf using SLAM-seq (Figs. 3B and EV6E,F), we wondered whether the maternal mRNA decay was also compromised later during the MZT. To address this, we performed a transcriptomic analysis at 6 hpf when the vast majority of maternal RNAs have been totally or partially eliminated (Vejnar et al, 2019; Medina-Muñoz et al, 2021). While the maternal mRNAs degraded by the maternal program only show a slight yet significant difference in their stability when comparing *bckdk* mRNA KD to embryos only injected with RfxCas13d (D, maximal vertical distance between the compared distribution = 0.148, *p* value = 6.5 e-07, Kolmogorov–Smirnov test), those eliminated by the zygotic program were more stable and displayed higher mRNA levels (D = 0.248, *p* value <2.2e-16, Kolmogorov–Smirnov test) (Fig. 4A). More strikingly, maternal RNAs whose clearance depended on miR-430 (Medina-Muñoz et al, 2021) were notably higher in the KD compared to the control embryos (D = 0.332 *p* value = 3.1e-11; golden targets: D = 0.368 *p* value = 3.8e-12, Kolmogorov–Smirnov

test) (Fig. 4A). Therefore, these results mimic a partial miR-430 lack-of-function molecular phenotype (Giraldez et al, 2006). To address whether the expression or processing of miR-430 was affected upon *bckdk* mRNA KD, we quantified the primary (pri-miR-430) and mature miR-430 by RT-qPCR in comparison to control embryos injected with RfxCas13d protein (Figs. 4B,C and EV7A). We observed that the level of pri-miR-430 was not significantly different in both control and Bckdk KD embryos at Dome or Shield stage (Fig. 4B). Interestingly, the mature miR-430 levels were lower at the dome stage in the *bckdk* mRNA KD compared to the control embryos, suggesting an alteration in the miR-430 processing. However, the amount of mature miR-430 was recovered at the shield stage (Figs. 4C and EV7B). Notably, injection of mature miR-430 in the *bckdk* KD embryos partially but significantly and specifically restored the levels of well-known mRNA targets of miR-430 that were less degraded upon *bckdk* mRNA KD (Fig. 4D), suggesting that delayed miR-430 processing was, at least in part, responsible for the reduced mRNA clearance.

Together, these results reveal that the lack of non-mitochondrial fraction of Bckdk (i) in part compromises the maternal RNA decay triggered by the zygotic program and (ii) more remarkably, delayed

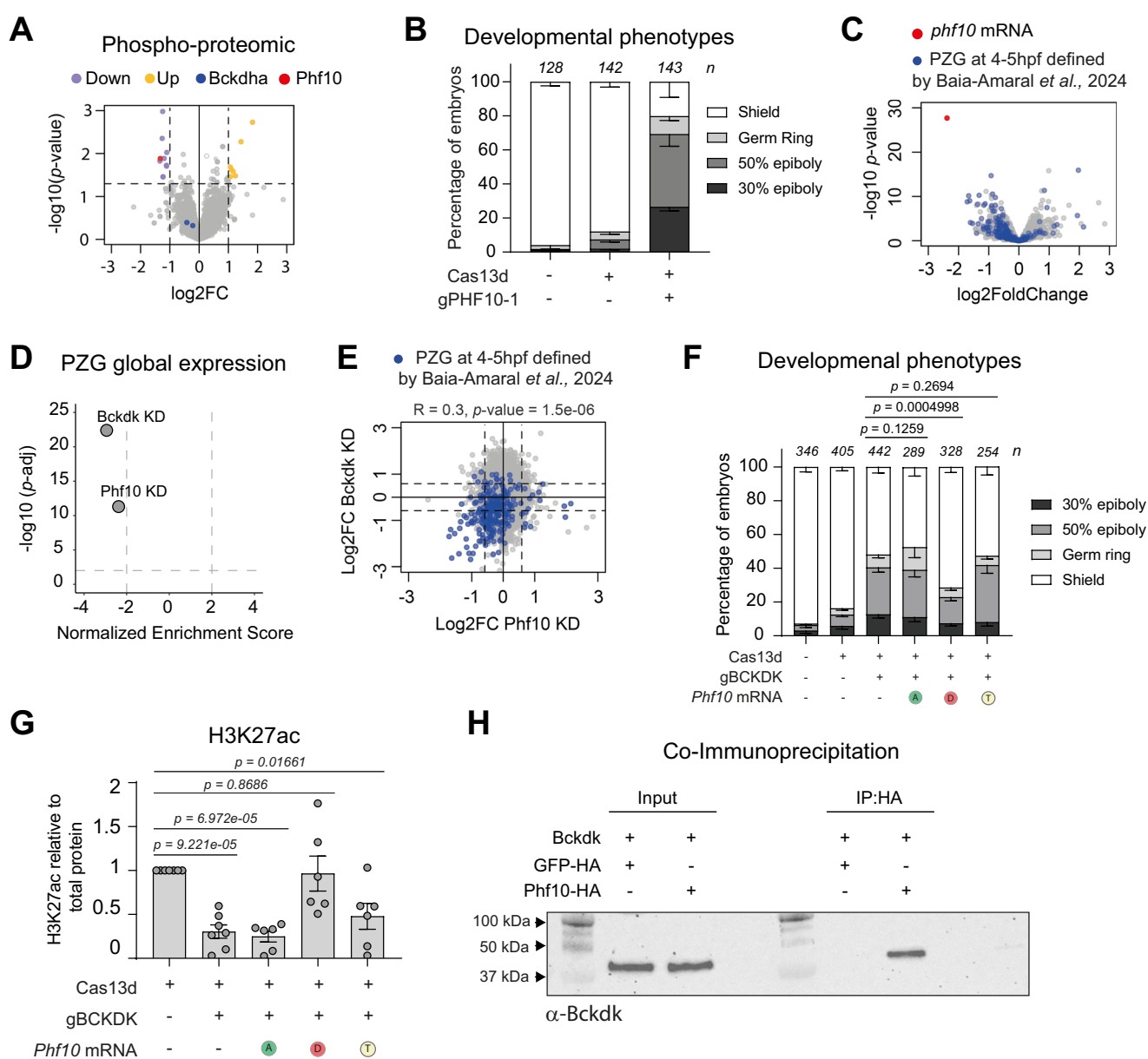

**A** Phospho-proteomic

**B** Developmental phenotypes

**C**

**D** PZG global expression

**E**

**F** Developmenal phenotypes

**G** H3K27ac

**H** Co-Immunoprecipitation

α-Bckdk

the processing of miR-430 and decreased the mRNA decay activity of this miRNA through the MZT.

## Bckdk regulates phospho-proteome during MZT and modulates Phf10 phosphorylation to control early development

Since Bckdk is a kinase, we asked whether the depletion of *bckdk* mRNA could affect the phosphorylation of proteins potentially involved in the regulation of MZT. To assess this, we performed a quantitative phospho-proteomic analysis in control embryos injected with Cas13d alone and in *bckdk* mRNA KD embryos at 4 hpf. First, 3336 phosphorylated proteins were detected in both phospho-proteomes and we observed 19 phospho-peptides from 16 proteins (2-fold change *p* value <0.05, empirical Bayes moderated *t*-test) to be differentially phosphorylated in the *bckdk* mRNA KD

compared to the control embryos (Fig. 5A; Phospho-proteins in control and KD embryos). 63% of the phospho-peptides (12 out of 19), belonging to 9 proteins, were significantly less phosphorylated upon Bckdk KD (Figs. 5A and EV8A; Dataset EV2). Notably, none of these nine proteins are predicted to localize in the mitochondria (Fig. EV8A), in contrast to a phospho-proteome upon Bckdk inhibition in rat liver where more than 55% of the identified targets localized in the mitochondria (White et al, 2018). Additionally, we observed that the phosphorylation state of the Bckdha motif (tyrighhS*tsddssa) described as the target of Bckdk in the mitochondria, did not significantly changed upon *bckdk* mRNA KD (Fig. 5A, blue dots). These results strengthened the hypothesis that the early developmental phenotype and MZT perturbation in the zebrafish embryos depleted of *bckdk* mRNA was due to the non-mitochondrial activity of this kinase (Fig. EV8A). Moreover, we also found seven phospho-peptides from seven proteins more

**Figure 5. Bckdk regulates phospho-proteome during MZT and modulates Phf10 phosphorylation to control early development and H3K27ac.**

(A) Scatter plot representing the fold change in phospho-peptides abundance levels and the associated *p* value from 4 biological replicates (*n* = 100 embryos/biological replicate) at 4 hpf in zebrafish embryos injected with RfxCas13d protein alone or with a mix of two gRNAs targeting *bckdk* mRNA. *p* values were calculated using a moderated *t*-test. Phospho-peptides abundance levels of Bckdha are depicted in blue. Phf10 phosphopeptide levels is represented in red. Less (Down) and more (Up) phosphorylated peptides are indicated in purple and yellow, respectively. Dashed lines indicated 1.5-fold in phosphorylated peptides levels and *p* value = 0.05. (B) Stacked barplots showing the percentage of phenotypes observed under the knockdown condition of *phf10* mRNA (injecting 3 ng/embryo of RfxCas13d protein and 1000 pg/embryo of a gRNA targeting *phf10* – gPHF10-1). The results are shown as the averages ± standard error of the mean of each developmental stage from four independent experiments. The phenotype selection criteria is described in Fig. 1C. Number of embryos evaluated (*n*) for each condition is shown. (C) Scatter plot representing the fold change in mRNA level and the associated *p* value from three biological replicates from RNA-Seq data (*n* = 20 embryos/biological replicate) at 4 hpf in zebrafish embryos injected with RfxCas13d protein alone or together with one gRNA targeting *phf10* mRNA. *p* values were calculated using the Wald test. *phf10* mRNA is represented in red. Pure zygotic gnes (PZG) mRNAs determined from Baia Amaral et al, 2024 are shown in blue. (D) Scatter plot of normalized enrichment score (NES) and adjusted *p* value associated of pure zygotic genes defined by Baia Amaral et al, 2024 upon the depletion of *bckdk* mRNA (Bckdk KD) or *phf10* mRNA (Phf10 KD) in zebrafish embryos (see Methods for details). NES and adjusted *p* value were calculated using GSEA (Subramanian et al, 2005; Mootha et al, 2003) (see Methods for details). (E) Scatter plot representing the log2 fold change in mRNA levels between embryos co-injected with RfxCas13d protein and gRNAs targeting *bckdk* mRNA (y axis) and embryos co-injected with RfxCas13d protein and a gRNA targeting *phf10* mRNA (x axis). Pure zygotic genes mRNAs (PZG) determined from Baia Amaral et al, 2024 data were represented in blue. Dashed lines indicated 1.5-fold in mRNA levels. Pearson correlation (*R* and *p* value) for PZG transcriptome between *bckdk* and *phf10* mRNA depletion is indicated. (F) Stacked barplots showing the percentage of phenotypes of zebrafish embryos under indicated conditions. *phf10* mRNA isoforms were co-injected (10 pg/embryo) together with RfxCas13d protein (3 ng/embryo) and a mix of 2 gRNAs (1000 pg/embryo) targeting *bckdk* mRNA. *phf10* mRNA isoforms are represented as; T Threonine, WT version, D: Aspatic acid, mimic the phosphorylation state mediated by Bckdk, A alanine, mimic a constitutively non-phosphorylated version. The results are shown as the averages ± standard error of the mean of each developmental stage from two independent experiments. The phenotype selection criteria was the same as described in Fig. 1C. Number of embryos evaluated (*n*) for each condition is shown. Exact *p* values are indicated above, $\chi^2$-test. (G) Barplots representing H3K27ac levels relative to total proteins as the averages ± standard error of the mean from 6–7 independent experiments under the rescue condition experiment similar to indicated in panel (F). Zebrafish embryos were collected at 4 hpf (*n* = 15–20 embryos/biological replicate) Exact *p* values are indicated above, Welch's *t*-test. (H) Co-immunoprecipitation assay targeting the HA epitope to analyze the interaction between Bckdk and Phf10 proteins. mRNAs encoding Phf10-HA or GFP-HA (50 pg per embryo) were co-overexpressed with *bckdk* mRNA (200 pg per embryo). At 6 hpf, proteins were immunoprecipitated using an anti-HA antibody, separated by SDS-PAGE, transferred to a membrane, and analyzed by immunoblotting with an anti-Bckdk antibody. A representative western blot from two independent experiments of Bckdk signal is shown under the indicated conditions. Source data are available online for this figure.

phosphorylated (Figs. 5A and EV8B; Dataset EV2) that may be an indirect effect triggered by the absence of Bckdk. In parallel, we performed quantitative proteomic and identified 4991 proteins, among which 11 were differentially accumulated upon *bckdk* mRNA KD, nine were downregulated, and two were more abundant (2-fold change *p* value <0.05, empirical Bayes moderated *t*-test; Fig. EV8C; Dataset EV2). None of the mRNAs encoding these proteins showed a significant change by SLAM-seq or RNA-seq data (Figs. 3A and EV3), and only 3 out of the 11 proteins were also detected in the phospho-proteome but without substantial differences between control and *bckdk* mRNA KD embryos. Similarly, only 2 out of 16 differentially phosphorylated proteins were found in the proteome, but without any significant change (Fig. EV8C; Dataset EV2). Therefore, this differential protein accumulation upon Bckdk depletion was likely due to translational or indirect post-translational regulation.

One of the proteins with the lowest level of phosphorylation upon *bckdk* mRNA KD was Phf10/Baf45a (Fig. 5A, red dot). Phf10 is a specific subunit of the Polybromo-associated BAF, pBAF, chromatin remodeling complex that belongs to the SWI/SNF family (Yuan et al, 2022; Kadoch and Crabtree, 2015; Brechalov et al, 2014) (Dataset EV2). Phf10 was less phosphorylated in a threonine (T16) in the *bckdk* mRNA KD embryos compared to the control (Fig. 5A; Dataset EV2). To address the role of Phf10 during MZT, we targeted its maternal mRNA using CRISPR-RfxCas13d. Interestingly, the embryos co-injected with Cas13d and gRNA targeting *phf10* maternal mRNA displayed similar epiboly defects comparable to the *bckdk* mRNA depletion (Fig. 5B and EV8D). RNA-seq and RT-qPCR data showed (i) a strong degradation of *phf10* mRNA in CRISPR-RfxCas13d injected embryos compared to controls (Cas13d injected only, Figs. 5C and EV8E), (ii) a general downregulation of PZG (Fig. 5C,D), and (iii) a positive correlation

between PZG transcriptome in *bckdk* and *phf10* mRNA depletions (Fig. 5E, R = 0.3, *p* value = 1.5e-06, Pearson correlation).

To define whether the phosphorylated state of Phf10 is one of the major driving forces of *bckdk* mRNA KD phenotype, we first defined the lowest dose of ectopic *phf10* mRNAs encoding for either WT or different mutants of Phf10 (T16A and T16D) tagged with an HA that did not trigger a developmental delay (Fig. EV8F,G). Strikingly, the Bckdk depletion phenotype was rescued by the expression of a mutant of *phf10* mRNA containing an aspartic acid in position (T16D) mimicking the Phf10 phosphorylation by Bckdk in WT conditions (Figs. 5F and EV8H,I). However, neither the wild-type nor the non-phosphorylatable mutant version with a threonine-to-alanine in position 16 (T16A) was able to rescue the *bckdk* mRNA KD phenotype (Fig. 5F and EV8H,I). Although the expression of all different Phf10 versions triggered a slight reduction in H3 levels (Fig. EV8J), strikingly, H3K27ac wild-type levels were only specifically restored by the T16D phospho-mimetic version of Phf10, indicating that the developmental rescue was associated with the recovery of normal histone acetylation (Figs. 5G and EV8K). To further investigate the interaction between Phf10 and Bckdk, we performed an immunoprecipitation assay in zebrafish embryos co-expressing Bckdk, Phf10-HA in zebrafish embryos and GFP-HA as a negative control. Pull-down of Phf10-HA revealed a specific interaction with Bckdk, which was not observed when using GFP-HA (Figs. 5H and EV8L).

Altogether, our data strongly suggest that (i) Phf10 is less phosphorylated in the *bckdk* mRNA KD and is an important factor during MZT contributing to the transcription of the genome during early zebrafish embryogenesis, (ii) a reduced T16 phosphorylation of Phf10 affects its function resulting in a decreased of H3K27ac and is partially responsible of the *bckdk* mRNA KD phenotype and iii) Phf10 and Bckdk interact in vivo.

## Discussion

While CRISPR-Cas DNA-targeting screenings have been widely used in zebrafish (Shah et al, 2015; Kroll et al, 2021; Hoshijima et al, 2019), a comprehensive and straightforward approach to elucidate the role of the maternally provided RNA during early development was required. Following our initial evidence that CRISPR-RfxCas13d efficiently depletes maternal RNA in zebrafish and other vertebrate embryos (Kushawah et al, 2020), we and others continued to refine and employ this system (Hernandez-Huertas et al, 2022; Moreno-Sánchez et al, 2025; da Silva Pescador et al, 2024; Kushawah et al, 2024; Ferreira et al, 2025; Kim and Hutchins, 2025). Here, we introduce the application of our optimized CRISPR-RfxCas13d system for maternal RNA-targeted screenings, aiming to uncover new genes regulating early vertebrate development. First, this work demonstrates an increased efficiency and robustness in mRNA level reduction (Fig. 1D, depletion median: 92%, lowest decreased: 58%) using an enhanced gRNA design (Hernandez-Huertas et al, 2022; Wessels et al, 2020) and purified RfxCas13d protein, which exceeds our original reported efficacy (Kushawah et al, 2020). This suggests that, in general, the lack of phenotype of the screening candidates is not likely due to a low KD efficiency (Fig. 1C,D). Furthermore, our findings indicate that our screening method can identify novel regulators of early embryogenesis. Indeed, we uncovered eight mRNAs out of 49 screened, whose depletion induced developmental deficiencies (Figs. 1C and EV1B–E). Notably, seven of these genes were associated with epiboly defects (Fig. 1C), whereas one (MIBP) was linked to developmental abnormalities later in embryogenesis (Fig. EV1B–E). Additionally, we did not observe collateral activity in our knockdowns (Figs. EV4A,B and EV5G) and, in fact, the *bckdk* mRNA KD phenotype could be complemented by injecting a non-targetable version of its mRNA (Fig. 2A,B). Indeed, we have recently demonstrated that collateral effects triggered by CRISPR-RfxCas13d in zebrafish embryos can only be detected upon targeting extremely abundant ectopic transcripts (Moreno-Sánchez et al, 2025). Consistently, mRNAs targeted in this study did not trigger collateral effects (Figs. EV4 and EV5G). This absence of collateral activity in zebrafish embryos could be due to our transient approach (RNP complexes) versus the constitutive expression of CRISPR-RfxCas13d components employed in mammalian cells, where RNA-targeting is accumulated over the time (Shi et al, 2023; Tong et al, 2022; Li et al, 2023; Moreno-Sánchez et al, 2025; Hart et al, 2025). Nevertheless, we acknowledge certain limitations of using CRISPR-RfxCas13d in zebrafish embryos. As a transient approach involving RNP complexes injection, it can result in mosaic phenotypes with incomplete penetrance, in contrast to the more uniform effects typically observed in maternal and zygotic (MZ) mutants. Similar incomplete penetrance has been reported with other knockdown approaches, such as morpholinos or antisense oligonucleotides used at non-toxic concentrations (Pauli et al, 2015; Stainier et al, 2017). However, generating MZ mutants is highly time-consuming and is not always feasible due to embryonic lethality or infertility (Abrams and Mullins, 2009; Kushawah et al, 2024). Moreover, while CRISPR-RfxCas13d efficiently targets maternal mRNAs to disrupt early development, it cannot deplete preexisting maternal proteins, a limitation shared with all RNA-targeting methods. Despite these constraints, here we show that CRISPR-RfxCas13d is a versatile and effective tool for elucidating gene function during early teleost development.

Second, gene expression regulation during the MZT occurs at multiple levels, including chromatin remodeling and modification, transcription, mRNA stability, translation, etc. (Murphy et al, 2018; Wike et al, 2021; Miao et al, 2022; Pálfy et al, 2020; Vastenhouw et al, 2010; Vejnar et al, 2019; Giraldez et al, 2006; Bazzini et al, 2012, 2014; Riesle et al, 2023; Bazzini et al, 2016). However, the significance of post-translational regulation and, particularly, protein phosphorylation state in this context remains understudied. Our research reveals a subset of kinases or proteins related to this activity and a phosphatase whose maternal mRNA KD leads to epiboly defects. While our investigations primarily concentrated on Bckdk for its role in ZGA and MZT, we also identified Mknk2a, a kinase potentially regulating ZGA, alongside other kinases potentially implicated in various early developmental processes beyond ZGA. Interestingly, our screening identified three genes linked to calcium signaling, with Cab39l and Calm2a exhibiting similar transcriptomic patterns upon KD, hinting at their interconnected molecular functions (Fig. EV1G). Notably, Calm2a and Calm1a KD triggered the depletion of other Calmodulins with 100% identity (Fig. EV2). Whether observed phenotypes are the consequence of individual or combined KD of Calmodulins is something that remains to be determined using more specific gRNAs. Nevertheless, given the established links between calcium signaling and critical developmental milestones such as epiboly, egg activation, fertilization, and cell cycle regulation (Webb and Miller, 2003; Sahu et al, 2017), exploring the roles of these calcium-dependent kinases could enhance our comprehension of early embryogenesis regulation. Consequently, our findings illuminate the crucial role of protein phosphorylation in governing the initial stages of embryogenesis.

Third, through the first phospho-proteome performed at ZGA, we demonstrated that the phosphorylation state of Bckdha, the main target of Bckdk in the mitochondria, was unaltered (Fig. 5A; Dataset EV2) in Bckdk-depleted embryos and consistently, BCAA levels were similar to the control condition (Suryawan et al, 1998; Murashige et al, 2022) (Fig. 2D,E). Besides, functional mitochondria are largely maternally provided, with an average of $\sim 2 \times 10^7$ of these organelles per one-cell stage zebrafish embryos, without renovation during MZT (Stackley et al, 2011; Otten et al, 2016), and no significant phosphorylation decrease in mitochondrial proteins is detected upon *bckdk* mRNA depletion. Interestingly, total Bckdk protein levels significantly decreased, but less than its mRNA upon KD conditions (log$_2$ fold: $-0.2$, $p$ value $= 0.03$, Dataset EV2). However, this is a remarkable reduction since only 81 out of the 4996 detected proteins (1.6%) show a significant alteration (up or down-regulated) upon *bckdk* mRNA KD, indicating that, although smaller than its maternal mRNA, the reduction of Bckdk as protein is robust and likely affecting the cytosolic Bckdk fraction. Consistently, Bckdk is highly maternally provided, and at the four-cell stage, Bckdk is among the top 5% most detected proteins and the top 15% within mitochondrial proteins (Yan et al, 2023). Therefore, this accumulated evidence reinforces the notion that maternal Bckdk is likely localized to extremely abundant mitochondria during MZT and strongly suggests that *bckdk* mRNA KD in zebrafish embryos did not affect the known Bckdk mitochondrial activity described in mammalian cells (Suryawan et al, 1998; Murashige et al, 2022). Notably, non-mitochondrial Bckdk has a

known role in cancer promotion and cell proliferation and migration (Tian et al, 2020; Xue et al, 2017; Xu et al, 2023) that, in turn, may be related to stem cell and pluripotency-like states (Chen and He, 2015; Hepburn et al, 2019). Thus, our approach allowed us to uncover a novel and specific role for non-mitochondrial Bckdk during early zebrafish embryogenesis, since a maternal mutant of this kinase, in the hypothetical case of being viable and fertile (Joshi et al, 2006), would have also affected its mitochondrial function.

Fourth, for the first time, we have employed SLAM-seq in conjunction with CRISPR-RfxCas13d-mediated KD in zebrafish embryos. This innovative approach has enabled us to comprehensively investigate the role of Bckdk during early embryogenesis. Our findings demonstrate that *bckdk* mRNA KD diminishes the accumulation of PZG and a large set of maternal-zygotic genes that are normally transcribed at 4 hpf (Fig. 3B). Interestingly, we also identified another set of genes that showed an increase in their mRNA levels and a smaller group which experienced an earlier expression. Remarkably, the vast majority of these groups of genes are usually transcribed during the MZT, indicating that, alongside ZGA, *bckdk* mRNA KD disrupts the expression dynamics of early expressed genes without affecting others later activated. This dynamic misregulation may involve transcriptional and/or mRNA stability components, as was recently shown for zygotically transcribed miR-430 targets (Baia Amaral et al, 2024). Upregulated genes were involved in translation and protein biogenesis, suggesting a possible attempt at gene expression compensation. Furthermore, we noted a modest yet detectable alteration in chromatin accessibility, which, did not exhibit a clear bias towards any specific transcription factor binding sites influenced by Bckdk depletion (Fig. 3D). These results suggest that the impact of *bckdk* mRNA KD could extend to general transcription factors or chromatin modifiers, rather than being confined to specific pioneer regulators where changes in the genome are more prominent upon their loss-of-function (Miao et al, 2022; Riesle et al, 2023; Pálfy et al, 2020). Indeed, *bckdk* mRNA depletion reduced H3K27ac, a chromatin modification required for ZGA (Chan et al, 2019; Miao et al, 2022; Sato et al, 2019) (Figs. 3E,F and EV6L,M). Among the known proteins involved in H3K27 acetylation in zebrafish (Chan et al, 2019), only CREBBPB was detected in our proteome or phospho-proteome at 4 hpf, but its levels showed no significant changes (Dataset EV2). Thus, we cannot exclude that the activity, level, or phosphorylation status of one or more of these proteins (EP300A, EP300B, CREBBPA, and CREBBP) might be affected by the absence of Bckdk. Instead, Phf10/Baf45a, a unique component of the pBAF chromatin remodeling complex, which is part of the broader SWI/SNF family (Yuan et al, 2022; Brechalov et al, 2014; Kadoch and Crabtree, 2015) and necessary to maintain the proliferation state of different cell types (Lessard et al, 2007; Krasteva et al, 2017), was identified as a key potential target of Bckdk. Upon Bckdk depletion, Phf10 was less phosphorylated at T16, a residue whose phosphorylation is conserved in human (Stuart et al, 2015). CRISPR-Cas13d-mediated *phf10* mRNA KD replicated both the developmental phenotype and PZG down-regulation observed with *bckdk* mRNA KD. Remarkably, only the phospho-mimetic version of Phf10 mutated at T16 significantly rescued the epiboly defects found in embryos depleted on Bckdk (Fig. 5F) and also restored the wild-type levels of H3K27ac (Fig. 5G). This suggests that one of the main Bckdk targets regulating

MZT is Phf10 and that Bckdk modulates pBAF activity during early development. Although pBAF is rather a reader of H3K27ac than a complex involved in the deposit of this epigenetic mark, its alteration can lead to reduced transcription and a lower histone acetylation (Basurto-Cayuela et al, 2024; Xiao et al, 2021; Carcamo et al, 2022). Interestingly, Phf10 phosphorylation has been reported to affect its stability in mammalian cells (Tatarskiy et al, 2017; Sheynov et al, 2020). However, we did not detect Phf10 in our proteome, likely due to the differential level of resolution of proteomic and phospho-proteomic approaches. Notably, the over-expression of Phf10 causes epiboly delay independently of the T16 phosphorylation (Fig. EV8G) status, indicating that a correct level of this protein is important to progress through development. Furthermore, Phf10 and Bckdk physically interact in zebrafish embryos (Figs. 5H and EV8L), likely in the cytosol since Phf10 does not localize in the mitochondria. Together with the short-term phospho-proteome analysis 4 hpf, where indirect effects are unlikely, these findings strongly suggest that Phf10 is a direct substrate of Bckdk.

In addition to Smarca2 and Smarca4a, which are well-known to be phosphorylated during mitosis to remove them from chromatin, we also detected phosphorylation of other SWI/SNF complex members including Arid1a, Arid1b, Arid2, Pbrm1, and Smarcb1, at 4 hpf (Mashtalir et al, 2018; Kadoch and Crabtree, 2015) (Dataset EV2), suggesting that protein phosphorylation could be an important post-translational regulator of the activity of this complex during MZT. Furthermore, our phospho-proteomic analysis detected phosphorylated proteins related to MZT such as Nanog, Brd3a, Brd3b, Brd4 and H1m (Lee et al, 2013; Chan et al, 2019; Kushawah et al, 2020; Riesle et al, 2023; Pálfy et al, 2020; Pérez-Montero et al, 2013). Although none of these proteins were identified as differentially phosphorylated between WT and Bckdk-depleted embryos, the phosphorylation status of these chromatin and transcription factors may still play a crucial role in regulating their function during the ZGA and MZT, independently of Bckdk.

Interestingly, we identified other proteins that did exhibit altered phosphorylation patterns after *bckdk* mRNA KD which may also contribute to disruptions in ZGA such as Znf148 (Fig. EV8A) with a predicted transcription factor activity and Lbh, a well-known transcriptional cofactor (Briegel and Joyner, 2001). Another interesting candidate is Ago3a, which displayed a decreased phosphorylation of a tyrosine residue in the *bckdk* mRNA KD (Figs. 5A and EV8A). Other phosphorylated residues in different Ago proteins have been associated with Argonaute–mRNA interactions (Bibel et al, 2022; Quévillon Huberdeau et al, 2017). The temporal reduction of miR-430 processing observed in Bckdk-depleted embryos during MZT triggers an increased stability of a subgroup of targets of this miRNA (Fig. 4A,C). However, Ago3a reduced phosphorylation might also be related to a potential decrease in the regulatory efficacy of miR-430. Intriguingly, we could not find any alteration in the phosphorylation state of Dicer1 or protein levels of Pasha, known proteins involved in miRNA processing, detected in our phospho-proteome and proteome, respectively (Dataset EV2). However, we cannot exclude the possibility that the resolution of these approaches did not allow us to detect changes in other proteins that could be responsible for the delayed biogenesis of miR-430. Further experiments would need to be done to deeply analyze the relation between miRNA processing and protein phosphorylation mediated by Bckdk.

In conclusion, we have been able to show that (i) CRISPR-RfxCas13d is a powerful tool to perform maternal mRNA KD screenings (49 genes) that combined with RNA-seq, SLAM-Seq, ATAC-seq, CUT&RUN, proteomics, and phospho-proteomics allow to investigate the downstream effects of gene perturbation across multiple regulatory layers. In addition, we have demonstrated that Bckdk regulates ZGA and MZT not only in zebrafish but also in other teleosts such as medaka. Finally, we have revealed that this regulation is, at least, mediated through a reduced biogenesis and activity of miR-430 and a post-translational phosphorylation of Phf10, whose alteration negatively affects H3K27ac and the activation of the zebrafish genome. The influence of Bckdk on the two main events of MZT, zygotic genome activation and maternal mRNA clearance, emphasizes the critical role of post-translational phosphorylation as a regulatory mechanism during embryogenesis and underscores the importance of further exploring the functions of kinases and phosphatases controlling early development.

# Methods

### Reagents and tools table

| Reagent/resource | Reference or source | Identifier or catalog number |
|---|---|---|
| **Experimental models** | | |
| Zebrafish wild-type strain AB/Tübingen (AB/Tu) | | |
| Medaka wild-type strain (iCab) | | |
| **Recombinant DNA** | | |
| pT3TS-Bckdk-HA | This study | |
| pT3TS-Phf10-HA-WT | This study | |
| pT3TS-Phf10-HA-16A | This study | |
| pT3TS-Phf10-HA-16D | This study | |
| pCS2-GFP | Giraldez Lab | |
| pT3TS-RfxCas13d-HA | Kushawah et al, 2020 | Addgene plasmid #141320 |
| pET28b-RfxCas13d | Kushawah et al, 2020 | Addgene plasmid #141322 |
| **Antibodies** | | |
| Anti-HA | Roche | #11867423001 |
| Anti-H3K27Ac | Abcam | #ab177178 |
| Anti-H3 | Santa Cruz Biotechnology | #sc-517576 |
| Anti-H3 | Abcam | #ab1791 |
| anti-mouse HRP-labeled | Sigma-Aldrich | #A5278 |
| anti-rabbit HRP-labeled | Abcam | #ab6721 |
| **Oligonucleotides and other sequence-based reagents** | | |
| Please refer to Dataset EV1 | | |
| **Chemicals, enzymes and other reagents** | | |
| FavorPrep Gel/PCR Purification Kit | FavorGen Biotech. Corp. | #FAGC1014 |

| Reagent/resource | Reference or source | Identifier or catalog number |
|---|---|---|
| AmpliScribe T7 Flash | Epicentre, Lucigen | #ASF3257 |
| Qubit RNA BR (Broad-Range) assay kit | Thermo Fisher | #Q10210 |
| QuikChange Multi Site-Directed Mutagenesis kit | Agilent | #200514 |
| XbaI | New England Biolabs | #R0145S |
| NcoI-HF | New England Biolabs | #R3193S |
| NotI-HF | New England Biolabs | #R3189S |
| BglII | New England Biolabs | #R0144S |
| mMESSAGE mMACHINE™ T3 | Thermo Fisher Scientific | #AM1348 |
| RNeasy Mini Kit | Qiagen | #74104 |
| TRIzol | Canvax | #AN1100 |
| iScript cDNA synthesis kit | Bio-Rad | #1708890 |
| SYBR® Premix-Ex-Taq (Tli RNase H Plus) | Takara | #RR420A |
| mirVana™ miRNA Isolation Kit | Thermo Fisher Scientific | #AM1561 |
| iScript Select cDNA Synthesis Kit | Bio-Rad | #708896 |
| DNA/RNA Shield | Zymo Research | #R1100-50 |
| Direct-zol RNA Miniprep Kit | Zymo Research | #R2050 |
| SMART-seq v4 Ultra Low Input RNA Kit | Takara | #634891 |
| Nextera XT DNA Library Preparation Kit | Illumina | #FC-131-1096 |
| DNA/RNA UD Indexes Set A | Illumina | #20027213 |
| Ampure XP bead | Beckman Coulter | A63882 |
| NEBNext Ultra II Directional RNA Library Prep Kit for Illumina | New England Biolabs | #E7490L |
| NEBNext Multiplex Oligos for Illumina | New England Biolabs | #E6440S |
| SPRIselect bead | Beckman Coulter | #B23318 |
| Library Compatibility Kit | Singular Genomics | #700141 |
| TruSeq Stranded mRNA Library Prep Kit | Illumina | #20020594 |
| TruSeq RNA Single Indexes Set A | Illumina | 20020492 |
| TruSeq RNA Single Indexes Set B | Illumina | 20020493 |
| SLAMseq Kinetics Kit —Anabolic Kinetics Module | Lexogen | #061 |
| QuantSeq 3' mRNA-seq Library Prep Kit | Lexogen | #144.96 |
| Tn5 | Dorothy Hodgkin Proteomics and Biochemistry (CABD, Seville, Spain) | |

| Reagent/resource | Reference or source | Identifier or catalog number |
|---|---|---|
| MinElute PCR Purification Kit | Qiagen | #28004 |
| NEBNext HighFidelity 2× PCR Master Mix | New England Biolabs | #M0541S |
| inactivated FBS | Fisher Scientific | #15575309 |
| pluriStrainer Mini 40 µM | pluriSelect | #43-10040-50 |
| CUTANA ChIC/ CUT&RUN Kit | EpiCypher | #14-1048 |
| 1X dsDNA HS Assay Kit | Thermo Fisher | #Q33230 |
| NEBNext® End Repair Module | New England Biolabs | E6050S |
| Klenow Fragment (3 → 5′ exo-) | New England Biolabs | #M0212S |
| T4 DNA ligase | New England Biolabs | #M0202S |
| Q5® High-Fidelity DNA Polymerase | New England Biolabs | #M0491S |
| KAPA Library Quantification Kit | Roche | #KK4824 |
| Leucine | Sigma-Aldrich | #L8000 |
| Isoleucine | Sigma-Aldrich | #I2752 |
| 13C6-Leucine | Cambridge Isotopes | #605239 |
| Valine | Sigma-Aldrich | #72-18-4 |
| L-Valine-1-13C | Sigma-Aldrich | #490164 |
| Dansyl chloride | Sigma-Aldrich | #311155 |
| Heptylamine | Sigma-Aldrich | #126802 |
| 10% GX Stain-FreeTM Fast CastTM Acrylamide Solutions | Bio-Rad | #1610183 |
| 12% GX Stain-FreeTM Fast CastTM Acrylamide Solutions | Bio-Rad | #1610185 |
| ClarityTM Western ECL Substrate | Bio-Rad | #1705061 |
| QUBIT protein kit | Thermo Fisher | #Q33211 |
| tris(2-carboxyethyl) phosphine | Thermo Fisher | #T2556 |
| 2-Chloroacetamide | Thermo Fisher | #148410050 |
| Mass spectrometry grade trypsin | Promega Gold | #V5280 |
| Fluorometric peptide assay | Thermo Scientific | #23290 |
| TMTpro 16plex reagents | Thermo Scientific | #A44520 |
| High-SelectTM Fe-NTA Phosphopeptide Enrichment Kit | Thermo Scientific | #A32992 |
| trifluoroacetic acid (TFA) | Sigma-Aldrich | #33077 |
| Pierce Co-Immunoprecipitation Kit | Thermo Scientific | #26149 |
| DAPI | Sigma-Aldrich | #D9542 |
| Phalloidin | Invitrogen | #A12379 |
| Tris-HCl | Sigma-Aldrich | #77-86-1 |
| NaCl | Sigma-Aldrich | #7647-14-5 |
| NaHCO3 | Sigma-Aldrich | #144-55-8 |

| Reagent/resource | Reference or source | Identifier or catalog number |
|---|---|---|
| Phosphatase\| Protease inhibitor EDTA free | Thermo Scientific | #78440 |
| DTT | Sigma-Aldrich | #D9779 |
| KCl | Sigma-Aldrich | #P9541 |
| NP-40 | Thermo Scientific | #85124 |
| Hepes | Sigma-Aldrich | #H3375 |
| MgSO4•7H2O | Sigma-Aldrich | #230391 |
| Ca(NO3)2 | Sigma-Aldrich | #13477-34-4 |
| PBS | Sigma-Aldrich | #P4417 |
| SDS | Sigma-Aldrich | #822050 |
| Tween 20 | Sigma-Aldrich | #P1379 |
| sodium deoxycholate | Sigma-Aldrich | #302-95-4 |
| PFA | Sigma-Aldrich | #158127 |
| **Software** | | |
| R | The R Project for Statistical Computing | https://www.r-project.org/ |
| Prism | GraphPad | https://www.graphpad.com/ |
| Illustrator | Adobe | https://www.adobe.com |
| Mfuzz (v. 2.62.0) | Bioconductor | https://www.bioconductor.org/packages/release/bioc/html/Mfuzz.html |
| RNAfold software | RNAfold webserver | http://rna.tbi.univie.ac.at//cgi-bin/RNAWebSuite/RNAfold.cgi |
| MEGA | MEGA | https://www.megasoftware.net/ |
| NIS-Elements D 4.60.00 software | Nikon | https://www.nikon.com/products/microscope-solutions/support/download/software/imgsfw/nis-d_v4300032.htm |
| Fiji | ImageJ | https://imagej.net/software/fiji/ |
| ImageLab | Bio-Rad | www.bio-rad.com/en-ch/product/image-lab-software |
| CFX manager software | Bio-Rad | https://www.bio-rad.com/es-es/sku/1845000-cfx-manager-software?ID=1845000 |
| NextSeq Control Software (v.2.2.0.4) | Illumina | https://support.illumina.com/downloads/nextseq-system-suite-v2-2-0.html |
| NextSeq RTA (v.2.4.11) | Illumina | https://support.illumina.com/sequencing/sequencing_instruments/nextseq-500/downloads.html |
| bcl2fastq2 (v.2.20) | Illumina | https://support.illumina.com/downloads/bcl2fastq-conversion-software-v2-20.html |
| sgdemux (v.1.2.0) | Singular Genomics | https://github.com/Singular-Genomics/singular-demux |
| STAR aligner (v.2.7.3a) | Github | https://github.com/alexdobin/STAR |
| RSEM (v.1.3.0) | Github | https://deweylab.github.io/RSEM/ |
| Deseq2 (v.1.42.0) | Bioconductor | https://bioconductor.org/packages/release/bioc/html/DESeq2.html |
| UpSetR (v. 1.4.0) | CRAN | https://cran.r-project.org/web/packages/UpSetR/index.html |
| gRNADesign | Github | https://github.com/hzuzu/gRNADesign |
| slamdunk map module (v.0.4.3) | Github | https://t-neumann.github.io/slamdunk/ |
| clusterProfiler (v.4.10.0) | Bioconductor | https://www.bioconductor.org/packages/release/bioc/html/clusterProfiler.html |

| Reagent/resource | Reference or source | Identifier or catalog number |
|---|---|---|
| ShinyGO | Ge et al, 2019 | https://bioinformatics.sdstate.edu/go/ |
| Bowtie2 (v.2.3.5) | Github | https://github.com/BenLangmead/bowtie2 |
| Samtools (v.1.9) | Samtools | https://www.htslib.org/ |
| Bedtools (v.2.29.2) | Bedtools | https://bedtools.readthedocs.io/en/latest/index.html |
| MACS2 | Github | https://hbctraining.github.io/Intro-to-ChIPseq/lessons/05_peak_calling_macs.html |
| Deeptools (v.3.5) | Deeptols | https://deeptools.readthedocs.io/en/3.5.5/ |
| Gimmemotifs (v.3.5) | Gimmemotifs | https://gimmemotifs.readthedocs.io/en/master/reference.html |
| UCSC Genome Browser | University of California, Santa Cruz. Genomics institute | https://genome.ucsc.edu/ |
| GREAT (v.3.0.0) | Stanford University | https://great.stanford.edu/great/public/html/ |
| Proteome Discoverer 2.4 | Thermo Fisher Scientific | https://www.thermofisher.com/us/en/home/industrial/mass-spectrometry/liquid-chromatography-mass-spectrometry-lc-ms/lc-ms-software/multi-omics-data-analysis/proteome-discoverer-software.html |
| Limma (v.3.54.2) | Bioconductor | https://bioconductor.org/packages/release/bioc/html/limma.html |
| biomaRt (v.2.54.1) | Bioconductor | https://bioconductor.org/packages/release/bioc/html/biomaRt.html |
| NormalyzerDE (v.3.18) | Bioconductor | https://www.bioconductor.org/packages/release/bioc/html/NormalyzerDE.html |
| Rstatix (v.0.7.2) | CRAN | https://cran.r-project.org/web/packages/rstatix/index.html |
| ggplot2 (v.3.4.2) | CRAN | https://cran.r-project.org/web/packages/ggplot2/index.html |
| dplyr (v.1.1.2) | CRAN | https://cran.r-project.org/web/packages/dplyr/index.html |
| Dgof (v.1.4) | CRAN | https://cran.r-project.org/web/packages/dgof/index.html |
| **Other** | | |
| Nikon DS-F13 digital camera | Nikon | https://www.microscope.healthcare.nikon.com/de_EU/products/cameras/ds-fi3 |
| Agilent 2100 Bioanalyzer | Agilent | https://www.agilent.com/en/product/automated-electrophoresis/bioanalyzer-systems/bioanalyzer-instrument |
| TapeStation | Agilent | https://www.agilent.com/en/product/automated-electrophoresis/tapestation-systems/tapestation-instruments |
| NextSeq 500 | Illumina | https://www.illumina.com/documents/products/appnotes/appnote-nextseq-500-wgs.pdf |
| NextSeq 2000 | Illumina | https://www.illumina.com/systems/sequencing-platforms/nextseq-1000-2000.html |
| Q-Exactive Plus Mass Spectrometer | Thermo Scientific | https://assets.thermofisher.com/TFS-Assets/CMD/brochures/BR-63890-LC-MS-Q-Exactive-Plus-Orbitrap-BR63890-EN.pdf |
| CFX Duet Real-Time PCR System | Bio-Rad | https://www.bio-rad.com/sites/default/files/2022-05/Bulletin_3283.pdf |
| Chemidoc MP | Bio-Rad | https://www.bio-rad.com/sites/default/files/2024-03/Bulletin_6873.pdf |
| Trans-Blot Turbo Transfer System | Bio-Rad | https://www.bio-rad.com/sites/default/files/webroot/web/pdf/lsr/literature/10000071567.pdf |
| Orbitrap Eclipse Tribrid Mass Spectrometer | Thermo Scientific | https://assets.thermofisher.com/TFS-Assets/CMD/Specification-Sheets/ps-65451-orbitrap-eclipse-tribrid-ms-ps65451-en.pdf |
| Microscopy Zeiss LSM 880 Airscan | Zeiss | https://health.ucdavis.edu/physiology/research/zeiss-microscope/documents/ZEISS-LSM-880.pdf |
| Andor EMCCD iXOn DU 879 camera | Oxford Instruments | |

## Zebrafish maintenance and embryo production

All experiments conducted using zebrafish at CABD adhere to national and European Community standards regarding the ethical treatment of animals in research and have been granted approval by the Ethical committees from the University Pablo de Olavide, CSIC and the Andalusian Government. Zebrafish wild-type strains AB/Tübingen (AB/Tu) were maintained and bred under standard conditions (Westerfield, 1995). Wildtype zebrafish embryos were obtained through natural mating of AB/Tu zebrafish of mixed ages (5–18 months). Selection of mating pairs was random from a pool of ten males and ten females. Zebrafish embryos were staged in hpf as described (Kimmel et al, 1995). Zebrafish experiments at Stowers Institute were done according to the IACUC-approved guidelines. Zebrafish embryos for microinjections were coming from random parents (AB, TF and TLF, 6–25 months old) mating from 4 independent strains of a colony of 500 fish. The embryos were pooled from random 24 males and 24 females for each set of experiments.

## Medaka maintenance and embryo production

All experiments involving medaka adhered to national and European Community standards for animal use in research and received approval from the Ethical Committees of the University Pablo de Olavide, CSIC, and the Andalusian Government. The wild-type medaka strain (iCab) was kept and bred under standard conditions. Fifteen to twenty pairs of 4-month-old males and females were randomly selected and crossed to produce embryos for the experiments described here. Embryos were injected at the single-cell stage following standard methods. Their progression through developmental stages was determined in hours post fertilization (hpf) using the described procedures (Iwamatsu, 2004).

## Selection of candidates for maternal screening

Maternal screening candidates were selected according to their levels of maternally provided RNA, translation levels, and functional annotation. mRNAs highly maternally provided (>50 TPM at 0 hpf in rib0 (Medina-Muñoz et al, 2021)) and highly translated at 2 hpf (>130 rpkm in ribosome profiling data (Bazzini et al, 2014; Chan et al, 2019)) were soft-clustered by Mfuzz Software in R (v. 2.62.0) (Kumar and Futschik, 2007) using ribosome

profiling data at 0, 2, 5, and 12 hpf (Bazzini et al, 2014; Chan et al, 2019). We selected mRNAs that showed high translational activity between 0 and 5 hpf, followed by a reduction at later time points (Dataset EV1). Among them, mRNA encoding kinases or phosphatases and factors directly related to their activity based on gene description by ZFIN database information (Bradford et al, 2022) were selected for the maternal screening.

## Guide RNA design and RfxCas13d protein and mRNA production

All guide RNAs (gRNAs) and Cas13d protein (RfxCas13d) employed in this study were designed according to a previous detailed protocol for zebrafish embryos (Hernandez-Huertas et al, 2022). Three gRNAs targeting the CDS were designed per each mRNA. All designed gRNAs for maternal screening were chemically synthesized by Synthego (Synthego Corp., CA, USA). gRNAs for *phf10* mRNA knockdown were generated by fill-in-PCR with a universal primer and three specific primers (Dataset EV1) with the Q5 High-Fidelity DNA Polymerase. Purify PCR product using FavorPrep Gel/PCR Purification Kit were used as a template for in vitro gRNA production using AmpliScribe T7 Flash Transcription Kit (Epicentre, Lucigen #ASF3257). Then, gRNAs were precipitated according to previous detailed protocol and resuspended in nuclease free water (Hernandez-Huertas et al, 2022). gRNAs were quantified using Qubit RNA BR (Broad-Range) assay kit (Thermo Fisher, #Q10210).

Constructions for *bckdk* and *phf10* mRNA rescue and over-expression experiments were designed by cloning the CDS sequence with specific primers (Dataset EV1) using cDNA from zebrafish embryos at 4 hpf or purchasing the CDS sequence from Integrated DNA Technologies (https://eu.idtdna.com/), respectively. To generate *bckdk* dead-kinase version, the Bckdk protein sequence from zebrafish was aligned with the Bckdk proteins from human, mouse and rat using MEGA software with the ClustalW algorithm. Analog residue of Tyrosine 331 in zebrafish (Tyrosine-348) was selected for substitution by an Alanine to generate a dead-kinase version since it has been reported that this point substitution reduces 95% of enzymatic activity (Wynn et al, 2000; Singh et al, 2024). Dead-kinase version of *bckdk* mRNA and different phosphorylation state isoforms for *phf10* mRNA containing HA epitope were generated by site-directed mutagenesis using QuikChange Multi Site-Directed Mutagenesis kit (#200514, Agilent), following manufacturer's instructions and specific primers (Dataset EV1). Plasmid constructions (Dataset EV1) were digested with *XbaI* for 2 h at 37 °C to linearize the DNA and then used for in vitro transcription using the mMESSAGE mMACHINE™ T3 (#AM1348, Thermo Fisher Scientific) following the manufacturer's protocols. mRNA products were purified using the RNeasy Mini Kit (#74104, Qiagen) and quantified using Nanodrop. All mRNAs used in this study were previously validated by western blot analysis, and experimental injections were adjusted according to it.

## Zebrafish embryo injection and image acquisition

One-cell stage zebrafish embryos were injected with 1 nL containing 3 µg/µL of purified RfxCas13d protein and 1000 ng/µL of gRNAs (individual or a mix of two to three gRNAs, see figure legends for details in each experiment). Between 10 and 50 pg per embryo of ectopic *gfp* and *bckdk* mRNA were injected for the rescue experiment. Between 10 and 50 pg of each version of *phf10* mRNA were injected for overexpression or rescue experiments. miR-430 duplexes (Dataset EV1) were purchased from IDT (https://eu.idtdna.com/) and resuspended in RNAse-free water. About 1 nL of 2.5 µM miR-430 duplex solution (equimolar mix of three duplexes miR-430a, miR-430b, and miR-430c) was injected at the one-cell stage, using single-use aliquots.

Zebrafish embryos' phenotypes were quantified at 2, 4, 6, and 24 hpf. Zebrafish embryo phenotypes pictures were performed using a Nikon DS-F13 digital camera and images processed with NIS-Elements D 4.60.00 software. GFP fluorescence signal were quantified using Fiji (ImageJ) software, 15 embryos in three images (five embryos/image) per condition were employed.

## Medaka embryo injection and image acquisition

One-cell stage medaka embryos were co-injected with 2–3 nL of a solution containing 3 µg/µL of purified RfxCas13d protein and 1000 ng/µL gRNA. Medaka embryos phenotypes were quantified at 24 and 48 hpf. Medaka embryo phenotypes pictures were performed using a Nikon DS-F13 digital camera and images were processed with NIS-Elements D 4.60.00 software.

## RT-qPCR

Total RNA to analyse mRNA levels by RT-qPCR were extracted from ten embryos per biological replicate. Embryos were collected at the specific timepoint and snap-frozen in liquid nitrogen. Total RNA was extracted using the standard TRIzol protocol as described in the manufacturer's instructions (Thermo Fisher Scientific). Then, cDNA was generated using 1000 ng of total purified RNA following the iScript cDNA synthesis kit (1708890, Bio-Rad) manufacturer's protocol. About 2 µl of a 1:5 cDNA dilution was used together with forward and reverse primers per each mRNA (2 µM; Dataset EV1) and 5 µl of SYBR® Premix-Ex-Taq (Tli RNase H Plus, Takara) in a 10 µl reaction. PCR cycling profile consisted in a denaturing step at 95 °C for 30 s and 40 cycles at 95 °C for 10 s and 60 °C for 30 s. *taf15* mRNA was used as a control for sample normalization.

## Primary miR-430 and mature miR-430 quantification

Ten embryos per biological replicate were collected at 4.3 and 6 hpf to analyse by RT-qPCR primary miR-430 and mature miR430 levels. Embryos were snap-frozen in liquid nitrogen at the indicated timepoint. Total RNA and RNA enriched in small RNA species were extracted using mirVana™ miRNA Isolation Kit (#AM1561, Thermo Fisher Scientific) following the manufacturer's instructions. cDNA for measuring primary-miR430 was generated using 200 ng of total purified RNA following the iScript cDNA synthesis kit (#1708890, Bio-Rad) manufacturer's protocol. Then, 2 µl of cDNA was used together with specific forward and reverse primers for primary miR-430 (Hadzhiev et al, 2023) (2 µM; Dataset EV1) and 5 µl of SYBR® Premix-Ex-Taq (Tli RNase H Plus, Takara) in a 10 µl reaction. *taf15* mRNA was used as a control for sample normalization.

About 100 ng of enriched small RNA fraction was used to generate cDNA following the iScript Select cDNA Synthesis Kit

(#708896, Bio-Rad) following the manufacturer's protocol and the specific primer 5′-GCAGGTCCAGTTTTTTTTTTTTTTTC-TACCCC-3′ (Dataset EV1). About 2 µl of cDNA was used together with specific primers for each mature miR-430 (a, b, and c) and the previous primer (2 µM; Dataset EV1), and 5 µl of SYBR® Premix-Ex-Taq (Tli RNase H Plus, Takara) in a 10 µl reaction. ncRNA u4atac (Khatri et al, 2023) was used for sample normalization in mature microRNA RT-qPCR.

## RNA-seq libraries and analysis

Twenty zebrafish embryos per biological replicate were collected at 4 or 6 hpf and snap-frozen in 300 µl of DNA/RNA Shield (#R1100-50, Zymo Research). Total RNA was then extracted using Direct-zol RNA Miniprep Kit (#R2050, Zymo Research) following the manufacturer's instructions. Ten medaka embryos per biological replicate were collected and dechorionated as described in Porazinski et al, 2010 (Porazinski et al, 2010) at 6.5 or 8 hpf. Total RNA was extracted using the standard TRIzol protocol as described in the manufacturer's instructions (Thermo Fisher Scientific). Total RNA from zebrafish and medaka embryos was quantified using the Qubit fluorometric quantification (#Q10210, Thermo).

For RNA-seq libraries for maternal screening associated with Figs. 1 and EV2, 3, cDNA was generated from 1.25 ng of high-quality total RNA (except for Cas13d control and *bckdk* mRNA KD where cDNA was generated from 2.5 ng of high-quality total RNA), as assessed using the Bioanalyzer (Agilent), according to manufacturer's directions for the SMART-seq v4 Ultra Low Input RNA Kit (Takara, #634891) at a 1/8th reaction volume and using the Mantis (Formulatrix) nanoliter liquid-handling instrument to pipette the reagents for cDNA synthesis (except for Cas13d control and *bckdk* mRNA KD samples where 1/4th of the reaction volume were used). Libraries were generated manually (except for *cab39l* and *calm2a* mRNA KD samples where libraries were generated with the Mosquito HV Genomics (SPT Labtech) nanoliter liquid-handling instrument), using the Nextera XT DNA Library Preparation Kit (Illumina, FC-131-1096) at 1/8th reaction volumes (except for Cas13d control and *bckdk* mRNA KD samples where 1/4th of the reaction volume were used) paired with IDT for Illumina DNA/RNA UD Indexes Set A (Illumina, 20027213), and purified using the Ampure XP bead-based reagent (Beckman Coulter, Cat. No. A63882). Resulting short fragment libraries were checked for quality and quantity using the Bioanalyzer and Qubit Fluorometer (Thermo Fisher). Equal molar libraries were pooled, quantified, and sequenced on a High-Output flow cell of an Illumina NextSeq 500 instrument using NextSeq Control Software 2.2.0.4 with the following read length: 70 bp Read1, 10 bp i7 Index and 10 bp i5 Index. Following sequencing, Illumina Primary Analysis version NextSeq RTA 2.4.11 and Secondary Analysis version bcl2fastq2 (v. 2.20) were run to demultiplex reads for all libraries and generate FASTQ files. Exception: libraries for Ppp4r2a, gfp and rfp KD employed in the maternal screening in Fig. 1 were performed similarly to libraries for medaka samples and Phf10 KD associated with Figs. 2 and 5 (see below).

RNA-seq libraries for medaka samples and *phf10* mRNA KD associated to Figs. 2 and 5, respectively were generated from 100 ng (or ≤100 ng) of high-quality total RNA, as assessed using the Bioanalyzer (Agilent), were made according to the manufacturer's

directions using a 25-fold (or 100-fold) dilution of the universal adapter and 13 cycles (or 16 cycles) of PCR per the respective masses with the NEBNext Ultra II Directional RNA Library Prep Kit for Illumina (NEB, Cat. No. E7760L), the NEBNext Poly(A) mRNA Magnetic Isolation Module (NEB, Cat. No. E7490L), and the NEBNext Multiplex Oligos for Illumina (96 Unique Dual Index Primer Pairs) (NEB, Cat. No. E6440S) and purified using the SPRIselect bead-based reagent (Beckman Coulter, Cat. No. B23318). The resulting short fragment libraries were checked for quality and quantity using the Bioanalyzer and Qubit Flex Fluorometer (Life Technologies). Equal molar libraries were pooled, quantified, and converted to process on the Singular Genomics G4 with the SG Library Compatibility Kit, following the "Adapting Libraries for the G4—Retaining Original Indices" protocol. The converted pool was sequenced on an F3 flow cell (Cat. No. 700125) on the G4 instrument with the PP1 and PP2 custom index primers included in the SG Library Compatibility Kit (Cat. No. 700141), using Instrument Control Software 23.08.1-1 with the following read length: 8 bp Index1, 100 bp Read1, and 8 bp Index2. Following sequencing, sgdemux (v. 1.2.0) was run to demultiplex reads for all libraries and generate FASTQ files.

All raw reads were demultiplexed into Fastq format, allowing up to one mismatch using Illumina bcl2fastq2 v2.18. Reads were aligned to UCSC genome danRer11 or to UCSC genome oryLat2 with the STAR aligner (v. 2.7.3a), using Ensembl 102 gene models. TPM values were generated using RSEM (v. 1.3.0). Fold change for each gene was calculated using the DESeq2 package (v. 1.42.0) after filtering genes with a count of 20 reads in all control libraries. Differential expression genes were selected, setting a corrected *p* value <0.05 and a fold change >3. Common downregulated genes were calculated using UpSet intersection visualization, the R package UpSetR (Lex et al, 2014) (v. 1.4.0). Only combinations with more than 50 commonly downregulated genes are shown in Fig. EV1G.

RNA-seq libraries for samples at 6 hpf in zebrafish embryos associated with Fig. 4 were generated from 100 ng of high-quality total RNA, as assessed using the Bioanalyzer (Agilent). Libraries were made according to the manufacturer's directions for the TruSeq Stranded mRNA Library Prep Kit (Illumina, Cat. No. 20020594), and TruSeq RNA Single Indexes Sets A and B (Illumina Cat. No. 20020492 and 20020493) and purified using the Ampure XP bead-based reagent (Beckman Coulter, Cat. No. A63882). Resulting short fragment libraries were checked for quality and quantity using the Bioanalyzer (Agilent) and Qubit Fluorometer (Life Technologies). Equal molar libraries were pooled, quantified, and sequenced on a High-Output flow cell of an Illumina NextSeq 500 instrument using NextSeq Control Software (v. 4.0.1) with the following read length: 76 bp Read1, 6 bp i7 Index. Following sequencing, Illumina Primary Analysis version NextSeq RTA 2.11.3.0 and bcl-convert-3.10.5 were run to demultiplex reads for all libraries and generate FASTQ files.

## Off-target analysis

To identify potential off-target effects in our maternal screening conditions, we used the mismatch module from https://github.com/hzuzu/gRNADesign to compute the number of mismatches for each of the three gRNAs used in each knockdown condition against

the full *Danio rerio* transcriptome. Genes with three or fewer mismatches were plotted in Fig. EV2.

## SLAM-seq libraries and analysis

s4-UTP (SLAMseq Kinetics Kit—Anabolic Kinetics Module; #061, Lexogen) were injected at 25, 50, and 75 pmol/embryo in one-cell-stage embryos under red light and kept in the dark until the desired timepoint. Due to the toxicity effect of s4-UTP in combination with the RfxCas13d protein, 25 mM aliquots were selected as the final concentration for the SLAM-Seq experiment. These reduced levels of s4-UTP could potentially compromise the coverage of label reads compared to optimized conditions in zebrafish embryos (Ugolini et al, 2024; Baia Amaral et al, 2024; Bhat et al, 2023). 25 embryos were collected at 4 hpf and snap-frozen in 500 μl of Trizol in tubes protected from light. Total RNA was extracted and then alkylated with iodoacetamide under dark conditions using SLAMseq Kinetics Kit—Anabolic Kinetics Module (#061, Lexogen) following the manufacturer's instructions.

3′-end mRNA sequencing libraries were generated, according to the manufacturer's instructions, from 200 ng of alkylated total RNA (except no-s4U Cas13d-alone sample, which was 87 ng), using the QuantSeq 3′ mRNA-seq Library Prep Kit for Illumina UDI Bundle (FWD, Lexogen GmbH, cat. no. 144.96). ERCC-92 RNA spike-ins were added at equal final molarity to all samples. For PCR amplification, 14 cycles were performed with the Lexogen Unique Dual Index (UDI) 12-nt Index Set B1. The resulting libraries were checked for quality and quantity using the Qubit Fluorometer (Life Technologies) and the Bioanalyzer (Agilent). Libraries were pooled, re-quantified, and sequenced as 100-bp single reads on the NextSeq 2000 (Illumina). Following sequencing, Illumina Primary Analysis RTA and bcl2fastq2 were run to demultiplex reads for all libraries and generate FASTQ files. Adapters were cut from FASTQ files using TRIMGALORE with the following parameters: -a AGATCGGAAGAGCACACGTCTGAACTCCAGTCAC; --length 30. All trimmed reads were mapped to the danRer11 genome using default parameters of the slamdunk map module (v. 0.4.3), then filtered with the filter module. Zebrafish ENS102 annotation for protein-coding genes 3′UTRs and non-coding genes coordinates were used as reference to get reads per 3′UTR or non-coding gene. To estimate background $T > C$ due to SNPs, a VCF file was obtained from Cas13d alone (no-S4U) and Cas13+gRNAs (no-s4U) with slamdunk snp module with the following parameters: -f 0.2. Only then, total Reads and labeled reads (containing at least 2 $T > C$ conversion events) were obtained with the slamdunk count module using the VCF file created in previous steps (unlabeled reads were estimated by subtracting labeled from total read counts).

Read counts per million (CPM) were obtained using DESeq2. Fold change for each gene and its associated *p* value was calculated using the DESeq2 package (v. 1.42.0) after filtering genes with a count of 20 reads in all control libraries for unlabeled and labeled data. One replicate from Bckdk KD and three from Cas13d control were discarded from the analysis due to their extremely low sequence depth and spike-ins over-representation.

## Gene set enrichment analysis and gene ontology analysis

Differential expression analysis from RNA-seq or SLAM-seq data generated were ranked according to its fold change compared to control conditions and used as input for gene set enrichment

analysis (GSEA) (Subramanian et al, 2005; Mootha et al, 2003), resulting in a normalized enrichment score with its adjusted *p* value. GSEA analyses were performed using the R Package clusterProfiler (v. 4.10.0). Conditions with a GSEA Normalized Enrichment score $<-2$ or $>2$ and a *p* adjusted $<0.01$ were considered significant.

Enrichment of Gene Ontology Biological Process terms was calculated using ShinyGO (Ge et al, 2019) terms with a false discovery rate (FDR), corrected *p* value $<0.05$, and with more than 20 genes represented are considered as enriched.

## ATAC-seq libraries and analysis

ATAC-seq assays were conducted following standard protocols (Buenrostro et al, 2013; Fernández-Miñán et al, 2016), with slight adjustments. A total of 80 embryos per experimental condition were collected at 4 hpf and dissolved in Ginzburg Fish Ringer buffer without calcium (55 mM NaCl, 1.8 mM KCl, 1.15 mM NaHCO₃) by gentle pipetting and shaking at 1100 rpm for 5 min. The samples were then centrifuged for 5 min at 500×*g* at 4 °C. Following removal of the supernatant, the embryos were washed with cold PBS and subsequently resuspended in 50 μl of Lysis Buffer (10 mM Tris-HCl, pH 7.4, 10 mM NaCl, 3 mM MgCl₂, 0.1% NP-40) by gentle pipetting. The equivalent volume of 70,000 cells was centrifuged for 10 min at 500×*g* at 4 °C, after which it was resuspended in 50 μl of transposition reaction mixture, comprising 1.25 μl of Tn5 enzyme and TAGmentation Buffer (10 mM Tris-HCl pH 8.0, 5 mM MgCl₂, 10% w/v dimethylformamide), followed by an incubation period of 30 min at 37 °C. Subsequent to TAGmentation, DNA was purified using the MinElute PCR Purification Kit (Qiagen) and eluted in 10 μl. Libraries were generated via PCR amplification using NEBNext HighFidelity 2× PCR Master Mix (NEB). The resultant libraries were purified again using the MinElute PCR Purification Kit (Qiagen), multiplexed, and then sequenced on a HiSeq 4000 pair-end lane, yielding ~100 million 49-bp pair-end reads per sample.

ATAC-seq reads were aligned to the GRCz11 (danRer11) zebrafish genome assembly using Bowtie2 (v.2.3.5), and those pairs separated by more than 2 kb were removed. The Tn5 cutting site was determined as the position −4 (minus strand) or +5 (plus strand) from each read start, and this position was extended 5 bp in both directions. Conversion of SAM alignment files to BAM was performed using Samtools 1.9 (Li et al, 2009). Conversion of BAM to BED files, and peak analyses, such as overlaps or merges, were carried out using the Bedtools 2.29.2 suite. Conversion of BED to BigWig files was performed using the genomecov tool from Bedtools and the wigToBigWig utility from UCSC (Quinlan and Hall, 2010)

ATAC peaks were called using MACS2, and peaks were pooled together to calculate fold change for each peak using Deseq2 package (v1.3.0). Differential accessibility regions were selected, setting a corrected *p* value $<0.05$ and a fold change $>2$-fold. Bed files were normalized according to the relative number of reads into peaks. Heatmaps and average profiles of ATAC-Seq data were performed using computeMatrix, plotHeatmap and plotProfile tools from the Deeptools 3.5 toolkit. Transcription Factor motifs were calculated using Gimmemotifs (v. 0.18.0) with standard parameters. Differential ATAC peak groups (setting a corrected *p* value $<0.05$ and a fold change $>1.5$ or 3-fold) from danRer11 were

converted to danRer7 using the Liftover tool of the UCSC Genome Browser (Hinrichs, 2006). These peaks were assigned to genes using the GREAT 3.0.0 tool with the basal plus extension association rule, with the following parameters: 5 kb upstream, 1 kb downstream, 1000 kb of maximum extension. These genes were associated with SLAM-Seq differential labeled data and represented as a violin plot for those conditions with >10 genes per group.

## CUT&RUN libraries and analysis

A total of 75 embryos per experimental condition and per antibody reaction were collected at 4 hpf and transferred to a small beaker with 10 mL of E3 medium and 1.5 mL of inactivated FBS (fetal bovine serum). Embryos were cleaned three times with 10 mL of Danieau (17.4 mM NaCl, 0.21 mM KCl, 0.12 mM MgSO4•7H2O, 0.18 mM Ca(NO$_3$)$_2$, 1.5 mM Hepes) 0.5X with inactive FBS. Then, embryos were transferred to a 2 mL Eppendorf tube containing 1 mL of deyolking Buffer (55 mM NaCl, 1.8 mM KCl, and 1.25 mM NaHCO$_3$). Embryos were disintegrating through carefully pipetting up and down 20 times. Samples were incubated for 5 min at 1000 rpm at 25 °C and then, centrifugated for 5 min at 500×$g$ at 4 °C and resuspended in 1 ml of cold PBS. Samples were filtered using a pluriStrainer Mini 40 µM (pluriSelect). About 75,000 cells per condition and antibody reaction were used for the CUT&RUN reaction following the CUTANA ChIC/CUT&RUN Kit (EpiCypher, 14-1048) manufacturer's protocol. In brief, Concanavalin A Magnetic Beads were first activated by washing twice with Bead Activation Buffer. Cells were bound to the activated beads and then incubated at 4 °C overnight with 0.5 µg of the corresponding antibody (IgG antibody was provided in the kit. Anti-H3K27Ac, #ab177178, Abcam). Then, samples were washed twice with cell permeabilization buffer and incubated with pAG-MNAse for 1 h at 4 °C. Samples were washed twice again with cell permeabilization buffer and incubated with 100 mM of calcium chloride for 20 min at 4 °C, reaction was stopped using Stop Buffer at 37 °C for 10 min. DNA was purified using SPRIselect beads provided in the kit and quantified using a Qubit fluorometer and 1X dsDNA HS Assay Kit (Q33230). CUT&RUN libraries were prepared following standard protocols. DNA end-repair was performed using the NEBNext® End Repair Module (NEB E6050S) at 20 °C for 30 min. 1 µl of spike-in *E. coli* DNA (from a 1:10 dilution of the amount provided in the kit, CUTANA ChIC/CUT&RUN Kit) was added to the master mix library reaction to use it for normalization. Samples were cleaned using magnetic bead purification (AMPure XP beads) with two 75% ethanol washes and eluted in TE buffer (10 mM Tris-HCl, pH 8.0, 1 mM EDTA). A-tailing was carried out using Klenow Fragment (3′→5′ exo-, NEB M0212S) at 37 °C for 30 min, followed by bead purification. Adapters were ligated overnight at 16 °C with T4 DNA ligase (NEB), using Illumina-compatible adapters adjusted for input concentration. Post-ligation, libraries underwent bead purification. Libraries were amplified using NEB Q5® High-Fidelity DNA Polymerase (M0491S) with optimized cycling conditions. Final libraries were cleaned with beads and quantified using the KAPA Library Quantification Kit (Roche). Libraries were pooled equimolarly based on concentration and size profiles, analyzed by TapeStation, and submitted for sequencing.

Sequencing reads were processed similarly to ATAC-seq analyses. Reads were aligned to the GRCz11 (danRer11) zebrafish genome assembly using Bowtie2 (v2.3.5), and read pairs separated by more than 2 kb were removed. SAM files were converted to BAM using Samtools (v1.9), and BAM to BED using Bedtools (v2.29.2). Peak calling was performed with MACS2, and peaks were pooled for fold change calculations using the DESeq2 package (v1.30.0). Differential regions were selected using $p$ value <0.05 and fold change >1.5. Bed files were normalized based on the *E. coli* spike-in control. IgG control signals were used to filter nonspecific peaks. Visualization, including heatmaps and average profiles, was performed with the Deeptools (v3.5) suite.

## Leucine-isoleucine and valine quantification

To analyse amino acid levels, 50 embryos were collected at 4 hpf and resuspended in 1 ml of deyolking Buffer (55 mM NaCl, 1.8 mM KCl, 1.25 mM NaHCO$_3$). Samples were incubated at 25 °C for 5 min with orbital shaking at 1100 rpm and then centrifuged at 300×$g$ for 30 s. Supernatant was removed and samples were washed with 1 ml of Wash Buffer (110 mM NaCl, 3.5 mM KCl, 2.7 mM NaHCO$_3$), shaking the tubes at 1100 rpm for 2 min. Samples were centrifuged at 300×$g$ for 30 s, and the supernatant was resuspended in 350 ul of methanol with 10 µg/ml of leucine used as an internal control measure or in 0.1 N HCl for valine analysis. Samples were sonicated for 15 min at 4 °C to enhance the supernatant and centrifuged again. Finally, samples were centrifuged for 10 min at 16,000×$g$ and the supernatant were stored at −80 °C until sample processing.

Leucine and isoleucine (Sigma-Aldrich), 13C6-leucine (Cambridge Isotopes), valine (Sigma-Aldrich), L-valine-1-13C (Sigma-Aldrich) and samples were derivatized by dansyl chloride (Sigma-Aldrich). About 3.5 mM amino acids were dissolved in 20 µl of pH 9.5 250 mM sodium bicarbonate buffer, 16 times more dansyl chloride was added and incubated at room temperature for 1 h. The reaction was quenched by adding 11 µl of 0.5 M heptylamine (Sigma-Aldrich) at room temperature for 15 min. About 40 µl of each sample was dried by vacuum centrifugation and reconstituted with 20 µl of 250 mM, pH 9.5 sodium bicarbonate buffer, then mixed with 2 µl 56 mM dansyl chloride for 1 h at room temperature. The reaction was quenched by adding 11 µl of 0.5 M heptylamine for 10 min at room temperature. 400 nM of derivatized 13C6-Leucine was spiked into 2, 10, 25, 50, 75, and 100 nM of derivatized leucine, respectively, as well as each derivatized sample before LC/MS analysis.

Amino acid samples were analyzed on a Q-Exactive Plus Mass Spectrometer (Thermo Scientific) equipped with a Nanospray Flex Ion Source and coupled to a Dionex UltiMate 3000 RSCLnano System. A 75 µm i.d. analytical microcapillary column was packed in-house with 100 mm of 1.9 µm ReproSil-Pur C18-AQ resin (Dr. Masch). AgileSLEEVE (Analytical Sales & Products) was used to maintain column temperature at 50 °C. The UPLC solutions were 0.1% formic acid in water for buffer A (pH 2.6) and 0.1% formic acid in acetonitrile for buffer B. The chromatography gradient has a 5-min loading time at 20%B, 23 min from 20 to 35%; 2 min to reach 100% B; and 10 min washing at 100% B. The nano pump flow rate was set to 0.5 ul/min. The Q-Exactive was set up to run a Parallel Reaction Monitoring (PRM) method with an inclusion list of 371.1736 m/z and 365.1535 m/z with HCD at 40%. Each standard curve concentration point and each sample were analyzed in triplicate. The peak area ratio between leucine and 13C6-leucine and the

peak ratio between valine and L-valine-1-13C was used to quantify the leucine and valine amounts in each sample.

## Protein sample preparation and Western blot

About 20–25 embryos were collected at 4 or 6 hpf and washed with 200 µl of Deyolking Buffer (55 mM NaCl, 1.8 mM KCl, 1.25 mM $NaHCO_3$). Then, embryos were lysed by adding another 200 µl of Deyolking Buffer and by pipetting up and down. Samples were incubated at 25 °C for 5 min with orbital shaking and centrifuged at 300×g for 30 s. Supernatant were removed and the pellet was washed by adding 300 µl of Wash Buffer (110 mM NaCl, 3.5 mM KCl, 10 mM TRIS-HCl, pH = 7.4). Samples were centrifuged at 300×g for 30 s and the supernatant was resuspended in 10 µl of SDS-PAGE buffer (160 mM Tris-HCl, pH 8, 20% Glycerol, 2% SDS, 0.1% bromophenol blue, 200 mM DTT). SDS-PAGE electrophoresis was conducted using 10 or 12% TGX Stain-FreeTM Fast CastTM Acrylamide Solutions (Bio-Rad). Subsequently, the protein gels were activated with a Chemidoc MP (Bio-Rad) and transferred to Nitrocellulose membranes via the Trans-Blot Turbo Transfer System (Bio-Rad). Following this, the membranes were blocked at room temperature for 1 h using Blocking Solution (5% fat-free milk in 50 mM Tris-HCl, pH 7.5, 150 mM NaCl (TBS) with 1% Tween 20).

The primary antibody Anti-H3K27Ac (#ab177178, Abcam) was diluted 1:5000, Anti-HA (#11867423001, Roche) was diluted 1:1000 and Anti-H3 #sc-517576 (Santa Cruz Biotechnology) in Bckdk KD experiments (Fig. EV6L,M) or #ab1791 (Abcam) in Phf10 rescue experiments. (Fig. EV8J) was diluted 1:500 or 1:5000, respectively. The secondary antibody, anti-mouse HRP-labeled (#A5278, Sigma-Aldrich), was diluted 1:5000 and anti-rabbit HRP-labeled (#ab6721, Abcam) was diluted 1:2000, all in Blocking Solution.

The membrane was then incubated with the primary antibody overnight at 4 °C. Post-primary antibody incubation, the membrane underwent three brief washes with TBS with 1% Tween 20 (TTBS) for 15 min each, followed by incubation with the secondary antibody for 60 min at room temperature. Washes were performed similarly to those with the primary antibody. Detection was accomplished using ClarityTM Western ECL Substrate (Bio-Rad), and images were captured with a ChemiDoc MP (Bio-Rad). H3K27ac and H3 intensity were calculated using ImageLab software and normalized to total Stain-Free total lane volumes (Gürtler et al, 2013).

## Protein extraction for proteomic analysis

About 100 embryos collected at 4 hpf were resuspended in 1 ml of Deyolking Buffer (55 mM NaCl, 1.8 mM KCl, and 1.25 mM $NaHCO_3$) by shaking the tubes at 1100 rpm for 5 min. Samples were centrifuged at 300×g for 30 s. Supernatant were removed and pellet washed with 1 ml of Wash Buffer (110 mM NaCl, 3.5 mM KCl, and 2.7 mM $NaHCO_3$) and shaken for 2 min at 1100 rpm. After centrifugation at 300×g for 30 s, samples were resuspended in homogenization buffer (50 mM Tris (pH 7.5), 1 mM DTT, phosphatase|protease inhibitor EDTA free). Samples were briefly sonicated (twice for 10 s) and centrifuged at 16,900×g for 10 min at 4 °C. The supernatant was kept on ice and protein concentration were measured with QUBIT protein kit (#Q33211, Thermo Fisher).

## Digestion for mass spectrometry

Samples were reduced by adding tris(2-carboxyethyl) phosphine (TCEP at 1 M) to 5 mM final at room temperature for 30 min. To carboxymethylate reduced cysteine residues, 5 µl of 2-chloroacetamide (CAM, made fresh at 0.5 M) were added and samples were incubated for 30 min protected from light at room temperature. Six volumes of pre-chilled (at −20 °C) acetone were added, and the precipitation was let to proceed overnight. Samples were next centrifuged at 8000×g for 10 min at 4 °C. The tubes were carefully inverted to decant the acetone without disturbing the protein pellets, which were air-dried for 2–3 min.

The precipitated proteins were dissolved in 100 µl of 50 mM TEAB, mass spectrometry grade trypsin (Promega Gold) was added at 1:40 w/w and the digestion were let to proceed at 37 °C overnight. The digested protein samples were centrifuged at 16,000×g for 30 min and transferred to new tubes. A fluorometric peptide assay (Pierce) was performed on 10 µl of each digested sample according to the manufacturer's instructions. Peptide amounts ranged from 30 to 120 µg.

## Tandem mass tag (TMT) labeling

Immediately before use, the vials containing frozen TMTpro 16plex reagents (Thermo Scientific) were let to warm up at room temperature. To each of the 0.5 mg vials, 20 µl of anhydrous acetonitrile were added. The TMT reagents were let to dissolve for 5 min with occasional vortexing, then the tubes were briefly centrifuged to gather the solution. For each sample, 20 µg were measured out based on the peptide assay, and the volumes were adjusted to 100 µl with 100 mM TEAB, then mixed with the 20 µl in TMT vials. The four replicate samples for the Cas13d control condition and Bckdk KD condition, were labeled with TMTpro--129N, -129C, -130C -130C, -131N, -131C, -132N, and -132C, respectively. The labeling reaction was allowed to proceed for 1 h at room temperature. To test labeling efficiency, 1 µl of samples were analyzed by LC/MSMS over a 2-h C18 reverse-phase (RP) gradient on an Orbitrap Eclipse Tribrid Mass Spectrometer (Thermo Scientific) with a FAIMS Pro interface, equipped with a Nanospray Flex Ion Source, and coupled to a Dionex UltiMate 3000 RSCLnano System. The TMT labeling levels were >99% for all detected peptides. The 12 differentially labeled samples (30 µl each) were combined in a new tube, and the resulting volume was reduced using a SpeedVac concentrator (Savant) to less than 10 µl (about 2 h).

## Phosphopeptide enrichment

High-SelectTM Fe-NTA Phosphopeptide Enrichment Kit (Thermo Scientific) was used for phosphopeptide enrichment. Lyophilized TMT-labeled peptides were suspended in 200 µl of binding buffer and loaded onto an equilibrated Fe-NTA spin column. After 30 min incubation, the supernatant was collected through centrifugation at 1000×g. About 100 µl of elution buffer were added to the spin column, and the eluate was collected through centrifugation at 1000×g. Both supernatant and eluate were dried immediately in a SpeedVac concentrator to less than 10 µl.

## High pH reverse-phase fractionation

The dried supernatant TMT-labeled peptide mixture was resuspended in 300 µl of 0.1% trifluoroacetic acid (TFA). One high pH fractionation cartridge (Pierce, cat. # 84868) was placed on a new 2.0 ml sample tube and 300 µl of the TMT-labeled peptide mixture were loaded onto the column. After centrifuging at 3000×g for 2 min, the eluate was collected as the "flow-through" fraction. The loaded cartridge was placed on a new 2.0 ml sample tube, washed with 300 µl of ddH20, and the eluate collected as "wash" fraction. An additional round of washing was performed using 300 µl of 5% acetonitrile in 0.1% TFA to remove unreacted TMT reagent. A total of 8 HpH RP fractions were collected by sequential elution in new sample tubes using 300 µl of 10, 12.5, 15, 17.5, 20, 22.5, 25, and 50% acetonitrile in 0.1% TFA. The solvents were evaporated to dryness using vacuum centrifugation. Dried samples were resolubilized in 44 µl of buffer A (5% acetonitrile in 0.1% formic acid, FA) before LC-MS analysis.

## Multiplexed mass spectrometry analysis

TMT-labeled peptides were analyzed on an Orbitrap Eclipse Tribrid Mass Spectrometer (Thermo Scientific) with a FAIMS Pro interface, equipped with a Nanospray Flex Ion Source, and coupled to a Dionex UltiMate 3000 RSCLnano System. Peptides (22 µl for each HpH RP fraction) were loaded on an Acclaim PepMap 100 C18 trap cartridge (0.3 mm inner diameter (i.d.), 5 mm length; Thermo Fisher Scientific) with the U3000 loading pump at 2 µL/min via the autosampler.

A 75 µm i.d. analytical microcapillary column was packed in-house with 250 mm of 1.9 µm ReproSil-Pur C18-AQ resin (Dr. Masch). AgileSLEEVE (Analytical Sales & Products) was used to maintain column temperature at 50 °C. The organic solvent solutions were water:acetonitrile:formic acid at 95:5:0.1 (volume ratio) for buffer A (pH 2.6) and 20:80:0.1 (volume ratio) for buffer B. The chromatography gradient was a 30-min column equilibration step in 2% B; a 5-min ramp to reach 10% B; 90 min from 10 to 40% B; 10 min to reach 90% B; a 5-min wash at 90% B; 0.1 min to 2% B; followed by a 15-min column re-equilibration step in 2% B. The nano pump flow rate was set to 0.180 µl/min.

The Orbitrap Eclipse was set up to run the TMT-SPS-MS3 method with three FAIMS compensation voltages (CV) at −40, −55, and −70 V and a cycle time of 1 s. Briefly, peptides were scanned from 400 to 1600 m/z in the Orbitrap at 120,000 resolving power before MS2 fragmentation by CID at 35% NCE and detection in the ion trap set to turbo detection. Dynamic exclusion was enabled for 45 s. The supernatant from phosphorylation peptide enrichment was analyzed with real-time search (RTS) enabled. During LC/MS data acquisition, ddMS2 spectra was searched against a non-redundant Danio rerio sequence database downloaded from NCBI 2021-03 and complemented with common contaminants. Carbamidomethyl (max = 57.0215 Da at C sites) and TMTpro 16plex (max = 304.2071 Da at Kn sites) were searched statically, while methionine oxidation (15.9949 Da) was searched as a variable modification. Synchronous precursor scanning (SPS) selected the top ten MS2 peptides for TMT reporter ion detection in the Orbitrap using HCD fragmentation at 65% NCE at 50,000 resolving power.

## MS/MS data processing

The LC/MSn dataset was processed using Proteome Discoverer 2.4 (Thermo Fisher Scientific). MS/MS spectra were searched against a zebrafish protein database (NCBI 2021-03) complemented with common contaminants. SEQUEST-HT implemented through Proteome Discoverer was set up as: precursor ion mass tolerance 10 ppm, fragment mass tolerance 0.3 Da, up to two missed cleavage sites, static modification of cysteine (+57.021 Da), and lysine and peptide N-termini with TMT tag (+304.2071 Da) and dynamic oxidation of methionine (+15.995 Da), and phosphorylation of serine, threonine, tyrosine (+79.9663 Da). Results were filtered to a 1% FDR at peptide level using Percolator through Proteome

## Phospho-proteomic and proteomic differential analysis

Normalized TMT (Tandem Mass Tag) Intensity readings were used for differential analysis by limma (version 3.54.2). Protein accession IDs were mapped to Ensembl Genes with biomaRt (v. 2.54.1). Differential phospho-peptides (phospho-proteome) and peptides (proteome) were selected, setting a *p* value <0.05 and a fold change >2.0, applying the empirical Bayes moderated *t*-test method. Localization of differential phospho-peptides upon *bckdk* mRNA knockdown was determined by Uniprot (Bateman et al, 2022).

Mass spectrometry data were normalized by NormalyzerDE (version 3.18). Fold change for each protein was calculated using limma (version 3.54.2), and resulting *p* values were adjusted with Benjamin–Hochberg method. Differential proteins were selected setting a *p* value <0.05 and a fold change >2.0 after filtering proteins present in all control samples.

## Co-immunoprecipitation

About 200 embryos were co-injected with 200 pg of *bckdk* mRNA and 50 pg of either Phf10-HA or GFP-HA mRNA. At 6 hpf, embryos were collected and lysed in 300 µL of RIPA buffer (50 mM Tris-HCl, pH 7.5, 150 mM NaCl, 1% NP-40, 0.1% SDS, 5 mM EDTA, and 0.5% sodium deoxycholate) by brief sonication (5 s on, 3 s off, for three cycles at 10% amplitude). Samples were rotated for 30 min at 4 °C and then centrifuged for 10 min at 4 °C. 1 µL of the supernatant was mixed with 9 µL of RIPA buffer and combined with 10 µL of SDS-PAGE buffer (160 mM Tris-HCl pH 8.0, 20% glycerol, 2% SDS, 0.1% bromophenol blue, and 200 mM DTT) to generate the input sample for western blot analysis. The remaining supernatant was used for co-immunoprecipitation, following the manufacturer's instructions for the Pierce Co-Immunoprecipitation Kit (Thermo Fisher Scientific, #26149). 5 µg of HA antibody (Roche, #11867423001) were used for antibody immobilization, and samples were incubated overnight at 4 °C with gentle mixing. Two elution fractions were collected using 40 µL of elution buffer each and mixed with 10 µL of 5X Lane Marker Sample Buffer provided in the kit. Input and both elution fractions were analyzed by Western blot using an anti-Bckdk antibody (A16104, Antibodies, diluted 1:1000).

## Immunofluorescence

Dechorionated embryos at 4 hpf were fixed overnight with 4% PFA at 4 °C in agitation. Then, they were washed twice with PBS at room

temperature (RT) and transferred to a 24-well plate and incubated overnight at 4 °C with shaking in 400 μL of primary antibody solution: DAPI 1:1000 (D9542-1g, Sigma-Aldrich), Phalloidin 1:100 (#A12379, Invitrogen), DMSO 5% in PBST (0.8% Tween 20, #). Afterwards, embryos were rinsed twice for 10 min with PBST at RT.

Image acquisition was performed in confocal microscopy Zeiss LSM 880 Airscan with a 40x oil objective and coupled to an Andor EMCCD iXOn DU 879 camera. Fluorescence images were then processed with Fiji (ImageJ). For analysing the number of total cells and mitotic cells, two different Z-stacks were selected from each embryo, using at least four embryos per independent experiment.

## Statistical analysis

No statistical methods were used to predetermine sample size. The experiments were not randomized, and investigators were not blinded to allocation during experiments and outcome assessment. The number of embryos, replicates, and experiments are indicated in figure legends.

For phenotypes derived from embryo microinjections, Chi-square statistical analyses were undertook using rstatix (v. 0.7.2) in R package, comparing delayed (30 and 50% epiboly-stages) with non-delayed embryos (germ ring and shield stages).

An unpaired two-tailed Mann–Whitney test was used to compare leucine-isoleucine or valine levels in control vs Bckdk KD embryos using Prism (GraphPad Software, La Jolla, CA, USA).

For western blot quantification analysis $p$ value was calculated using one-way Welch's $t$-test with Prism (GraphPad Software, La Jolla, CA, USA). Quantitative RT-qPCR, RIN analysis and GFP intensity $p$ values were calculated using an unpaired $t$-test or one-way ANOVA test with Prism (GraphPad Software, La Jolla, CA, USA) and the standard error of the mean (SEM) was used to show error bars. Grubbs's test was performed first for outlier identification. Unpaired two-tailed Mann–Whitney test with Prism (GraphPad Software, La Jolla, CA, USA) was also employed for calculating $p$ value for the distribution of differential expression genes associated with differential accessibility regions.

$p$ value and distance (D, maximal vertical distance between the compared distribution) from the comparison of cumulative distribution of RNA levels at 6 hpf were calculated using Kolmogorov–Smirnov Tests by dgof (v1.4) in R package.

## Data availability

RNA-seq, ATAC-seq, CUT&RUN, and SLAM-seq data have been deposited in GEO under accession code GSE268294 (https://www.ncbi.nlm.nih.gov/geo/query/acc.cgi?acc=GSE268294). Proteomics, phospho-proteomics, and isoleucine-leucine data have been deposited at MassIVE MSV000094733 Valine quantification data have been deposited at MassIVE MSV000094867.

The source data of this paper are collected in the following database record: biostudies:S-SCDT-10_1038-S44318-025-00617-8.

## Peer review information

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

## Acknowledgements

We thank all members of the Moreno-Mateos and the Bazzini laboratories for their intellectual and technical support. This work was supported by Ramon y Cajal (RyC-2017-23041), PGC2018-097260-B-I00, PID2021-127535NB-I00, MDM-2016-0687, and CEX2020-001088-M grants funded by MICIU/AEI/ 10.13039/501100011033 by "ERDF A way of making Europe", "ERDF/EU" and by ESF Investing in your future from Ministerio de Ciencia, Innovación y Universidades and European Union (MAM-M). MAM-M was the recipient of the Genome Engineer Innovation 2019 Grant from Synthego. The Moreno-Mateos lab has been also co-financed by the Spanish Ministry of Science and Innovation with funds from the European Union NextGenerationEU (PRTR-C17.I1) and the Regional Ministry of University, Research and Innovation of the Autonomous Community of Andalusia within the framework of the Biotechnology Plan applied to Health, grant CNS2022-135564 funded by MICIU/AEI/10.13039/501100011033 and by the European Union NextGenerationEU/PRTR, the European Regional Development Fund (FEDER 80% of the total funding) by the Ministry of Economy, Knowledge, Business and University, of the Government of Andalusia, within the framework of the FEDER Andalusia 2014-2020 operational program within the objective "Promotion and generation of frontier knowledge and knowledge oriented to the challenges of society, development of emerging technologies (grant UPO-1380590)" and by the Fondo Europeo de Desarrollo Regional (FEDER) and Consejería de Transformación Económica, Industria, Conocimiento y Universidades de la Junta de Andalucía, within the operative program FEDER Andalucía 2014-2020 (01—Refuerzo de la investigación, el desarrollo

tecnológico y la innovación, grant P20_00866). The CABD is an institution funded by Pablo de Olavide University, Consejo Superior de Investigaciones Científicas (CSIC), and Junta de Andalucía. LH-H was a recipient of ayudas para contratos predoctorales para la formación de doctores contemplada en el Subprograma Estatal de Formación del Programa Estatal de Promoción del Talento y su Empleabilidad en I + D + i, en el marco del Plan Estatal de Investigación Científica y Técnica y de Innovación 2017-2020 (Ministerio de Ciencia e Innovación). JC-C is supported by the Junta de Andalucía Predoctoral Grant (PREDOC_01569), respectively. IM-S was a recipient of the Margarita Salas Postdoctoral contract funded by "NextGenerationEU", Plan de Recuperación, Transformación y Resilencia and Ministerio de Universidades (recualificación del sistema universitario español 2021-2023, Pablo de Olavide University Call). JMS-P was funded by an Emergia grant from Junta de Andalucía (EMC21_00188). This study was supported by the Stowers Institute for Medical Research. AAB was awarded a US National Institutes of Health grant (NIH-R01 GM136849 and NIH R21OD034161). This work was performed as part of thesis research for the GdSP, Graduate School of the Stowers Institute for Medical Research grants. We thank our colleagues Juan Ramon Martínez-Morales, Barbara Pernaute and Thomas Spruce (CABD, Seville, Spain), and Jose Carlos Reyes (CABIMER, Seville, Spain) for the critical reading of the manuscript. We also thanks Rhonda Egidy and Anoja Perera from Sequencing and Discovery Genomics facility at (Kansas, MO, USA), Danielson Baia Amaral (Stowers Institute) for technical support in performing injections, embryo collection, photography, and qPCR for Valine quantification experiments, applying Slamdunk pipeline and providing suggestions on the manuscript draft, Juan J. Tena, Thirsa Brethouwer and Cielo Centola (CABD, Seville, Spain) for H3K27ac antibody, and helping with CUT&RUN experiment and synopsis image, respectively.

## Author contributions

**Luis Hernández-Huertas**: Data curation; Software; Formal analysis; Validation; Investigation; Visualization; Methodology; Writing—review and editing. **Ismael Moreno-Sánchez**: Data curation; Software; Formal analysis; Validation; Investigation; Visualization; Methodology; Writing—review and editing. **Jesús Crespo-Cuadrado**: Data curation; Investigation; Methodology. **Ana Vargas-Baco**: Data curation; Investigation. **Gabriel da Silva Pescador**: Data curation; Software; Formal analysis; Investigation; Methodology; Writing—review and editing. **Ying Zhang**: Data curation; Software; Formal analysis; Investigation; Methodology; Writing—review and editing. **Zhihui Wen**: Data curation; Software; Formal analysis; Investigation; Methodology; Writing—review and editing. **Laurence Florens**: Data curation; Software; Formal analysis; Investigation; Methodology; Writing—review and editing. **José M Santos-Pereira**: Software; Formal analysis; Methodology; Writing—review and editing. **Ariel A Bazzini**: Conceptualization; Resources; Data curation; Formal analysis; Supervision; Funding acquisition; Investigation; Methodology; Writing—original draft; Project administration. **Miguel A Moreno-Mateos**: Conceptualization; Resources; Data curation; Formal analysis; Supervision; Funding acquisition; Investigation; Methodology; Writing—original draft; Project administration.

Source data underlying figure panels in this paper may have individual authorship assigned. Where available, figure panel/source data authorship is listed in the following database record: biostudies:S-SCDT-10_1038-S44318-025-00617-8.

## Disclosure and competing interests statement

The authors declare no competing interests.

# Expanded View Figures

**Figure EV1.  CRISPR-RfxCas13d maternal screening identifies candidates with a role along and after MZT in zebrafish.**

(**A**) RT-qPCR analysis showing levels of *nanog* mRNA at 4 hpf in the indicated conditions. Results are shown as the averages ± standard error of the mean from two experiments with two biological replicates per experiment (*n* = 10 embryos/biological replicate) for RfxCas13d protein alone (Cas13d) and RfxCas13d plus a mix of three gRNAs targeting *nanog* mRNA (Nanog KD). *taf15* mRNA was used as a normalization control. Exact *p* value is indicated above, an unpaired *t*-test. (**B**) Stacked barplots showing the percentage of wildtype (WT), developmentally altered or dead zebrafish embryos at 24 hpf after the depletion of *bckdk, cab39l, calm1a, calm2a, mknk1, mknk2a,* and *ppp4r2a* mRNAs. Embryos were previously divided at 6 hpf according to their developmental phenotype. No: embryos between germ ring and shield stage at 6 hpf. Yes: embryos between 30 and 50% epiboly at 6 hpf. The results are shown as the averages ± standard error of the mean of each developmental stage from two independent experiments. mRNAs encoding for kinases are labeled in blue and mRNAs encoding for phosphatases are labeled in green. The number of embryos evaluated (*n*) for each condition is shown. (**C**) Stacked barplots showing the percentage of wildtype and developmentally altered embryos from the CRISPR-RfxCas13d screening conditions that did not present more than 35% of embryos with epiboly defects at 6 hpf (Fig. 1C). The results are shown as the averages ± standard error of the mean of each developmental stage from at least two independent experiments. mRNAs encoding for kinases are labeled in blue and mRNAs encoding for phosphatases are labeled in green. The number of embryos evaluated (*n*) for each condition is shown. (**D**) Representative pictures of embryos injected with RfxCas13d and three gRNAs targeting *mibp* mRNA (gMIBP) compared to uninjected embryos (WT) evaluated at 30 hpf (scale bar, 0.5 mm). Class I: curved tail (mild phenotype). Class II: shorter tail and partial microcephaly (severe phenotype). Class III: notochord malformation and microcephaly (extremely severe phenotype). (**E**) Stacked barplots showing percentage of observed phenotypes at 30 hpf after the injection of RfxCas13d protein (Cas13d) (3 ng/embryo) alone or together with a mix of 3 gRNAs targeting *mibp* mRNA (gMIBP) (1000 pg/embryo). The results are shown as the averages ± standard error of the mean of each developmental stage from three independent experiments. The phenotype selection follows the criteria of panel (**D**). The number of embryos evaluated (*n*) for each condition is shown. (**F**) RT-qPCR analysis showing levels of *mibp* mRNA at 4 hpf in the indicated conditions. Results are shown as the averages ± standard error of the mean from two experiments with two biological replicates per experiment (*n* = 10 embryos/biological replicate) for RfxCas13d protein alone (Cas13d) and RfxCas13d plus a mix of 3 gRNAs targeting *mibp* mRNA (gMIBP). *taf15* mRNA was used as a normalization control. Exact *p* value is indicated above (unpaired *t*-test). (**G**) Intersection visualization analysis showing the number of downregulated genes shared between the mRNA knockdowns. Intersections with more than 50 shared genes are shown. Source data are available online for this figure.

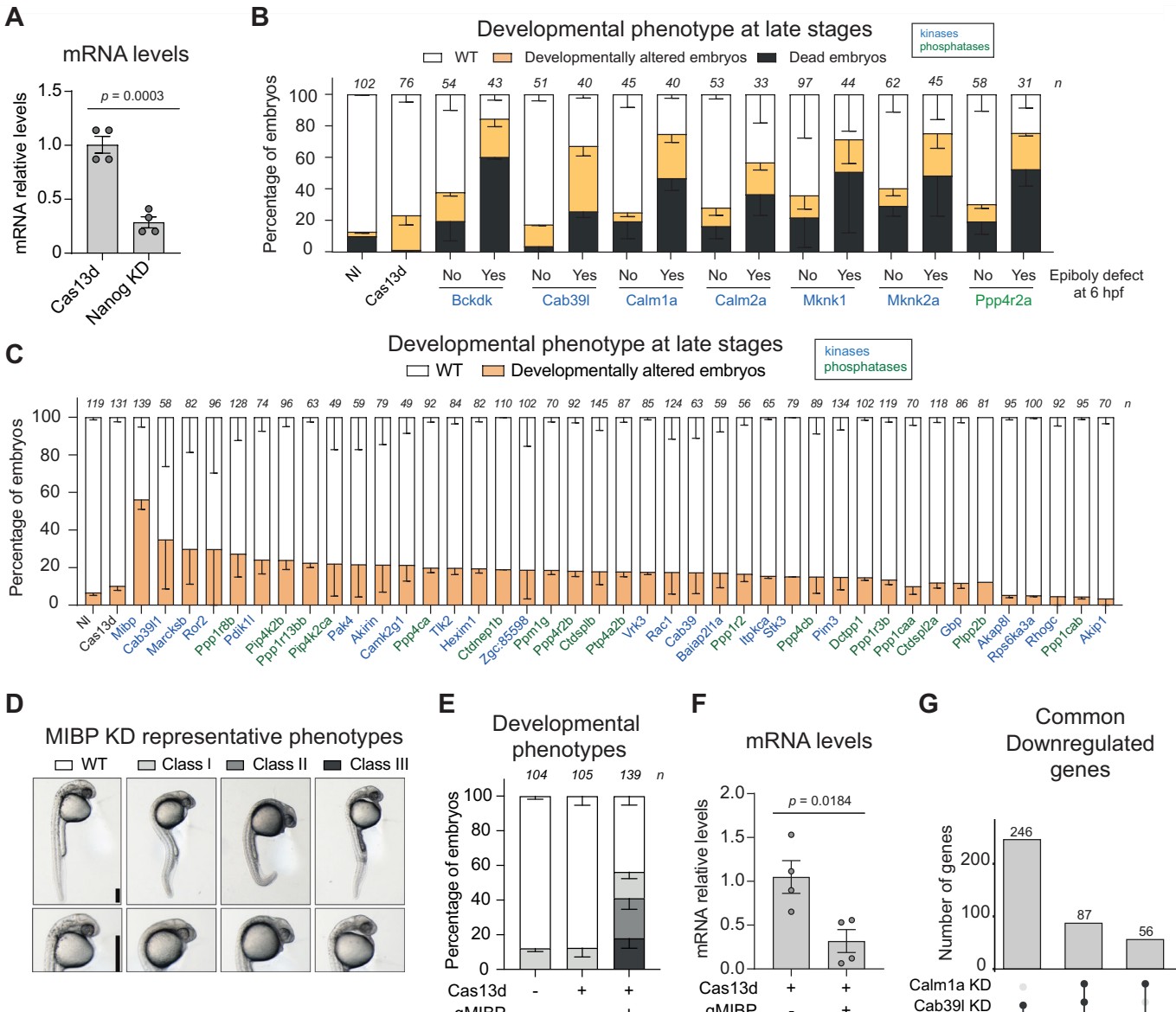

**A** mRNA levels

**B** Developmental phenotype at late stages

**C** Developmental phenotype at late stages

**D** MIBP KD representative phenotypes

**E** Developmental phenotypes

**F** mRNA levels

**G** Common Downregulated genes

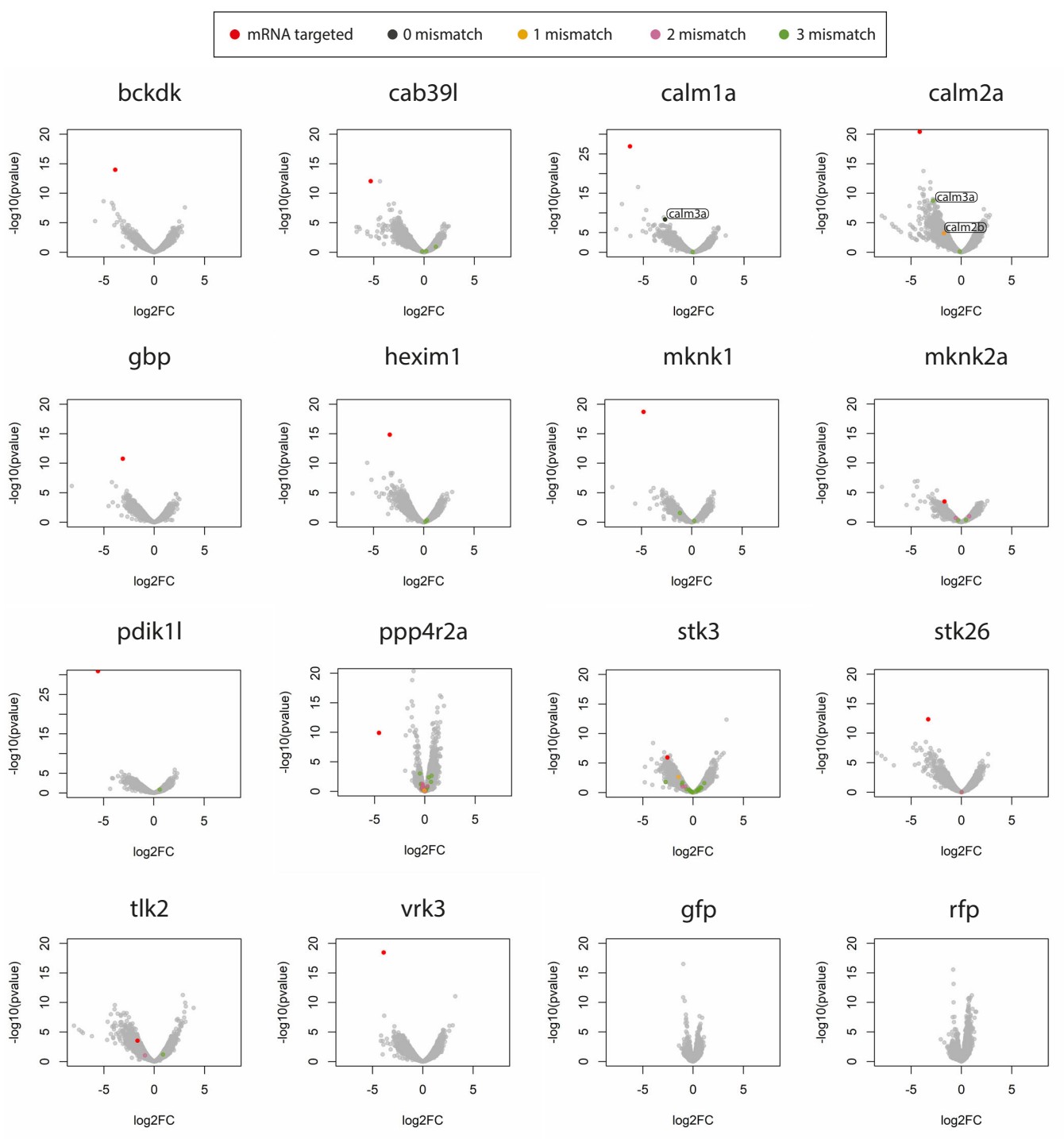

**Figure EV2. CRISPR-RfxCas13d triggers efficient and specific depletion of targeted mRNAs with minimal off-target effects.**

Scatter plots showing the fold change in mRNA levels and *p* values from a minimum of two biological RNA-seq replicates at 4 hpf. *p* values were calculated using the Wald test. Data are shown for seven candidates that caused epiboly defects (positive candidates) in at least 35% of injected embryos, seven candidates that did not meet this developmental phenotype threshold (negative candidates), and two non-targeting control conditions (gRNAs designed to target *gfp* and *rfp* mRNA). The mRNA targeted in each condition is highlighted in red. mRNAs that could potentially be recognized by the gRNAs used in each condition (allowing up to three mismatches) are also indicated: zero mismatches in black, one mismatch in yellow, two mismatches in pink, and three mismatches in green.

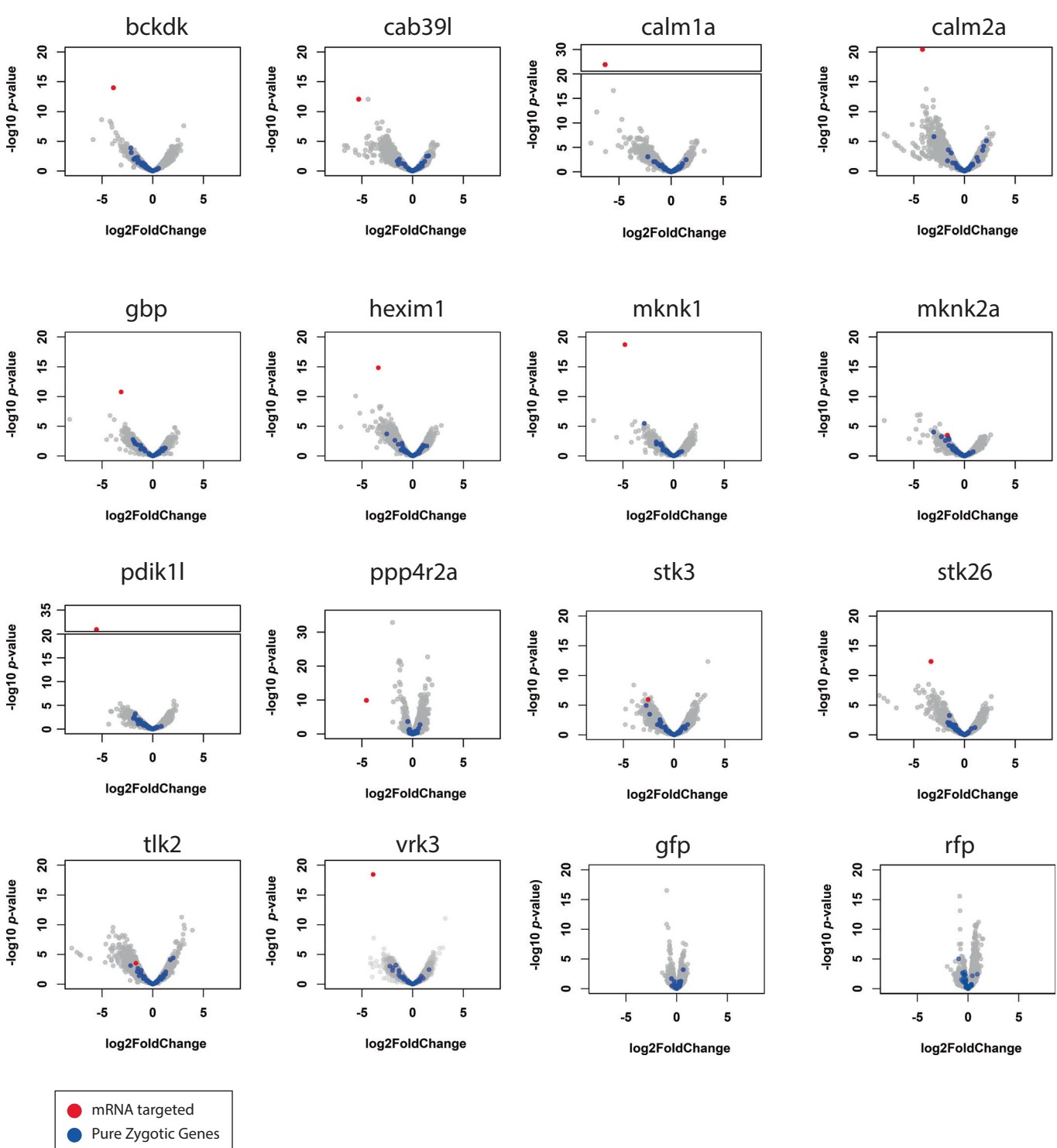

**Figure EV3. Transcriptome analysis shows a global downregulation of PZG upon *bckdk* and *mknk2a* mRNA depletion.**

Scatter plots representing the fold change in mRNA levels and *p* value from a minimum of two biological RNA-seq replicates at 4 hpf of seven candidates with epiboly defects (positive candidate) in at least 35% of injected embryos and seven candidates that did not pass this developmental phenotype filter (negative candidate) and two non-targeting control conditions (gRNAs designed to target *gfp* and *rfp* mRNA). *p* values were calculated using the Wald test. Pure zygotic genes mRNAs defined by Lee et al, 2013 are indicated in blue. The mRNA targeted in each condition is highlighted in red.

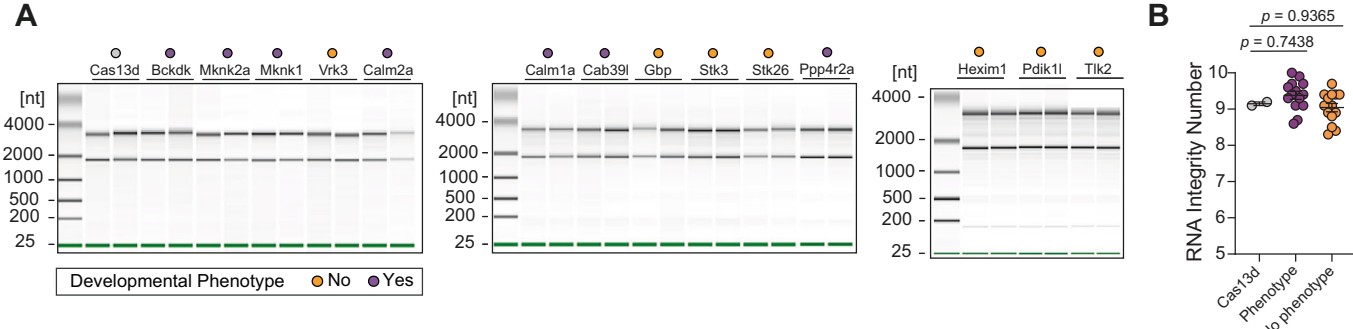

**Figure EV4. RNA-targeting mediated by CRISPR-RfxCas13d RNP complexes does not show collateral activity in zebrafish embryos.**

RNA integrity analysis (RIN) from samples used for RNA-Seq in Figs. EV2 and EV3 analysed by Agilent Bioanalyzer 2100. Electrophoresis gel (A) and RIN-associated (B). Developmental phenotype associated to each KD condition is indicated as purple dots (positive candidate) or orange dots (negative candidate). The RfxCas13d control condition is indicated as gray dots. One-way ANOVA comparing RIN from all the samples between them is shown in (B). Exact p values are indicated above. Source data are available online for this figure.

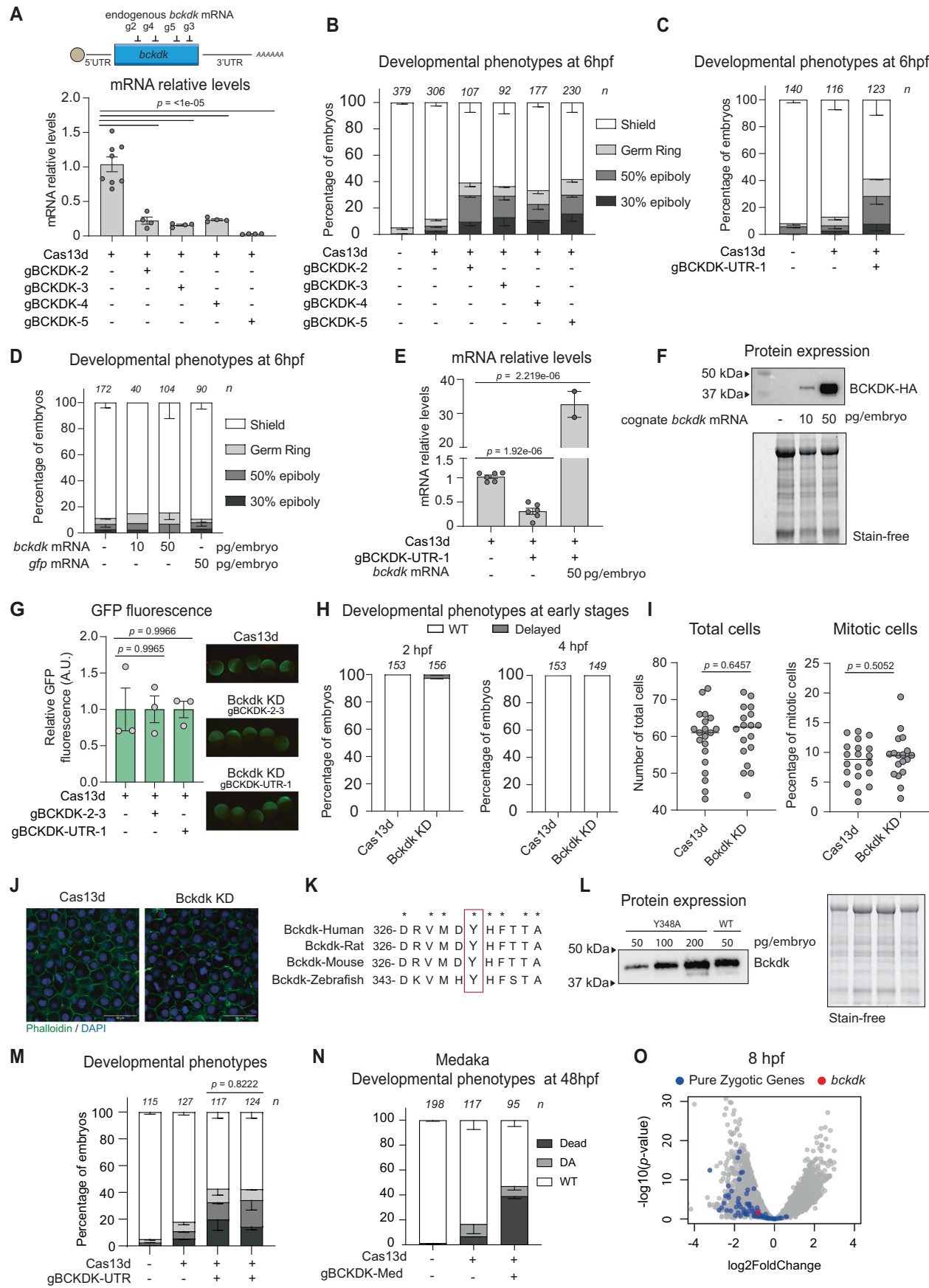

◀ **Figure EV5. The developmental phenotype upon *bckdk* mRNA depletion can be genetically rescued and is recapitulated using independent gRNAs and in different teleost models.**

(A) Schematic representation of endogenous *bckdk* mRNA and individual gRNAs employed (Top). RT-qPCR analysis showing levels of *bckdk* mRNA at 4 hpf in zebrafish embryos co-injected using individual gRNAs targeting *bckdk* mRNA and RfxCas13d protein. Results are shown as the averages ± standard error of the mean from two independent experiments with four biological replicates each (*n* = 10 embryos/biological replicate). *taf15* mRNA was used as a normalization control. Exact *p* values are indicated above (one-way ANOVA) (Bottom). (B) Stacked barplots showing the percentage of zebrafish embryos in different developmental stages quantified at 6 hpf using individual gRNAs (1000 pg/embryo) targeting *bckdk* mRNA co-injected with RfxCas13d protein (3 ng/embryo). The phenotype selection criteria were the same as described in Fig. 1C. The results are shown as the averages ± standard error of the mean of each developmental stage from at least two independent experiments. The number of embryos evaluated (*n*) is shown for each condition. (C) Stacked barplots showing developmental phenotypes at 6 hpf upon depletion of *bckdk* mRNA using a gRNA targeting the 3′UTR (gBCKDK-UTR-1). The phenotype selection criteria were the same as described in Fig. 1C. The results are shown as the averages ± standard error of the mean of each developmental stage from two independent experiments. The number of embryos evaluated (*n*) is shown for each condition. (D) Stacked barplots showing developmental phenotypes at 6 hpf in zebrafish embryos injected with the cognate *bckdk* mRNA at 10 or 50 pg per embryo or with an mRNA encoding for GFP at 50 pg per embryo. The phenotype selection criteria were the same as described in Fig. 1C. The results are shown as the averages ± standard error of the mean of each developmental stage from two independent experiments. The number of embryos evaluated (*n*) is shown for each condition. (E) RT-qPCR analysis showing levels of *bckdk* mRNA at 4 hpf in zebrafish embryos in the rescue experiment (Fig. 2B). Results are shown as the averages ± standard error of the mean from one to three independent experiments with at least two biological replicates each (*n* = 10 embryos/ biological replicate). *taf15* mRNA was used as a normalization control. Exact *p* values are indicated above (unpaired *t*-test). (F) Western blot showing Bckdk-HA protein expression at 6 hpf in zebrafish embryos injected with 10 or 50 pg/embryo of the cognate *bckdk-HA* mRNA (Top panel). Bottom panel shows stain-free signal (Gürtler et al, 2013) of the gel as loading control. (G) Barplots showing GFP fluorescence signal in zebrafish embryos injected with RfxCas13d alone (Cas13d) or with 2 gRNAs targeting *bckdk* mRNA in the coding sequence (Bckdk KD; gBCKDK-2-3) or one in the 3′UTR (Bckdk KD; gBCKDK-UTR-1) together with 50 pg of *gfp* mRNA. GFP signal is quantified from three biological replicates of five embryos each. Exact *p* values are indicated above, unpaired *t*-test. Representative fluorescence microscopy images used for the quantification are shown. (H) Stacked barplots representing the percentage of embryos normally developed (WT) or delayed at 2 or 4 hpf in zebrafish embryos injected with RfxCas13d protein alone (Cas13d) (3 ng/embryo) or with a mix of three gRNAs (1000 pg/embryo) targeting *bckdk* mRNA (Bckdk KD). The results are shown as the averages ± standard error of the mean of each developmental stage from two independent experiments. Number of embryos evaluated (*n*) is shown for each condition. (I) Number of total cells or percentage of mitotic cells in zebrafish embryos at 4 hpf injected with RfxCas13d protein alone (Cas13d) or with a mix of three gRNAs targeting *bckdk* mRNA (Bckdk KD). The black line represents the median from two independent experiments with at least four embryos each. Exact *p* values are indicated above (unpaired *t*-test). (J) Representative immunofluorescence used for quantification in panel (I) showing zebrafish embryos injected with RfxCas13d alone (Cas13d) or with gRNAs targeting *bckdk* mRNA (Bckdk KD), fixed at 4 hpf with PFA 4% and incubated with Phalloidin and DAPI (see Methods for details; Scale bar, 100 μm). (K) Schematic representation of the Bckdk protein alignment among human, mouse, rat, and zebrafish. The red square highlights a conserved tyrosine residue, whose substitution with alanine has been reported to reduce kinase activity by 95% (Wynn et al, 2000; Singh et al, 2024). Asterisks (*) indicate amino acids conserved across all analyzed species. (L) Western blot showing Bckdk-HA protein expression at 6 hpf in zebrafish embryos injected with 50, 100, or 200 pg/embryo of the kinase-dead *bckdk* mRNA variant, or with 50 pg of the wild-type bckdk mRNA (left panel). The right panel shows the Stain-Free gel signal used as a loading control (Gürtler et al, 2013). (M) Stacked barplots showing the percentage of observed phenotypes in zebrafish embryos injected with RfxCas13d protein (3 ng/embryo) together with one gRNA targeting the 3′UTR of the endogenous *bckdk* mRNA (gBCKDK-UTR) (1000 pg/ embryo), and co-injected with a dead-kinase *bckdk* mRNA version (200 pg/embryo). The results are shown as the averages ± standard error of the mean of each developmental stage from two independent experiments. The number of embryos evaluated (*n*) is shown for each condition. Exact *p* value is indicated above ($\chi$2-test). (N) Stacked barplots representing the percentage at 48 hpf of wildtype (WT), developmentally altered (DA) or dead medaka embryos injected with RfxCas13d protein (6 ng/ embryo) alone or together with a gRNA (gBCKDK-Med) (2000 pg/embryo) targeting *bckdk* mRNA. The results are shown as the averages ± standard error of the mean of each developmental stage from three independent experiments. The number of embryos evaluated (*n*) is shown for each condition. (O) Scatter plots representing the log2 fold change in mRNA level and the associated *p* value from three biological RNA-seq replicates (*n* = 10 embryos/biological replicate) at 8 hpf in medaka embryos from the comparison between embryos injected only with RfxCas13d protein or together with a gRNA targeting *bckdk* mRNA. *bckdk* mRNA is represented in red. Pure zygotic genes mRNAs (PZG) from medaka embryos determined by Li et al, 2020 data were depicted in blue. Source data are available online for this figure.

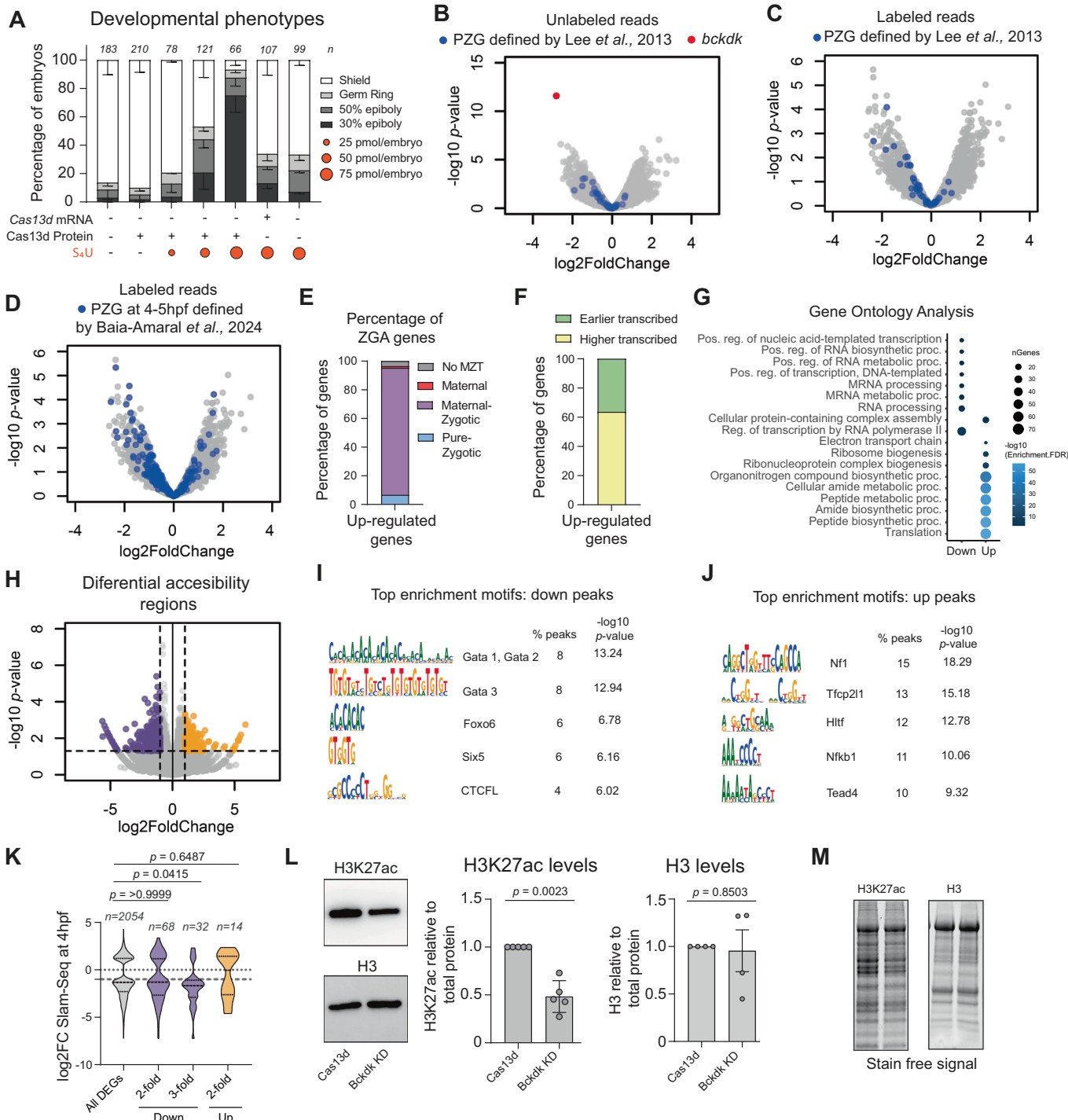

**Figure EV6.   SLAM-seq and ATAC analysis in *bckdk* mRNA KD.**

(**A**) Stacked barplots showing developmental phenotypes at 6 hpf in zebrafish embryos injected with RfxCas13d protein (3 ng/embryo) or *RfxCas13d* mRNA (150 pg/embryo) alone or together with different amounts of $S_4U$ (25, 50, or 75 mM). The phenotype selection criteria were the same as described in Fig. 1C. The results are shown as the averages ± standard error of the mean of each developmental stage from at least two independent experiments. The number of embryos evaluated (*n*) is shown for each condition. (**B**) Volcano plot representing the log2 fold change in mRNA level from unlabeled reads (SLAM-Seq data) and the associated *p* value from two and four biological replicates (*n* = 25 embryos/biological replicate) at 4 hpf from embryos injected with RfxCas13d protein alone or with a mix of two gRNAs targeting *bckdk* mRNA, respectively. *p* values were calculated using the Wald test. *bckdk* mRNA is represented in red. Pure zygotic genes mRNAs (PZG) determined by Lee et al, 2013 data were depicted in blue. Volcano plots representing the fold change in mRNA level from labeled reads and the associated *p* value from three biological SLAM-Seq replicates of zebrafish embryos at 4 hpf from two and four biological replicates (*n* = 25 embryos/biological replicate) from embryos injected with RfxCas13d protein alone or with a mix of two gRNAs targeting *bckdk* mRNA, respectively. *p* values were calculated using the Wald test. *bckdk* mRNA is represented in red. Pure zygotic genes mRNAs (PZG) determined by Lee et al, 2013 data were depicted in blue in panel (**C**) and an updated list of pure zygotic genes mRNAs (PZG) determined by Baia Amaral et al, 2024 data were depicted in blue in panel (**D**). (**E**) Stacked barplot showing percentage of upregulated genes in SLAM-Seq data upon *bckdk* mRNA depletion that belong to different categories. Genes were classified according to Baia Amaral et al, 2024. No MZT: Genes that are not present between 0 and 7 hpf; Maternal: Genes maternally provided as mRNA but not zygotically transcribed; Maternal-and-Zygotic: Genes maternally provided and zygotically transcribe between 4 and 7 hpf; or pure zygotic: genes not maternally provided as mRNA and zygotically transcribed between 4 and 7 hpf. (**F**) Stacked barplot showing percentage of upregulated MZT genes (maternal-and-zygotic or pure zygotic genes) in SLAM-Seq data upon *bckdk* mRNA depletion that belong to different categories. Genes were classified according to transcript levels in wild-type conditions at 4 hpf in labeled data. Higher transcribed: Genes with more than ten CPM (counts per million) in labeled data at 4 hpf in WT conditions; Earlier transcribed: Genes with less than ten CPM in labeled data at 4 hpf in WT conditions. (**G**) Gene Ontology enrichment analyses of biological processes for down-(Down) or up-regulated genes (Up) from the comparison of SLAM-Seq data between zebrafish embryos injected with RfxCas13d alone and together with two gRNAs targeting *bckdk* mRNA. Terms with a false discovery rate (FDR) lower than 0.05 and with more than 20 genes represented are shown and considered as enriched. (**H**) Differential analyses of chromatin accessibility between zebrafish embryos injected with RfxCas13d protein alone or co-injected with two gRNAs targeting *bckdk* mRNA from ATAC-seq data at 4 hpf from two biological replicates per condition (*n* = 80 embryos/biological replicate). The log2 fold change (FC) and *p* value associated is represented. *p* values were calculated using the Wald test. Regions with a significant decrease in accessibility are represented as purple dots (log2FC <−1 and *p* value <0.05), and regions with a significant increase in accessibility are represented as orange dots (log2FC >1 and *p* value <0.05). Vertical and horizontal dashed lines indicate 1.5-fold and *p* value = 0.05, respectively. Motif enrichment analyses from the decreased (**I**) and increased (**J**) ATAC regions (down and up peaks, respectively) in *bckdk* mRNA knockdown condition. The top five motifs are represented with their motif logos, transcription factor name, percentage of peaks containing the motif and enrichment *p* value. (**K**) Violin plots showing the distribution of log2 fold change of RNA levels (SLAM-Seq) from all differential expression genes (DEGs) and those associated with less accessible (Down) or more accessible (Up) regions from ATAC-Seq data. Dash lines and dot lines inside the violin plots indicate the mean and quartiles, respectively. Gray dot line and dash line outside the violin plots indicate 0-fold and 1.5-fold in RNA levels, respectively. Exact *p* values are indicated above, Mann–Whitney *U*-test. The number of differentially expressed genes (*n*) for each category are shown. (**L**) Representative western blot images for H3K27ac and H3 of embryos injected with RfxCas13d protein alone (Cas13d) or together with two gRNAs targeting bckdk mRNA (Bckdk KD) (left). Barplots represent H3K27ac or H3 levels relative to total proteins as the averages ± standard error of the mean at least four biological replicates from two or three independent experiments. Zebrafish embryos were collected at 4 hpf (*n* = 25 embryos/biological replicate). Exact *p* values are indicated above (Welch's *t*-test) (Right). (**M**) Stain-free signal (Gürtler et al, 2013) of the gels employed as loading control for Western blot in panel (**L**). Source data are available online for this figure.

**A**

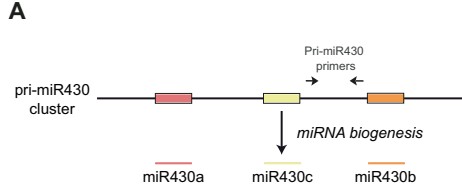

**B**

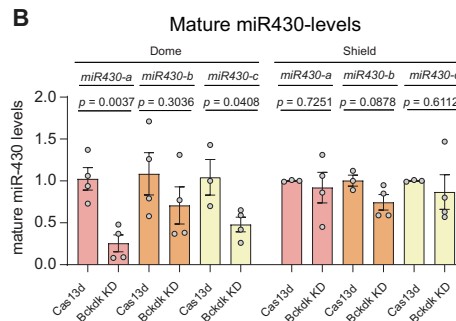

**Figure EV7. *bckdk* mRNA depletion affects the processing of miR-430.**

(A) Schematic representation of miR-430 triplet, RT-qPCR primer employed for measuring of primary miR-430 levels are indicated with black arrows (Adapted from Hadzhiev et al, 2023). (B) RT-qPCR analysis showing levels of mature miR-430 isoforms (miR430-a, red; miR430-b, orange; and miR430c, yellow) at 4.3 hpf (Dome) and 6 hpf (Shield). Results are shown as the averages ± standard error of the mean from three independent experiments with two biological replicates each ($n = 10$ embryos/ biological replicate) for RfxCas13d protein alone and RfxCas13d plus 2 gRNAs targeting *bckdk* mRNA (see Methods for details). ncRNA *u4atac* was used as a normalization control. Exact *p* values are indicated above (unpaired *t*-test). Same data as represented in Fig. 4C.

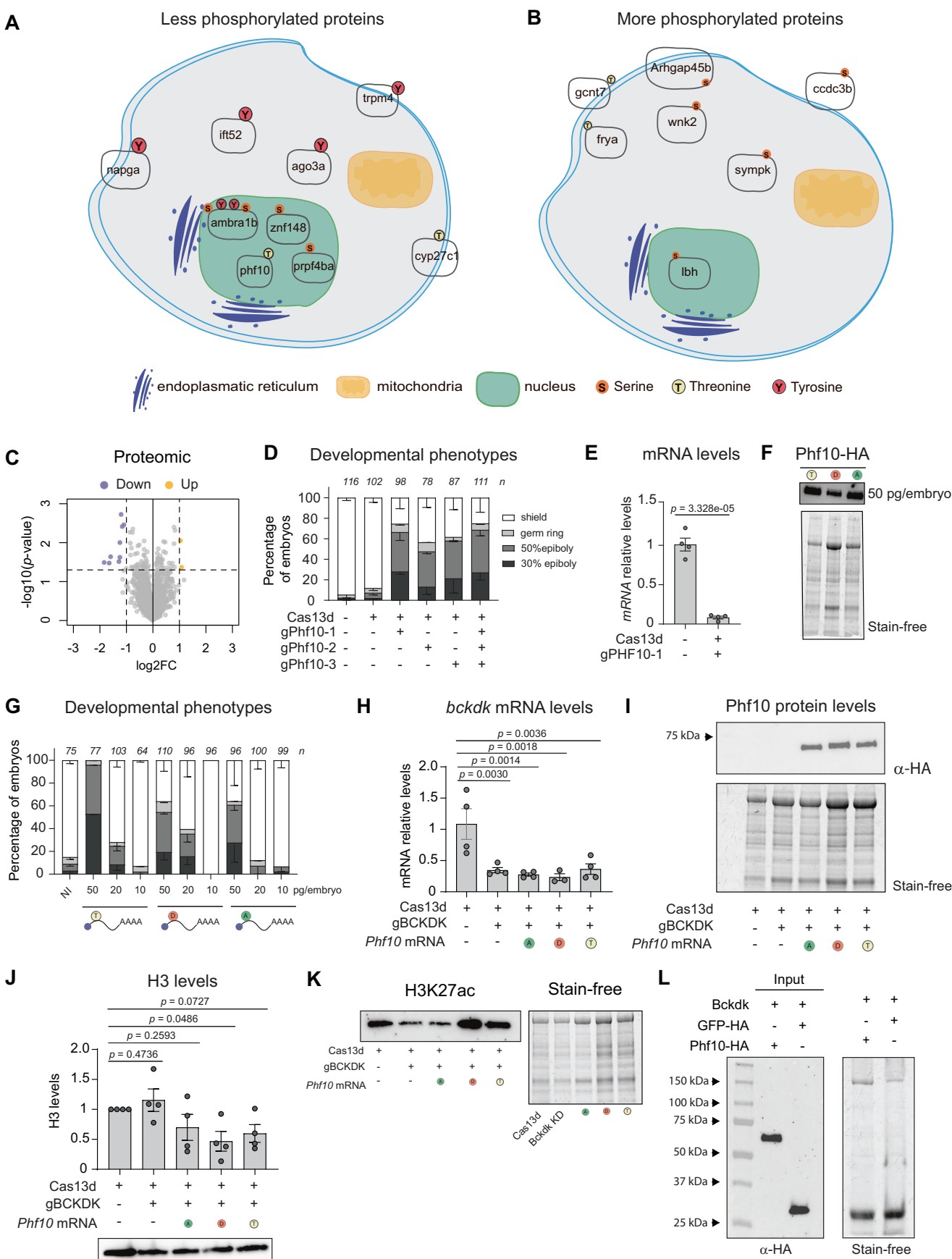

◀ **Figure EV8. Phospho-proteomic analysis upon *bckdk* mRNA depletion identifies Phf10 as a potential Bckdk target controlling MZT.**

Schematic representation of the proteins found less (**A**) or more (**B**) phosphorylated upon *bckdk* mRNA knockdown condition. Its known cell localization (Bateman et al, 2022) and the target residues are represented. (**C**) Scatter plot representing the log2 fold change in protein level and the associated *p* value from four biological replicates ($n = 100$ embryos/biological replicate) at 4 hpf from embryos injected with RfxCas13d protein alone or with a mix of two gRNAs targeting *bckdk* mRNA, respectively. *p* values were calculated using a moderated *t*-test. Less (Down) or more (Up) abundant proteins are indicated in purple and yellow, respectively. Dashed lines indicated 2-fold in proteins levels and *p* value = 0.05. (**D**) Stacked barplots showing the percentage of phenotypes observed at 6 hpf from embryos injected with RfxCas13d (3 ng/embryo) alone or together with indicated gRNAs (1000 pg/embryo) targeting *phf10* mRNA. The results are shown as the averages ± standard error of the mean of each developmental stage from at least two independent experiments. The phenotype selection criteria were the same as in Fig. 1C. Number of embryos evaluated (n) is shown for each condition. (**E**) RT-qPCR analysis showing levels of *phf10* mRNA at 2 hpf in zebrafish embryos co-injected using a gRNA (gPHF10-1) targeting *phf10* mRNA and RfxCas13d protein. Results are shown as the averages ± standard error of the mean from two independent experiments with two biological replicates each ($n = 10$ embryos/biological replicate). *taf15* mRNA was used as normalization control. Exact *p* value is indicated above, unpaired *t*-test. (**F**) Western blot showing Phf10-HA expression at 6 hpf in zebrafish embryos injected with 50 pg/embryo of the *phf10-HA* mRNA versions used in Fig. 5. Stain-free signal (Gürtler et al, 2013) of the gel as loading control. (**G**) Stacked barplots showing the percentage of phenotypes observed at 6 hpf from embryos injected with 10, 20, or 50 pg/embryo of *phf10* mRNA WT (Phf10-16T) or modified in the residue phosphorylated by BCKDK. Phf10-16D (Aspartic acid, mimic the phosphorylation state mediated by Bckdk), Phf10-16A (Alanine, mimic a constitutively non-phosphorylatable version). The number of embryos evaluated (*n*) for each condition is shown. The results are shown as the averages ± standard error of the mean of each developmental stage from two independent experiments. The phenotype selection criteria were the same as in Fig. 1C. Number of embryos evaluated (*n*) is shown for each condition. (**H**) RT-qPCR analysis showing *bckdk* mRNA levels at 4 hpf in zebrafish embryos co-injected with RfxCas13d protein and two gRNAs targeting *bckdk* mRNA, either alone or together with different versions of *phf10* mRNA used in Fig. 5. T: threonine (wild-type version); D: aspartic acid (mimics the phosphorylated state mediated by Bckdk); A: alanine (mimics a constitutively non-phosphorylatable version). Results are presented as the mean ± standard error of the mean from two independent experiments, each with two biological replicates ($n = 10$ embryos per replicate). *taf15* mRNA was used as the normalization control. Exact *p* values are indicated above, one-way ANOVA). (**I**) Western blot showing Phf10-HA expression at 4 hpf in the rescue conditions indicated in panel (**H**) (Top). Stain-free signal (Gürtler et al, 2013) of the gel as loading control (Bottom). A representative western blot from two independent experiments of HA signal is shown under the indicated conditions. (**J**) Barplots representing H3 levels relative to total proteins as the averages ± standard error of the mean from four independent experiments under the rescue condition experiment similar to indicated in panel (**H**) (Top). Zebrafish embryos were collected at 4 hpf ($n = 15-20$ embryos/biological replicate) (ns non-significant. Exact *p* values are indicated above, Welch's *t*-test). Western blot showing H3 protein levels (Bottom). (**K**) Representative western blot image for H3K27ac of embryos co-injected with RfxCas13d protein and two gRNAs targeting *bckdk* mRNA, either alone or together with different versions of *phf10* mRNA described in panel (**H**). Zebrafish embryos were collected at 4 hpf ($n = 25$ embryos/biological replicate) (Left). Stain-free signal (Gürtler et al, 2013) of the gel as loading control (Right). (**L**) Western blot showing Phf10-HA and GFP-HA expression employed as input for the co-immunoprecipitation assay (Fig. 5H). Phf10-HA (50 pg/embryo) or GFP-HA (50 pg/embryo) were co-overexpressed with *bckdk* mRNA (200 pg/embryo). Stain-Free signal (Gürtler et al, 2013) of the gel as loading control (Right). Source data are available online for this figure.

