## [Peer Review File · The EMBO Journal]

CRISPR-RfxCas13d screening uncovers Bckdk as a post-translational regulator of maternal-to-zygotic transition in teleosts

Luis Hernandez-Huertas, Ismael Moreno-Sánchez, Jesús Crespo-Cuadrado, Ana Vargas-Baco, Gabriel da Silva Pescador, Ying Zhang, Zihui Wen, Laurence Florens, José Santos-Pereira, Ariel Bazzini, and Miguel Angel Moreno-Mateos

Corresponding author(s): Miguel Angel Moreno-Mateos (mamormat@upo.es) , Ariel Bazzini (arb@stowers.org)

Review Timeline:

Submission Date:	30th Jun 25
Editorial Decision:	12th Aug 25
Revision Received:	28th Aug 25
Accepted:	1st Oct 25

Editor: Cornelius Schneider

Transaction Report:

This manuscript was transferred to The EMBO JOURNAL following peer review at another journal.

Reviewers' Comments:

Reviewer #1:

Remarks to the Author:

Authors of Hernandez-Huertas et al. present their manuscript describing CAS13 mediated knockdown of the protein kinase Bckdk in early zebrafish embryos. They provide experimental evidence that Bckdk has a potential role in maternal zygotic transition, demonstrating a developmental delay, reduction in H3K27ac, impaired function of mir-430, and reduced phosphorylation of putative Bckdk targets, including Baf45a. Authors have demonstrated that injection of an RNP (Ribonucleoprotein) combining Cas13 with guide RNAs specific for Bckdk clearly impacts the early stages of zebrafish development, but from my assessment, the mechanism by which Bckdk functions has not been identified, and authors have not convincingly eliminated the possibility that off-target or indirect impacts of developmental delay account for the observed outcomes. Authors would need to perform an extensive series of additional experiments to establish function and eliminate these potential caveats.

We thank the reviewer for their positive feedback and thoughtful comments. In response to their concerns, we have conducted additional experiments specifically designed to address the points raised. We have also clarified several aspects of our study accordingly.

Our new data further support the conclusion that CRISPR-RfxCas13d is a robust and specific tool for mRNA knockdown (KD) in zebrafish embryos. Importantly, our results demonstrate that the molecular effects observed upon *bckdk* mRNA depletion are not a secondary consequence of the epiboly defects. Rather, the developmental abnormalities arise from the knockdown of *bckdk* mRNA. Below, we address and explain each of the reviewer's concerns and suggestions, all of which have helped us improve the clarity and strength of our study.

Major concerns -

1) Authors rely heavily on RNA-seq in an attempt to demonstrate that ZGA genes are specifically impacted by their manipulations. This is a flawed experimental design. This is because delayed development for any reason, irrespective of Bckdk function, would result in down-regulation of ZGA genes, because these genes are just turning on in embryos and preventing embryos from growing and developing properly would prevent these genes from turning on. Based on this logic, it is not at all surprising that all their manipulations cause ZGA genes to be mostly down-regulated. In the majority of cases (Figure 1F, 2E, 5C, S2, S5B, S5C, S5D) injection of their Cas13 guide RNA RNPs causes ZGA genes to be down-regulated.

We understand the reviewer's concern, and we apologize if our experimental conditions or descriptions were not sufficiently clear. However, we respectfully believe this may be an unfortunate misunderstanding.

The reviewer's concerns would be valid if epiboly defects were present at the time point when molecular characterization was performed. However, to determine whether ZGA was affected upon KD of different candidate genes—both those with and without developmental delay—we conducted polyA RNA-seq at 4 hours post fertilization (hpf). At this time point, no developmental differences were observed between candidate and control

embryos, and all embryos were appropriately staged at the sphere stage, as expected. **In fact, we specifically selected the 4 hpf time point to avoid potential confounding effects from developmental delay on genome-wide transcription.**

This stage (sphere; 4 hpf) is commonly used as a reference for transcriptomic and genomic studies investigating ZGA (e.g., Shan et al., 2019 Dev Cell; Lee et al., 2013 Nature; Miao et al., 2022 Mol Cell; Fukushima et al., 2024 EMBO Reports). In addition, we performed a more detailed developmental analysis for our primary candidate, *bckdk* mRNA KD. We observed no differences in total cell number, mitotic index, or developmental progression before 4 hpf between control and *Bckdk* KD embryos (see Extended Data Fig. 5H–J). As stated in the manuscript (line 149), for all candidates that exhibited epiboly defects, these alterations appeared only after 4 hpf. This phenomenon (ZGA alteration without developmental changes) has been previously observed by our group and others. For example, even with strong ZGA inhibition induced by α -amanitin or triptolide—treatments that completely block RNA polymerase II-dependent transcription—no developmental abnormalities are evident at 4 hpf. Arrests occur only after this time, when gastrulation and epiboly begin (Lee et al., 2013; Joseph et al., 2017 eLife; Shan et al., 2019; Miao et al., 2022; Fukushima et al., 2024). Therefore, all transcriptomic data presented in our study (e.g., Figures 1E and F, 2F, 3B, 5C–E, Extended Data Fig. 2 and 3, 6B–6D) were collected at a developmental time point when all embryos—zebrafish and medaka—were at the same stage. Consequently, transcriptomic differences cannot be attributed to developmental timing discrepancies resulting from the KD. We have added a small paragraph to clarify this point (lines 183–187). This has been similarly applied to other omic or molecular biology experiments such as CUT&RUN, ATAC-seq, proteomics and phospho-proteomics or different western blots all performed at 4 hpf.

Furthermore, we note that for some of the KD conditions that did produce developmental defects at 6 hpf (*Cab39l*, *calm1a*, *calm2a*, *mknk1a*, and *ppp4r2a*), none showed significant ZGA disruption at 4 hpf (Pure Zygotic Genes, (PZG) Normalized Enrichment Scores between -2 and 2 ; see Fig. 1F and Extended Data Fig. 3). These findings demonstrate that not all KDs resulting in later developmental phenotypes necessarily impact ZGA.

Finally, while we acknowledge that RNA-seq-based detection of ZGA perturbation may have inherent variability, we analyzed a broad set of knockdowns (14 candidates plus two new negative controls, see below) with and without phenotypes to define what constitutes a ZGA perturbation signature. This comprehensive approach enabled us to identify the most robust candidates in our screen. While most knockdowns showed a Pure Zygotic Genes (PZG) Normalize Enrichment Score between -2 and 2 , *mknk2a* and especially *bckdk* mRNA KD showed significant global downregulation of PZG expression.

To further support our conclusions, we have now included additional negative controls using non-targeting gRNAs against *gfp* and *rfp* mRNAs, which are not present in the wild-type zebrafish transcriptome, as well as a positive control gRNA targeting *nanog* mRNA. As shown in updated Fig. 1C and E and Extended Data Fig. 2 and 3, the new negative controls did not induce epiboly defects or transcriptomic changes at 4 hpf, while *Nanog* KD induced the expected developmental phenotype after 4 hpf (Kushawah et al., 2020 Dev Cell; Lee et al., 2013).

In summary, despite possible limitations—as with any experimental approach—we have demonstrated that our design is well-suited to detect significant ZGA perturbations at a stage when developmental defects are not yet evident. This allows us to confidently identify regulators of the maternal-to-zygotic transition (MZT) for further study.

Additionally, it is quite concerning that their manipulations cause several non-ZGA genes to be impacted. Is this do to off-target effects?

In the next point we answer this question regarding non-ZGA genes altered upon different candidates KD (broader transcriptome effect) and our comprehensive analysis of the off-targets (off-target activity).

2) Cas13 non-targeting guide RNA controls are necessary for all experiments. This is because the binding of Cas13 to gRNAs could impact any off-target outcomes, which are wholly unexplored in this manuscript. Comparison of their RNA-Seq results generated from targeting Cas13 experiments with RNA-Seq from negative controls, including the Cas13 alone and the non-targeting Cas13 RNP injection, is necessary to eliminate this concern.

We understand the reviewer's concern, as this issue has generated significant discussion regarding CRISPR-based RNA targeting in mammalian systems. Bellow, we discuss and address off-targets point and other questions potentially related to the specificity of the system.

Off-Target Activity:

CRISPR-RfxCas13d off-targets are typically defined as transcripts that contain exact or near-exact matches (up to three mismatches) with the gRNA target region (Wessels et al., 2024 Nat Biotechnol; Wei et al., 2023 Cell Syst; Shembrey et al., 2024 Sci Adv). Multiple studies, including our own, have demonstrated that CRISPR-RfxCas13d does not induce significant off-target effects, either *in vivo* or *ex vivo* (Kushawah et al., 2020 Dev Cell; Konermann et al., 2018 Cell; Wessels et al., 2020 Nat Biotechnol). In our previous work (Kushawah et al., 2020), we compared knockdown conditions using Cas13d alone and found no off-target effects for multiple gRNAs, confirming the specificity and efficacy of the system in zebrafish and other vertebrate embryos.

As suggested by the reviewer, we now reanalyzed our RNA-seq data (48 gRNAs from 16 knockdowns, including two new non-targeting controls for *gfp* and *rfp* mRNA) for potential off-target effects. Consistent with previous findings, we did not detect any significant downregulation of potential off-target transcripts (up to three mismatches) under our optimized conditions (see new Extended Data Fig. 2). The only exception was the calmodulin family, where high sequence similarity between paralogs led to partial knockdown of related transcripts. Specifically, while *calm1a* and *calm2a* were depleted by 99% and 95%, respectively, we also observed depletion of *calm3a* and *calm2b* likely due to gRNAs with perfect or near-perfect matches. This case reflects a unique challenge in targeting gene families with high sequence identity. In all other transcriptomes (n=14), no significant off-target effects were detected. We have included this analysis in the manuscript (lines 189–199 and new Extended Data Fig. 2).

Collateral Activity:

The reviewer may also be referring to the collateral activity described for Cas13 systems, in which specific on-target cleavage activates, and exposes through a conformational change, the nuclease domains of Cas13, resulting in non-specific RNA degradation. This effect has been reported under conditions of high gRNA and Cas13 expression through strong

promoters when targeting highly abundant RNAs ultimately leading to cellular toxicity, rRNA degradation, and reduced RNA integrity (Tong et al., 2023 Nat Biotechnol; Shi et al., 2023 Commun Biol; Ai et al., 2022 NAR; Kelley et al., 2022 Cell Rep; Li et al., 2023 Genome Biol). Notably, it has been recently reported that this effect can be highly mitigated when Cas13d expression is not high and comes from low-copy Cas13d lines (Hart SK et al., Nature Bio.).

Using our optimized transient delivery method (Cas13d mRNA or RNPs injected at the one-cell stage), we do not observe these collateral effects. We have recently published a comprehensive study (Moreno-Sanchez I et al., 2025 Nat Communications) showing that collateral activity in zebrafish is only detected when ectopic, highly abundant transcripts such as *gfp* or *dsRed* are targeted. Targeting highly expressed endogenous mRNAs (e.g., top 20 most abundant polyadenylated transcripts during the first 6 hpf) did not induce collateral effects, even with multiple detection methods. The most sensitive assay—28S rRNA fragmentation—did not reveal any degradation in our screen (see Extended Data Fig. 4), supporting that no collateral activity occurred under any condition tested. This is described in lines 214-222 and discussed in lines 547-555.

Broader Transcriptome Effects:

We agree that KD of maternal transcripts could lead to broader transcriptional changes beyond ZGA-specific genes. While these secondary effects are outside the primary scope of our study, we analyzed them and did observe that certain KDs (e.g., calmodulins and *cab39l*) share similar transcriptomic signatures, suggesting related roles in early development (Extended Data 1G; manuscript lines 209-213). Further investigation into the mechanisms behind these shared changes may be valuable, but falls beyond the goals of the current study, which focuses on kinases and phosphatases directly regulating ZGA (discussed in lines 578–589).

Negative Controls:

While similar controls were previously included in our earlier work (Kushawah et al., 2020), we appreciate the reviewer's suggestion to include additional negative controls here to further validate the robustness of our approach. Therefore, we have now included non-targeting gRNAs (*gfp* and *rfp*) co-injected with purified RfxCas13d protein. These conditions were compared to Cas13d-only-injected controls for both developmental phenotype and transcriptomic impact. As shown in Fig. 1C, no significant developmental delay was observed (fewer than 15% of embryos at 50% or 30% epiboly, similar to controls). RNA-seq at 4 hpf showed no significant impact on PZG expression (Normalize Enrichment Score = 1.35 and -1.66 for *gfp* and *rfp* gRNAs, respectively; Fig. 1E). These data confirm that the strong PZG downregulation observed in *mnk2a* and particularly *bckdk* mRNA KD is specific to loss of those transcripts. This information has been added to lines 182-183 and 205 (Fig. 1E and F).

Rescue Experiments and Additional Controls:

Our rescue experiments for *bckdk* mRNA KD using cognate or a dead-kinase mutant *bckdk* mRNA, or a phospho-mimicking version of Phf10 (Fig. 2A-C, Fig 5F and G and Extended Fig. 5K-M and 8G-K) also support the specificity of both the molecular and developmental phenotypes.

Furthermore, in this work we show that independent gRNAs targeting *bckdk* or *phf10* mRNAs displayed similar developmental phenotype (Extended Data Figure 5A-C and 8D). Comparably, we have previously shown that several independent gRNAs targeting the same mRNA show similar molecular and/or phenotype (Kushawah G et al., 2020, Da Silva Pescador G et al., 2024 Moreno-Sanchez et al., 2025).

Moreover, our current study already includes 7 RNP negative controls (21 gRNAs in total) targeting endogenous transcripts that: i) Do not cause developmental phenotypes, ii) induce significant and specific mRNA depletion, iii) do not substantially alter ZGA (PZG Normalize Enrichment Score between -2 and $+2$), and iv) do not trigger collateral activity.

Altogether, we believe that our new analyses, additional controls, and supporting data from previous studies comprehensively address the reviewer's concerns. We are confident that our manuscript now provides a clear and thorough evaluation of CRISPR-RfxCas13d specificity and reliability in zebrafish embryos.

3) Ectopic introduction of separate factors seems to rescue Bckdk knockdown, suggesting that Bckdk is a regulator of independent biological pathways. Authors need to provide a mechanistic explanation and additional experimental evidence to explain these outcomes. Does Bckdk regulate mir430, Baf45, or both? How can it be both?

We appreciate the reviewer's comments on our rescue experiments addressing the molecular and developmental effects of *bckdk* mRNA depletion. Our findings suggest that Bckdk regulates the maternal-to-zygotic transition (MZT) through distinct and likely independent mechanisms. As a kinase, Bckdk has the potential to regulate multiple signaling pathways, which was a key rationale behind our decision to focus on kinases and phosphatases in early embryogenesis.

In this study, we show that Bckdk: i) Modulates miR-430 biogenesis, affecting its function; and ii) Promotes Phf10/Baf45a phosphorylation, which is important for early zebrafish development, since a) Phf10 depletion causes epiboly defects and b) a Phf10 phospho-mimicking version specifically rescues the developmental delay and H3K27ac wild-type levels (new data) upon Bckdk KD. Below, we summarize what we did and what we have now performed to further clarify the molecular mechanisms driven by Bckdk controlling MZT.

1. Bckdk Regulates miR-430 Biogenesis

To distinguish between a transcriptional versus a biogenesis defect in miR-430 upon *bckdk* mRNA KD, we performed RT-qPCR for both primary and mature forms of the miRNA. The results showed a specific downregulation of mature miR-430 in Bckdk KD embryos, with no change in primary transcript levels. We then performed RNA-seq at 6 hpf, a time when maternal mRNAs are being actively degraded. We observed stabilization of transcripts known to be direct targets of miR-430. While transcripts whose degradation was depending in ZGA were also affected (although less than miR-430 targets), maternally cleared RNAs were much less affected, highlighting the specificity of the effect.

To further validate these results, we performed a rescue experiment by injecting mature miR-430 in Bckdk depleted embryos. This restored the degradation of miR-430 targets that were otherwise stabilized in Bckdk KD embryos. While we had previously examined three miR-430 targets by qRT-PCR, we now provide transcriptome-wide analysis (new Fig. 4D). This analysis showed a significant and specific depletion of known miR-430 target transcripts (categorized into "miR-430" and "miR-430 golden" groups—see Methods) in the rescue condition, compared to the rest of the transcriptome. All these results together confirm that Bckdk promotes miR-430 biogenesis and, therefore, regulates its activity. We have updated the manuscript accordingly (lines 431–436).

Although we have not yet uncovered the precise molecular mechanism by which Bckdk regulates miR-430 biogenesis—due to the resolution limits of our proteomic and

phospho-proteomic datasets—we have acknowledged this limitation in the Discussion (lines 687–694). Interestingly, some miR-430 targets remained stable even after miRNA rescue. This could be due to incomplete penetrance or additional regulatory effects on miR-430 activity by Bckdk. Notably, our phospho-proteome data identified reduced phosphorylation of Ago3a in bckdk KD embryos. Whether phosphorylation of this Argonaute protein modulates miR-430 function remains an open question and is discussed in the manuscript (lines 679–686).

2. Bckdk Controls Phf10 Function via Phosphorylation

Our data indicate that Bckdk also regulates early development by phosphorylating Phf10 at T16. Only the phospho-mimicking version of Phf10 (and not the wild-type or non-phosphorylatable forms) was able to rescue the developmental delay and restore H3K27ac levels in Bckdk KD embryos (new Fig. 5F and G, Extended data Fig. 8H-K lines 500–511), suggesting that this phosphorylation event is functionally critical.

To investigate the potential interaction between Bckdk and Phf10, we have now performed co-immunoprecipitation experiments in zebrafish embryos expressing bckdk and Phf10-HA or Gfp-HA (as a control). Bckdk was specifically detected by western blot only in the Phf10 pull-down, indicating an *in vivo* interaction. These results are shown in the new panel of Fig. 5H Extended Data Fig. 8 L, with two independent replicates included in the source data and the description added to the main text (lines 511–516).

Although it remains to be determined whether Bckdk directly phosphorylates Phf10, two lines of evidence support this possibility: (i) Bckdk and Phf10 interact *in vivo*, and (ii) our phospho-proteome data were collected only 4 h after egg fertilization. This short-term effect suggests the observed changes in phosphorylation may be direct (Discussed in lines 658-662)

Altogether, our results demonstrate that Bckdk promotes miR-430 biogenesis and regulates Phf10 phosphorylation—both critical for MZT and early zebrafish development. These findings significantly advance our understanding of how Bckdk influences early embryogenesis through multiple molecular pathways. Importantly, the fact that Bckdk regulates distinct and independent mechanisms highlights the potential for other proteins regulating protein phosphorylation to play similarly multifaceted roles during early development, supporting the broader relevance of investigating kinases and phosphatases in the context of MZT and the need to knock them down.

It is also concerning that rescuing Bckdk knockdown required the addition of 6 to 30-fold more mRNA compared with endogenous mRNA levels.

We appreciate reviewer comments. This is an interesting point that we acknowledge in the text when we mentioned the mosaic nature of the embryo microinjection procedure (lines 243-244). In this rescue experiment, there is a mosaicism with not only CRISPR-Cas13d reagents (see below) but also with ectopic mRNAs used for the rescue that may ultimately require increasing the concentration of reagents (without causing any effect in WT embryos as we showed) to rescue the developmental phenotype in all embryos. Additionally, the increased amount of *bckdk* mRNA required for phenotypic rescue could be due to a lower translation efficiency compared to the endogenous transcript. The need for elevated mRNA levels to achieve rescue is not uncommon in zebrafish and has been reported for other genes. For instance, *nanog* maternal and zygotic mutant rescue requires injection of 120 pg of mRNA per embryo—well above the estimated endogenous level (Kuznetsova et al., 2023; Pálffy et al., 2020; Veil et al., 2018).

Importantly, this rescue experiment indicates that the developmental phenotype results from the loss of maternal and cytosolic *bckdk* mRNA. Injection of an equivalent amount of *gfp* mRNA failed to rescue the epiboly defect, supporting the specificity of the rescue. This also directly addresses the reviewer's concern regarding potential off-target effects (point 2), demonstrating that the observed phenotype is not due to unintended activity of the CRISPR-Cas13d system.

4) It is concerning that only 40% of embryos are impacted by *Bckdk* knockdown. Is this because the majority of embryos are able to somehow overcome *Bckdk* knockdown or is it because the Cas13 technology is ineffective? This raises additional concerns about the effectiveness of their methodology and potential off-target effects.

We appreciate the reviewer's insightful comments and are grateful for the opportunity to address them. Under optimal conditions, utilizing three gRNAs concurrently, we observed epiboly defects in approximately 60–70% of embryos when targeting *bckdk* mRNA (Fig. 1C). However, transient RNA targeting mediated by CRISPR-RfxCas13d has inherent limitations, some of which we have discussed here and in our previous report (Kushawah et al., 2020).

First, compared to maternal mutants, this approach may have a reduced impact due to the presence of maternally provided proteins that can buffer the effects of maternal mRNA knockdown. The contribution of maternal proteins can vary slightly between cells or embryos, influencing individual knockdown penetrance. However, obtaining maternal mutants for many genes is challenging due to potential lethality or infertility associated with such mutations (Kushawah et al., 2024, BioRxiv PMID: 39574587). Therefore, despite its limitations, CRISPR-RfxCas13d offers an alternative approach, as previously discussed (Kushawah et al., 2020).

Second, the CRISPR-RfxCas13d method can result in mosaic and transient targeting. While some embryos may exhibit complete elimination of the target mRNA, others may show partial reduction, even when the average decrease is globally high. The efficiency of depletion varies among gRNAs, significantly impacting phenotype penetrance, as demonstrated in our previous reports (Kushawah et al., 2020; Moreno-Sanchez et al., 2025). Consequently, varying targeting efficacy per embryo or cell can contribute to mosaic phenotypes.

Third, the timing and kinetics of mRNA depletion might influence phenotype penetrance, especially when molecular perturbations are assessed as early as 3.5–4 hours post-fertilization (hpf). For instance, embryos with near-complete mRNA reduction at 2 hpf may exhibit more penetrant phenotypes (e.g., embryos at 30% epiboly) than those where maximal depletion occurs later. We measured mRNA levels at 4 hpf alongside RNA-seq data assessing zygotic genome activation for Pure Zygotic Genes (PZG) (Fig. 1D-F, Extended Data Fig. 3). However, we lack information on whether maximal depletion levels occurred earlier or later for each candidate and, importantly, the knockdown kinetics per embryo. This mosaicism was observed in Kushawah et al. (2020), and, in that report, we acknowledged these limitations, which are also present in other transient approaches such as morpholinos or antisense oligonucleotides (ASOs).

Especially, achieving a high level of *bckdk* mRNA depletion is necessary for obtaining highly penetrant phenotypes. gRNAs targeting the 3' untranslated region (UTR) of *bckdk* mRNA induced lower mRNA depletion, resulting in reduced phenotype penetrance, suggesting that the observed phenotype is highly dependent on the degree of depletion. This observation aligns with findings from other mRNA knockdowns in our laboratory (Kushawah et al. 2020, Moreno-Sanchez et al., 2025). For example, targeting the *tbxta* 3' UTR produced a less

penetrant phenotype than targeting its open reading frame (Kushawah et al. 2020). Lower penetrance when targeting the 3' UTR is also documented for other CRISPR-Cas13 systems in mammalian cell culture assays (Wessels et al., 2020).

Furthermore, *Bckdk* depletion leads to upregulation of a subset of genes involved in translation and protein production during ZGA (Fig. 3B and Extended Data Fig. 6E-G). We speculate that this response attempts to compensate for decreased transcription of essential genes, as increased translation of these mRNAs could buffer global downregulation. This potential compensatory mechanism may vary among cells and embryos, adding to phenotype penetrance variability.

Additionally, the target gene's essentiality during early development is crucial. Even for fundamental ZGA factors such as *Nanog*, we did not achieve 100% phenotype penetrance, even with high depletion levels, compared to maternal and zygotic (MZ) mutants (Kushawah et al., 2020). However, the penetrance was comparable to previous descriptions using morpholinos (Lee et al., 2013). We have now included *nanog* mRNA depletion as a positive control (Fig. 1C), showing that 90–95% of embryos presented epiboly defects (between 30–50% epiboly), while MZ *nanog* mutants exhibited more severe developmental arrest (sphere–dome) in 100% of embryos (Veil et al., 2018). Notably, varying targeting levels can be useful for uncovering phenotypes that might be lethal with full penetrance.

This work serves as a proof-of-principle, demonstrating that CRISPR-RfxCas13d technology enables targeting of multiple mRNAs efficiently, uncovering new factors involved in MZT through maternal mRNA KD screenings. We demonstrate that this is feasible despite the approach's intrinsic mosaicism (per embryo or cell and due to knockdown kinetics), which was anticipated and previously discussed in our earlier work. A similar screening approach using MZ mutants would be more complex and costly, aside from technical challenges associated with potential lethality or infertility of maternal mutants (Kushawah et al., 2024. BioRxiv). Our optimized technology offers an alternative for revealing new factors controlling early development in zebrafish and other vertebrate embryos. Like any approach, it has limitations, advantages, and disadvantages. Although these points have been partially covered in our previous reports (Kushawah et al., 2020; Hernandez-Huertas et al., 2022), we have added a paragraph discussing these potential limitations in the discussion (lines 555–568).

Regarding off-target effects, we have extensively addressed this concern in a previous section.

The lead author's original paper describing these techniques (Kushawah et al) was published 4 years ago, and this research group remains the only member of the zebrafish community employing the method. For such a promising methodology, with the potential to fill a major gap in zebrafish genetics community (the inability to experimentally deplete maternal RNAs and investigate maternally provided gene function in embryos), it is concerning that this Cas13-based method has not been more widely adopted.

We respectfully disagree with the reviewer's opinion. Our lab published the first report using CRISPR-RfxCas13d in a transient approach (Kushawah G. et al., *Developmental Cell*, 2020). Subsequently, in 2022, we shared a comprehensive step-by-step protocol (Hernandez-Huertas L. et al., *STAR Protocols*), which we know is actively being used—not only by the zebrafish community studying early development, but also by researchers working in other vertebrate systems. For instance, our CRISPR-RfxCas13d plasmids have been requested over 100 times via Addgene. It is important to consider that 2020–2022 were heavily impacted by

the COVID-19 pandemic, which significantly slowed research productivity worldwide (e.g., Heo S. et al., 2022, PMID: 36530543). Even in our own labs, completing this and related projects where we further optimized the CRISPR-RfxCas13d technology took over three years (e.g., Moreno-Sanchez I. et al., Nature Communications, 2025; De Silva Pescador G. et al., Cell Reports, 2024; Kushawah G. et al., bioRxiv, 2024, and the present work).

Given this context, it is understandable that other laboratories—especially those less familiar with the technology—may have required more time to successfully adopt and implement it. To facilitate this process, and as mentioned earlier, we published a detailed protocol that includes a troubleshooting guide and practical solutions (Hernandez-Huertas L. et al., STAR Protocols).

Below, and as an example, we list several published studies in which our CRISPR-RfxCas13d technology (via RNP or mRNA-gRNA complexes) has been applied in zebrafish and other animal models:

Ferreira F. et al. Stretch-induced endogenous electric fields drive directed collective cell migration in vivo. *Nature Materials* (2025). Model System: *Xenopus*

Kim M. & Hutchins E.J. CRISPR-Cas13d as a molecular tool to achieve targeted gene expression knockdown in chick embryos. *Developmental Biology* (2025). Model system: Chick

Escot S. et al. Nance-Horan-Syndrome-like 1b controls mesodermal cell migration by regulating protrusion and actin dynamics during zebrafish gastrulation. *Comm. Biol.* (2025). Model system: zebrafish

Huang, F. et al. Integrative analysis identifies the atypical repressor E2F8 as a targetable transcriptional activator driving lethal prostate cancer. *Oncogene* (2025). Model system: Mouse

Shangguan, H. et al. Knockdown of Kmt2d leads to growth impairment by activating the Akt/ β -catenin signaling pathway. *G3 Genes|Genomes|Genetics* (2024). Model system: zebrafish

Nishimura T. et al. Sterilization of fish through adaptable gRNAs targeting *dnd1* using CRISPR-Cas13d system. *Aquaculture* (2024) Model system: medaka and rainbow trout

Del Prado, J. A.-N. et al. Comparing robotic and manual injection methods in zebrafish embryos for high-throughput RNA silencing using CRISPR-RfxCas13d. *Biotechniques* (2024). Model system: zebrafish

Zhu, W. et al. Reading and writing of mRNA m6A modification orchestrate maternal-to-zygotic transition in mice. *Genome Biol* (2023). Model system: mouse

Ma Y-F. et al. Efficient nanoparticle-based CRISPR-Cas13d induced mRNA disruption of an eye pigmentation gene in the white-backed planthopper, *Sogatella furcifera*. *Insect Science* (2023) Model System: Insect: white-backed planthoppers.

Serres MP et al. MiniBAR/GARRE1 is a dual Rac and Rab effector required for ciliogenesis. *Dev Cell*. 2023. Model system: zebrafish

Liu Y. et al. Characteristics of Shisa Family Genes in Zebrafish. *Int. J. Mol Sci.* (2023). Model system: zebrafish

Li S. et al. Linker histone H1FOO is required for bovine preimplantation development by regulating lineage specification and chromatin structure. *Biology of Reproduction* (2022). Model system: cattle

Accordingly, we have added some of these references in the discussion (line 530)

Is it too difficult to employ? Are there considerable off-target effects? Does it lack robustness? Authors seem to wholly ignore the deficiencies and focus only on their anticipated outcomes in impacted embryos.

Based on our results and those from other laboratories, the CRISPR-RfxCas13d transient approach in vertebrate embryos is a robust, straightforward, and specific method for mRNA knockdown. Recently, we further optimized this RNA-targeting system in zebrafish embryos (Moreno-Sanchez I. et al., *Nature Communications* 2025). In that study, we targeted multiple mRNAs and addressed several limitations of the technology, including nuclear RNA targeting, sustained transcript depletion for genes expressed after gastrulation, and the identification of highly efficient gRNAs using computational models previously developed in mammalian cell culture systems. Importantly, we also conducted a comprehensive analysis of potential collateral effects in zebrafish embryos under our optimized CRISPR-RfxCas13d knockdown conditions. As discussed above, our data confirm that the system is specific and does not induce significant collateral activity.

Moreover, last year we published a study demonstrating knockdown of *znf281b*, a maternal mRNA whose protein accumulates rapidly within the first two hours post-fertilization—similar to *Nanog*, *Pou5f3* (*Oct4*), and *Sox19b*—which affects zygotic genome activation, and its phenotype was also rescued (da Silva Pescador G. et al., *Cell Reports*, 2024). We also uploaded a preprint showing the knockdown of 10 spatiotemporally regulated maternal genes; five of these exhibited strong phenotypes at 24 hours, and three of them were fully rescued by ectopic expression of the target genes (Kushawah G. et al., 2024, bioRxiv). As mentioned earlier, to facilitate the adoption of this technology, we published a step-by-step protocol (Hernandez-Huertas L. et al., *STAR Protocols*, 2022), which includes clearly defined troubleshooting strategies and critical steps. This protocol has proven helpful to the broader scientific community working with zebrafish and other model organisms.

We and others have shown that CRISPR-RfxCas13d transient approaches (using RNP complexes or mRNA-gRNA formulations) are very robust phenocopying known loss-of-function mutants. Indeed, CRISPR-RfxCas13d can accurately reproduce well-characterized developmental loss-of-function phenotypes by targeting genes such as *tbxta* (no tail), *slc45a2* (albino), *dnd1*, *nanog*, *rx3*, among many others (Kushawah G. et al., 2020; Moreno-Sanchez I. et al., 2025). Furthermore, we were able to both rescue and reproduce known molecular effects and also identify novel regulators of the maternal-to-zygotic transition (MZT), such as *Brd3a*, whose developmental role had not been previously demonstrated (Kushawah G. et al., 2020). In addition, we have shown that our optimized CRISPR-RfxCas13d protocol does not significantly trigger collateral activity when targeting endogenous mRNAs—an important consideration, as collateral cleavage rather than off-target effects is the main concern in the Cas13 field. This has been confirmed both in this manuscript as a proof-of-principle that the technology can be systematically used for large-scale maternal mRNA screening and in our recent *Nature Communications* study. Besides, CRISPR-RfxCas13d can be successfully applied not only in zebrafish but also across different vertebrate embryos, as shown by our work (Kushawah et al, 2020) and others (see references above).

As we have emphasized, like any other method, CRISPR-RfxCas13d has its advantages and limitations. We have already discussed these in detail in our previous publication (Kushawah G. et al., *Developmental Cell*, 2020 and 2024). While we believe it is not necessary to re-demonstrate previously published findings in this manuscript, we have nevertheless

included a dedicated paragraph in the Discussion section (lines 555–568), where we address the possible limitations of using the CRISPR-RfxCas13d system that were also discussed in the limitations section of our earlier Developmental Cell article.

5) If Bckdk truly impacts Baf45, as authors conclude, then the resulting impacts on chromatin should be stronger. Why are chromatin accessibility impacts so modest? Why is there only a minor impact on H3K27ac? Furthermore, impacts on H3K27ac could be a mere consequence of developmental delay, rather than direct impacts from Bckdk loss.

We thank the reviewer for these insightful questions and will do our best to clarify them. Phf10/Baf45a is a component of the pBAF complex, a dynamic and multi-subunit member of the SWI/SNF family known to regulate gene expression through various mechanisms. As discussed in our manuscript, pBAF functions primarily as a reader of the H3K27ac epigenetic mark, rather than contributing to its deposition. However, it has been reported that disruptions in pBAF can lead to transcriptional downregulation and decreased histone acetylation (see references 91–93 in our manuscript). Our data show that chromatin accessibility is not significantly affected in Bckdk-depleted embryos in the context of ZGA disruption. However, we do observe a significant reduction in H3K27ac levels. One possible explanation is that loss of Bckdk leads to reduced phosphorylation of Phf10, which in turn impairs pBAF activity. This could result in decreased H3K27ac levels and, consequently, diminished transcription during early development.

Notably, we have performed a **new experiment** and show that only the phosphomimetic version of Phf10 was able to rescue both the developmental phenotype and H3K27ac levels (Fig. 5 F and G, Extended Data Fig. 8H-K), reinforcing the idea that Phf10 phosphorylation is necessary for maintaining H3K27ac and proper transcriptional regulation during development. We propose a scenario where H3K27ac levels are reduced without a global change in chromatin accessibility. This is consistent with previous findings: inhibition of histone acetylation has been shown to cause widespread downregulation of ZGA without affecting the chromatin accessibility landscape in zebrafish (Miao L. et al., 2023, Molecular Cell) and more recently in Drosophila embryos (Marsh AJ et al, BioRxiv, 2024). Altogether, these findings support the conclusion that Bckdk regulates H3K27ac levels through its effect on Phf10 phosphorylation.

Importantly, this reduction in H3K27ac is not due to developmental delay. As stated in our response to point 1 regarding RNA-seq data, these experiments were also conducted at 4 hpf, when control and Bckdk-depleted embryos displayed comparable developmental staging (Extended Data Fig. 5H–J).

Furthermore, we have now included CUT&RUN data analyzing H3K27ac, which show a global reduction in H3K27ac signal upon *bckdk* mRNA knockdown (52% of peaks affected: Fold change > 1.5 *p*-value < 0.05, Fig. 3E and F). While the resolution of CUT&RUN was lower than ChIP-seq, our results nonetheless confirm this global downregulation of H3K27ac.

We would also like to emphasize that a ~50% reduction in H3K27ac is not modest. In zebrafish embryos treated with SGC-CBP30, an inhibitor of p300/CBP activity, a comparable reduction in H3K27ac (at 5 µg/mL) results in developmental defects similar to those observed with *bckdk* knockdown (see Reviewer Figure 1 below) (Chan S.H. et al., 2019, Developmental Cell). Higher concentrations of SGC-CBP30 lead to even more dramatic reductions in H3K27ac and developmental arrest, comparable to that induced by α -amanitin or triptolide, which inhibit RNA polymerase II. While this was partially addressed in Chan et al., we wanted to

clarify this point and show to the reviewer that a 40-50% reduction in H3K27ac has significant biological consequences in zebrafish embryos.

Reviewer Figure 1. A) Stacked bar plots showing the percentage of phenotypes observed at 6 hpf in embryos treated with 5 μ g/ μ L or 20 μ g/ μ L SGC-CBP30 (a P300 inhibitor). Control embryos were treated with DMSO. Results represent the mean \pm standard error of the mean (SEM) from two independent experiments. Phenotype classification criteria were the same as in Fig. 1C. The number of embryos analyzed (n) is indicated for each condition. **B)** Barplots showing H3K27ac levels relative to total protein, presented as the mean \pm SEM of two biological replicates from two independent experiments. Embryos were treated with 5 μ g/ μ L or 20 μ g/ μ L SGC-CBP30 or DMSO (control) and collected at 4 hpf (n = 20 embryos per biological replicate). **C)** Representative western blot for H3K27ac levels in embryos treated as in (B), collected at 4 hpf (n = 20 embryos per biological replicate). Left: H3K27ac signal. Right: Stain-Free gel image as a loading control.

6) Authors spend the majority of the manuscript describing down-regulation impacts on zygotic gene transcription due to their many manipulations, but in figure 4A the zygotically transcribed genes appear to be more highly expressed in the Bckdk knockdown embryos. Why does this result not match the results of other data presented, such as 1F & 2E?

We apologize if Fig. 4A was not clear and caused a bit of confusion. In this figure, we show the levels of maternal transcripts at 6 hpf whose clearance depends on the activation of the zygotic genome (zygotic program), as defined by Vejnar C.E. et al. (2019, Genome Research) were upregulated. Therefore, in line with the ZGA disruption observed in *bckdk* mRNA knockdown embryos, we detected increased stability of this subgroup of maternal mRNAs.

Interestingly, this effect was most pronounced for transcripts targeted by miR-430, whose degradation is known to depend on the zygotic program. In contrast, we observed little effect on maternal mRNAs that are typically cleared by the maternal program—an mRNA clearance mechanism that operates independently of ZGA. This distinction further supports our interpretation that the transcriptomic perturbations seen at 6 hpf (a time point when Bckdk-depleted embryos exhibit epiboly defects) are primarily driven by ZGA disruption.

Therefore, the data presented in Fig. 4A reflect the dynamics of specific maternal mRNA populations at a later developmental stage and are conceptually distinct from the transcriptomic data shown in Figs. 1 and 2 and Extended Data Fig 2 and 3.

7) Knockdown of Bckdk caused both increased and decreased gene expression changes, but the authors completely ignored the increasing genes, potentially due to their narrow focus on ZGA. Could valuable information pertaining to mechanism be gleaned from deeper analysis of genes that are increasing?

We thank the reviewer for this suggestion, but we already analyzed this point in our initial manuscript. To specifically address the dynamics of the upregulated genes, we implemented labeled transcriptomics (SLAM-seq) in combination with CRISPR-Cas13 knockdown. Therefore, we used SLAM-seq to further investigate the expression of the zygotic genome activation (ZGA) program upon *Bckdk* knockdown. From the SLAM-seq data, we analyzed both purely zygotic genes (PZGs) and zygotically expressed genes with maternal contribution. We identified a subset of genes that were more highly transcribed at 4 hpf in *Bckdk*-depleted embryos (Fig. 3B), therefore these genes are increasing. We characterized these genes in terms of their expression dynamics, molecular functions, and normal timing of activation in wild-type embryos. These analyses are detailed in the Results section (lines 333–339; Fig. 3B and Extended Data Fig. 6E–G) and further discussed in the Discussion (lines 615–624). Our conclusions from this analysis are: i) The vast majority of upregulated genes are *bona fide* ZGA genes, normally activated between 4 and 7 hpf, ii) approximately 36% of these genes appear to be prematurely activated rather than simply overexpressed and iii) GO term enrichment analysis suggests that this upregulation reflects a compensatory response to the global downregulation of ZGA, as many of these genes are involved in translation and protein biogenesis.

In addition, we recently reported that, beyond targeting maternal transcripts, miR-430 can also regulate zygotically transcribed mRNAs during ZGA (Baia-Amaral D. et al., 2024). Given this and building on our additional analysis of more abundant transcripts at 4 hpf upon *Bckdk* knockdown, we also examined whether these upregulated mRNAs were enriched for miR-430 targets. This could help explain their increased abundance, especially in light of the reduced levels of mature miR-430 we observed (Fig. 4C). Interestingly, we found that 30% of the upregulated transcripts were known miR-430 targets (Baia-Amaral et al., 2024). We have now updated the text with this analysis in the manuscript (lines 339–344 and 620–623).

Altogether, these results indicate that impairment of miR-430 activity responsible for maternal mRNA clearance may already be detected at 4 hpf using SLAM-seq. This helps to partially explain the observed **upregulation of a subset of genes** belonging to different waves of the ZGA.

8) For Figure 3E, rather than investigating expression changes for genes that are both DEGs and associated with accessibility changes, it would be more appropriate for authors to assess overall gene expression changes for all genes that are associated with increased or decreased chromatin accessibility.

We thank the reviewer for this suggestion. We have performed a similar analysis and observed a slight but statistically significant association between genes with reduced chromatin accessibility (≥ 3 -fold decrease) and lower expression levels. However, due to the relatively low number of differentially accessible peaks—and the inherent uncertainty in confidently assigning these peaks to specific target genes (which may include false positives)—no additional strong conclusions could be drawn from this analysis beyond what we had already reported. Overall, our data continue to support the conclusion that chromatin accessibility is only minimally affected upon *bckdk* mRNA knockdown.

Minor concerns:

1. The authors quantified mRNA levels after Cas13d knockdown, could they use an antibody for *bckdk* or an affinity tagged version?

We appreciate the reviewer's question. Our proteomic analysis revealed a modest but detectable reduction in Bckdk protein levels in *bckdk* mRNA knockdown embryos (log₂ fold change of -0.2; Supplementary Table 2, highlighted now in the text, lines: 600-603). This decrease is smaller than the corresponding mRNA depletion, which is expected given that a substantial portion of Bckdk protein is likely maternally deposited within the mitochondrial fraction. Based on our data strongly suggesting an unaltered mitochondrial activity upon Bckdk KD (Fig. 2D and E and Extended data Fig. 8A), we think this maternally provided mitochondrial pool constitutes most of the detectable Bckdk protein and continues to function normally in mitochondria, as evidenced by the similar phosphorylation levels of its main known target in the mitochondria, Bckdha, between control and knockdown embryos (Fig. 5A).

To distinguish between mitochondrial and non-mitochondrial Bckdk pools, we attempted immunofluorescence analysis. However, the mitochondrial signal at 4 hpf in zebrafish embryos was not clearly resolved, unlike in mammalian cell culture systems. As a result, it was technically challenging to specifically assess the depletion of the non-mitochondrial Bckdk fraction—presumably the one responsible for the early developmental defects. However, the similar BCAA levels and Bckdha phosphorylation patterns together with the lack of mitochondrial proteins with an altered phosphorylation level strongly support the conclusion that Bckdk mitochondrial activity is intact and loss of the non-mitochondrial Bckdk pool is driving the maternal-to-zygotic transition (MZT) defects (Discussed in lines 455-464 and 591-600).

2. Rather than looking at total H3K27ac levels it would be more informative if authors investigate H3K27ac changes via CHIP-Seq or CUT&Run.

We agree with the reviewer that their suggested experiment can consolidate the results on H3K27 ac changes **at the genomic level. We have now carried out a CUT&RUN to investigate H3K27ac levels in WT and BCKDK-depleted embryos** and observed a global reduction in H3K27ac levels upon Bckdk KD. Indeed, we detected a reduction in 52% of identified peaks along zebrafish genome at 4 hpf (decreased of > 1.5-fold change *p-value* < 0.05). Therefore, this data independently confirms our previous results based on western-blot analysis and demonstrate that *bckdk* mRNA KD triggers H3K27ac depletion.

3. Figure 5F – Why does phf10 WT mRNA injection exacerbate the epiboly defects? Does phf10 mRNA injection alone for the phenotypes?

We agree with the reviewer that this is an interesting point. Overexpression of Phf10, regardless of its phosphorylation state, induces epiboly defects (Extended Data Fig. 8F and G), suggesting that perturbing the balance of the endogenous pBAF complex can disrupt early development—likely impacting ZGA as well. This is further supported by our findings showing reduced Phf10 phosphorylation in *bckdk* mRNA knockdown embryos (Fig. 5A and F) and even more striking effects following *phf10* mRNA knockdown (Fig. 5B–D).

For the Bckdk KD rescue experiments, we carefully used concentrations of *phf10* mRNAs that do not produce significant developmental defects when injected into wild-type embryos. Initially, our rescue data were based on two independent experiments. However, we have now repeated these experiments several times. With the inclusion of these new replicates, it is clear that the previously observed exacerbated phenotype upon WT *phf10* overexpression was likely due to some experimental variability and has been mitigated by the expanded dataset.

Additionally, these new replicates were used to assess H3K27ac levels following injection of different *phf10* mRNA variants. Notably, only the phosphomimetic version of Phf10 was able to fully restore H3K27ac levels to those of wild-type embryos, which correlates with the successful developmental rescue observed under these conditions (New Fig. 5F and G and Extended Data Fig. 8 H-K). In contrast, neither the wild type nor the non-phosphorylatable versions of Phf10 were able to achieve the same effect, reinforcing the importance of Phf10's phosphorylation state in regulating early development and chromatin modification.

Reviewer #2:

Remarks to the Author:

A. Summary of the key results

The manuscript by Hernandez-Huertas et al. describes developing an RfxCas13d targeted RNA degradation screen to identify maternal RNA critical for maternal-zygotic transition in zebrafish. From 49 targets that are maternally expressed and annotated to regulate protein phosphorylation, the authors identified 7 affecting gastrulation. They focused on Bckdk, a kinase that regulates branched amino acid (BCAA) catabolism. They showed that the phenotype of Bckdk knockdown can be rescued by a RfxCas13d-resistant bckdk mRNA. Bckdk knockdown decreases the expression of pure zygotic genes and impairs miR-430 biogenesis without affecting mitochondrial BCAA catabolism. Instead, Bckdk knockdown reduces H3K27Ac levels and subtly changes chromatin accessibility, discovering a new function of this kinase. Bckdk knockdown also causes epiboly defects in medaka, another fish model. To identify the potential mediator of the epigenetic changes, they performed a phospho-proteomic analysis and found that Bckdk knockdown reduces T16 phosphorylation of Phf10/Baf45a, a component of the Polybromo-associated BAF (pBAF) complex. They demonstrated that a phospho-mimetic Phf10-T16D can rescue the epiboly defects in Bckdk knockdown embryos.

B. Originality and significance

The study is the first to apply RfxCas13d-mediated mRNA degradation in identifying maternal mRNAs important for MZT. These findings that Bckdk and Phf10 are important for MZT are novel.

C. Data & methodology: validity of approach, quality of data, quality of presentation

The data are of high quality and methods are appropriate and state of the art. The manuscript is clearly written.

D. Appropriate use of statistics and treatment of uncertainties

Proper statistical analyses were used.

E. Conclusions: robustness, validity, reliability

Conclusions were largely based on solid results.

We really appreciate reviewer's comments and analysis of our manuscript highlighting the novelty of the use of CRISPR-RfxCas13d approach for maternal screening, the discovery of new factors involved in MZT and the quality and robustness of the methods and data supporting our conclusions.

F. Suggested improvements: experiments, data for possible revision:

1) Given the effectiveness of the mRNA knockdown (>92%), it is unclear why only a small

fraction of the embryos displayed epiboly defects. Is it due to individual differences in protein levels of these mRNAs? There needs to be a solid explanation for the low penetrance.

We appreciate the reviewer's insightful comment, which was also raised by Reviewer 1 (comment #4), and we are therefore providing the same response below.

Under optimal conditions, utilizing three guide RNAs (gRNAs) concurrently, we observed epiboly defects in approximately 60–70% of embryos when targeting *bckdk* mRNA (Fig. 1C). However, transient RNA targeting mediated by CRISPR-RfxCas13d has inherent limitations, some of which we have previously discussed (Kushawah et al., 2020).

First, compared to maternal mutants, this approach may have a reduced impact due to the presence of maternally provided proteins that can buffer the effects of maternal mRNA knockdown. The contribution of maternal proteins can vary slightly between cells or embryos, influencing individual knockdown penetrance. However, obtaining maternal mutants for many genes is challenging due to potential lethality or infertility associated with such mutations (Kushawah et al., 2024, BioRxiv PMID: 39574587). Therefore, despite its limitations, CRISPR-RfxCas13d offers an alternative approach, as previously discussed (Kushawah et al., 2020).

Second, the CRISPR-RfxCas13d method can result in mosaic and transient targeting. While some embryos may exhibit complete elimination of the target mRNA, others may show partial reduction, even when the average decrease is globally high. The efficiency of depletion varies among gRNAs, significantly impacting phenotype penetrance, as demonstrated in our previous reports (Kushawah et al., 2020; Moreno-Sanchez et al., 2025). Consequently, varying targeting efficacy per embryo or cell can contribute to mosaic phenotypes.

Third, the timing and kinetics of mRNA depletion might influence phenotype penetrance, especially when molecular perturbations are assessed as early as 3.5–4 hours post-fertilization (hpf). For instance, embryos with near-complete mRNA reduction at 2 hpf may exhibit more penetrant phenotypes (e.g., embryos at 30% epiboly) than those where maximal depletion occurs later. We measured mRNA levels at 4 hpf alongside RNA-seq data assessing zygotic genome activation for Pure Zygotic Genes (PZG) (Fig. 1D-F, Extended Data Fig. 3). However, we lack information on whether maximal depletion levels occurred earlier or later for each candidate and, importantly, the knockdown kinetics per embryo. This mosaicism was observed in Kushawah et al. (2020), and, in that report, we acknowledged these limitations, which are also present in other transient approaches such as morpholinos or antisense oligonucleotides (ASOs).

Especially, achieving a high level of *bckdk* mRNA depletion is necessary for obtaining highly penetrant phenotypes. gRNAs targeting the 3' untranslated region (UTR) of *bckdk* mRNA induced lower mRNA depletion, resulting in reduced phenotype penetrance, suggesting that the observed phenotype is highly dependent on the degree of depletion. This observation aligns with findings from other mRNA knockdowns in our laboratory (Kushawah et al. 2020, Moreno-Sanchez et al., 2025). For example, targeting the *tbxta* 3' UTR produced a less penetrant phenotype than targeting its open reading frame (Kushawah et al. 2020). Lower penetrance when targeting the 3' UTR is also documented for other CRISPR-Cas13 systems in mammalian cell culture assays (Wessels et al., 2020).

Furthermore, *Bckdk* depletion leads to upregulation of a subset of genes involved in translation and protein production during ZGA (Fig. 3B and Extended Data Fig. 6E-G). We speculate that this response attempts to compensate for decreased transcription of essential genes, as increased translation of these mRNAs could buffer global downregulation. This

potential compensatory mechanism may vary among cells and embryos, adding to phenotype penetrance variability.

Additionally, the target gene's essentiality during early development is crucial. Even for fundamental ZGA factors such as *Nanog*, we did not achieve 100% phenotype penetrance, even with high depletion levels, compared to maternal and zygotic (MZ) mutants (Kushawah et al., 2020). However, the penetrance was comparable to previous descriptions using morpholinos (Lee et al., 2013). We have now included *nanog* mRNA depletion as a positive control (Fig. 1C), showing that 90–95% of embryos presented epiboly defects (between 30–50% epiboly), while MZ *nanog* mutants exhibited more severe developmental arrest (sphere–dome) in 100% of embryos (Veil et al., 2018). Notably, varying targeting levels can be useful for uncovering phenotypes that might be lethal with full penetrance.

This work serves as a proof-of-principle, demonstrating that CRISPR-RfxCas13d technology enables targeting of multiple mRNAs efficiently, uncovering new factors involved in MZT through maternal mRNA KD screenings. We demonstrate that this is feasible despite the approach's intrinsic mosaicism (per embryo or cell and due to knockdown kinetics), which was anticipated and previously discussed in our earlier work. A similar screening approach using MZ mutants would be more complex and costly, aside from technical challenges associated with potential lethality or infertility of maternal mutants (Kushawah et al., 2024. BioRxiv). Our optimized technology offers an alternative for revealing new factors controlling early development in zebrafish and other vertebrate embryos. Like any approach, it has limitations, advantages, and disadvantages. Although these points have been partially covered in our previous reports (Kushawah et al., 2020; Hernandez-Huertas et al., 2022), we have added a paragraph discussing these potential limitations in the discussion (lines 555–568).

2) The relationship between Bckdk and Phf10 can be made clearer. The current data shows that Phf10 is downstream of Bckdk but does not demonstrate that Phf10 is a substrate of Bckdk, as implied by the authors.

We agree with the reviewer that we have not provided unequivocal evidence demonstrating that Phf10 is a direct substrate of Bckdk, and we originally acknowledged this in the manuscript. However, to address this point, we have now conducted a series of experiments showing that Bckdk interacts with Phf10 in zebrafish embryos and strongly suggesting that it regulates its phosphorylation state.

First, to demonstrate that Bckdk and Phf10 interact *in vivo*, we performed a co-immunoprecipitation assay. We overexpressed in zebrafish embryos Bckdk together with *phf10-HA* mRNA, and with *gfp-HA* mRNA as a negative control. After 6 hours post injection, we performed the immunoprecipitation of the HA epitope and we carried out a western blot to detect Bckdk protein. Remarkably, we only observed Bckdk in the elution fraction coming from the embryos injected with Phf10-HA indicating that Bckdk and Phf10 interact *in vivo*. We have now, included a new panel in Fig. 5H and Extended Data Figure 8L and added our 2 independent replicates to the source data. This information has been also included into the main text (lines 511-516).

Furthermore, we have completed our rescue experiments using different *phf10* mRNA variants and confirmed that only the phosphomimetic version of Phf10 can effectively rescue the developmental defects caused by Bckdk KD (Fig. 5F and Extended Data Fig. 8 H and I). Importantly, this version also restores H3K27ac levels to those observed in wild-type embryos

(Fig. 5G, Extended Data Fig. 8 J and K lines 506-511) while other versions (wild type and non-phosphorylatable) are unable to do it.

Lastly, our phosphoproteomic analysis was performed shortly after fertilization and Bckdk depletion (4 hours). This short timeframe supports the idea that many of the peptides showing reduced phosphorylation may represent direct Bckdk targets.

Altogether, our data support a model in which Bckdk interacts with Phf10 (likely in a direct manner) and regulates its phosphorylation state, which is essential for proper early development in zebrafish.

3) It remains unclear how Bckdk enhances H3K27ac. As the authors pointed out, the Phf10-containing pBAF is a reader rather than a depositor of H3K27ac. Is the kinase activity necessary for increased H3K27ac? This may be addressed by injecting mRNA encoding a kinase-dead mutant protein in control and *bckdk* KD embryos.

As the reviewer mentioned, Phf10 is a component of the pBAF complex, which has been described more as a reader than a writer of H3K27ac. However, as discussed in the manuscript, there are biological contexts in which pBAF disruption can lead to reduced transcription and decreased histone acetylation (Discussed in lines 650-652).

As suggested by the reviewer, we designed a *bckdk* mRNA mutant based on a previously described point mutation (Y348A), known to reduce kinase activity by approximately 95% (Wynn et al., 2000, Singh et al., 2024). We used this kinase-dead mutant mRNA in a rescue experiment following *bckdk* mRNA knockdown, using a gRNA targeting the 3' UTR of *bckdk*. Interestingly, this mutant isoform failed to rescue the developmental phenotype (Fig. 2C and Extended Data Fig. 5K-M; lines 264–277). These results demonstrate that Bckdk's kinase activity is essential for proper early development.

Additionally, as described above, we completed our *bckdk* mRNA knockdown rescue experiments using different *phf10* mRNA variants. We confirmed that the developmental phenotype is specifically and almost fully rescued by the phosphomimetic version of Phf10 (Fig. 5F and Extended Data Fig. 8H and I). Furthermore, only this phosphomimetic version was capable of restoring H3K27ac levels to those seen in wild-type embryos Fig. 5G and Extended Data Fig. 8J and K).

Altogether, these data indicate that Bckdk's kinase activity is required to regulate the MZT and to control Phf10 phosphorylation and consequently H3K27ac levels and ZGA in zebrafish.

Editorial:

Ln36-37, the phrase "mRNAs encoding protein kinases and phosphatases" is imprecise. Many of the mRNAs encode regulators of protein kinases and phosphatases, but themselves are not protein kinases and phosphatases.

Ln42-43, 112-113, the phrase "Phf10 constitutively phosphorylated" is inaccurate. The mutation mimics phosphorylation but cannot be phosphorylated.

Ln353, the phrase "maternal and zygotic encoded mechanisms" should be "maternally and zygotically encoded mechanisms".

Ln494-495, the phrase "three kinases linked to calcium signaling" is inaccurate. Cab39l, Calm1a, and Calm2a are not kinases.

Ln 538-540, please revise the sentence "CREBBPB among the known proteins potentially

involved in H3K27 acetylation in zebrafish was the only identified in our proteome or phospho-proteome at 4 hpf without significant changes.”

Ln639, “high translated” should be “highly translated”.

Ln642, revise the phrase “a dynamic of highly translation between 0-5 hpf”.

Ln643, “Among then” should be “Among them”.

G. References: appropriate credit to previous work?

References are appropriate.

H. Clarity and context: lucidity of abstract/summary, appropriateness of abstract, introduction and conclusions.

Appropriate except for a few inaccurate phrases listed in F.

We appreciate reviewer’s corrections that we have changed all these mistakes along the manuscript

Reviewer #3:

Remarks to the Author:

Hernandez-Huertas et al. present an enhanced version of their CRISPR-RfxCas13d system, designed to efficiently knock down gene expression in early embryos. Using this improved system, they conduct a proof-of-principle study to investigate the involvement of approximately 50 kinases and phosphatases in the early stages of embryonic development in zebrafish, specifically during the maternal-to-zygotic transition (MZT). The authors demonstrate efficient knockdown of nearly all targeted genes and identify a handful of genes that significantly impact the MZT. They focus their downstream analysis on the kinase Bckdk, performing a comprehensive set of experiments to elucidate its role in MZT. These experiments include gene expression measurements at various levels (open chromatin, RNA, newly synthesized RNA, protein, and phosphoproteomics), identification of a significantly altered histone modification (H3K27ac), and observation of partial impairment in miR-430 processing. The data also led them to investigate a chromatin remodeling factor, Phf10, whose knockdown affected MZT. Notably, overexpression of Phf10's constitutively active phosphorylated form rescued the effects of Bckdk knockdown. Furthermore, the authors demonstrate that the importance of bckdk in early embryonic development is conserved in medaka, highlighting the evolutionary significance of this kinase.

This work is exemplary in its comprehensive approach and depth of analysis. The authors have successfully established an improved CRISPR-RfxCas13d system for systematically screening genes involved in early vertebrate development, with a particular focus on MZT. What sets this study apart is not only the initial screening but also the thorough follow-up analysis of their top hit. Rather than providing a superficial overview, the authors conduct an in-depth characterization of the identified gene's involvement in MZT. As a result, this study presents a complete narrative and serves as an excellent blueprint for systematic studies of genes involved in MZT, including how to conduct comprehensive follow-up analyses. The quality and thoroughness of this work fully warrant its publication. I have only minor comments, which the authors should be able to address readily.

We sincerely appreciated these reviewer’s words on our work and thank their statements recommending the publication of our manuscript in "Journal X" after minor corrections

- Maybe mention earlier that you do this study in zebrafish, especially when you summarize what you did in the introduction.

We thank the reviewer for this suggestion, we have added it in lines 90-91.

- The selection criteria for the kinases – could be potentially a bit better described in the text.

We appreciate this suggestion and have now provided additional detail regarding the selection criteria for kinase and phosphatase candidates in lines 133–134.

- Did you actually knockdown positive controls that you know affect development? For example, the experiment in lines 169 to 173: it would have been great to have here a positive control to know what I can expect. If they simply were not included it is not a major thing, but actually would be interested to hear the authors thought on why they have excluded them.

We appreciate the reviewer's suggestion, which helps clarify what can be expected when targeting an essential factor for MZT and ZGA using CRISPR-RfxCas13d. In response, we have now included *nanog* mRNA knockdown as a control in our screen. Specifically, we added a bar graph showing the percentage of embryos with epiboly defects and a qPCR analysis confirming the efficiency of *nanog* mRNA knockdown (updated Fig. 1C and Extended Data Fig. 1A; lines 142–143 and 150-151).

This data also correlates what we had previously shown where CRISPR-RfxCas13d-mediated knockdown of *nanog* mRNA closely recapitulates both the developmental and molecular phenotypes observed with Nanog functional depletion using Morpholinos (Kushawah G. et al., 2020; Lee M. et al., 2013).

- Lines 183 to 186: "Indeed, analyzing downregulated genes from the 7 candidate KDs revealed that proteins related to calcium signaling and kinase activity shared the highest number of depleted genes among all possible comparisons, suggesting a potential common role for these proteins (Extended Data Fig. 1F)."

Extremely nitpicky - you mean genes - proteins here is a bit confusing as you talk about RNA-seq before. Could be misunderstood.

We apologize for the confusion caused by this sentence. What we intended to convey is that, based on RNA-seq data from knockdowns of mRNAs encoding proteins involved in calcium signalling and kinase activity, we observed that these proteins may share a common function. To clarify this point, we have now corrected the sentence and added more detail about the analysis in the revised manuscript (lines 209–213), figure legend and Methods (lines 913–915).

- Lines 394 to 397: "First, 3336 phosphorylated proteins were detected in both phospho-proteomes and we observed 19 phospho-peptides from 16 proteins to be differentially phosphorylated in the *bckdk* mRNA KD compared to the control embryos (Fig. 5A; Phosphoproteins in control and KD embryos)."

What about total protein levels for these phosphopeptides? Did they also change?

That is an excellent question, and we have indeed analyzed this. Unfortunately, for most of the proteins showing changes in our phosphoproteomic dataset, we did not detect them in the proteomic dataset—likely due to the differences in sensitivity between these two approaches. Among the few phospho-peptides that were detected in both datasets (only two), we did not observe significant differences in protein abundance between Bckdk KD and wild-type embryos. We have provided this data now in the manuscript (lines 475-477).

- Lines 406 to 409: “These results strengthened the hypothesis that the early developmental phenotype and MZT perturbation in the zebrafish embryos depleted of *bckdk* mRNA was due to the non-mitochondrial activity of this kinase (Extended Data Fig. 7A).”

This could also be secondary effects - maybe just slightly rephrase - does not need to be direct effect of the kinase what is a bit implied here, like you write just below for the upregulated phosphosites and actually thereby you enhance your implications that the effect is direct for the downregulated. However, you do not have definite evidence for that.

We apologize for the lack of clarity. What we intended to convey is that the maternal-to-zygotic transition (MZT) alterations observed in Bckdk KD embryos are unlikely to result from Bckdk’s mitochondrial activity. Another way to say this is that we have found no evidence supporting any effect of the knockdown on mitochondrial function. Therefore, our conclusion is based on two key observations: first, our phosphoproteomic analysis did not reveal phosphorylation changes in any known mitochondrial proteins, including Bckdha—a well-characterized Bckdk target; and second, BCAA levels were not significantly reduced in *bckdk*-depleted embryos. Moreover, in our proteomic dataset, Bckdk is still clearly detected under knockdown conditions, which likely reflects the persistence of the maternally provided mitochondrial fraction, consistent with the data mentioned above.

In addition, we now provide new evidence strongly suggesting that the kinase activity of Bckdk is directly involved in regulating Phf10 phosphorylation, which in turn controls early zebrafish development, ZGA and H3K27ac levels.

First, we have now demonstrated that Bckdk and Phf10 interact in zebrafish embryos, as shown by a pull-down assay (Fig. 5H and Extended Data 8L; lines 511–516). Since Phf10 is a nuclear and cytosolic protein (where it is translated), this suggests that it is the non-mitochondrial pool of Bckdk that likely phosphorylates Phf10. Whether this interaction is direct or not remains unknown. However, our phosphoproteomic analysis was conducted at 4 hpf, soon after fertilization and Bckdk depletion, suggesting that many of the peptides with reduced phosphorylation may represent direct Bckdk targets.

Second, we generated a kinase-dead version of Bckdk and showed that it is unable to rescue the developmental defects observed upon *bckdk* mRNA knockdown (Fig. 2C and Supplementary Fig. 5K-M; lines 264–277), highlighting the necessity of Bckdk’s kinase activity.

Third, we completed our rescue experiments using different *phf10* mRNA variants and confirmed that only the phosphomimetic version of Phf10 can nearly fully rescue both the developmental phenotype and the reduction in H3K27ac levels seen in Bckdk-depleted embryos (Fig. 5F and G and Extended Data Fig. 8H-K; lines 496–511).

Taken together, these results strongly support the conclusion that non-mitochondrial Bckdk kinase activity regulates Phf10 phosphorylation, thereby modulating H3K27ac levels and orchestrating proper MZT progression in zebrafish.

- Lines 412 to 414: “Furthermore, 11 proteins were differentially accumulated upon bckdk mRNA KD where 9 were down-regulated and 2 more abundant (2-fold change $p < 0.05$, Supp. Fig 7C, Supp. Table 2).”

Not 100% clear what is meant here. Do you mean 11 of the 16 that showed phospho changes? Based on the figure actually it seems that it is 11 in the total proteome. This is actually quite little especially if compared to the strong RNA changes. Could the author speculate why that is? I expect the main reason is the developmental time point of protein measurements. Do the authors expect this to be very different a few hours later in development.

Another thing – it would be great to provide the information how many proteins were identified in total added here (the information is available in Table S2, but would be nice to add). Actually, did the authors measure Bckdk – was it downregulated at the protein level? If I look in suppl table 2 I think that it seems that it is significantly downregulated but only slightly (\log_2 fold of -0.2)? It is ok that is slightly – this can also have an effect and the authors should potentially report that. However, as such it could be that 4 hpf was a little bit too early to do proteomics and phosphoproteomics – later time points might have shown stronger changes. However, that is a minor point and more of a personal opinion and a strong point can also be made to look as early as possible, but it is before any phenotype can actually be detected (based on Fig S4H). I might have missed it, but it would be interesting to hear the authors’ reasoning to look already that early?

We thank the reviewer for pointing this out, as it was not clearly explained in the manuscript. As mentioned above, the sensitivity of the phosphoproteome and the regular proteome analyses is different. Of the 11 proteins with significantly altered abundance, 3 were also found in the phospho-proteome but none of them appeared among the differentially phosphorylated proteins. Similarly, only 2 of the 16 differentially phosphorylated proteins were found in the proteome, and neither showed significant changes in protein levels. We have now further clarified this point in lines 471-477.

As the reviewer anticipated, we think these subtle differences are likely due to the early developmental time point at which the proteomic analyses were performed. Although our transcriptomic data show clear changes, most of the protein-level alterations at this stage are likely driven by post-translational mechanisms—a hypothesis we initially proposed and, to some extent, have now demonstrated. This fits with the biology of the maternal-to-zygotic transition (MZT), a process that begins in the absence of zygotic transcription and is initially regulated at the post-transcriptional and post-translational levels.

We also agree that more prominent proteomic and post-translational changes are likely to occur later in development upon Bckdk KD. However, we chose to analyze embryos at 4 hpf because this is when we first observe ZGA defects, and more importantly, prior to any visible developmental phenotypes. We reasoned that this was the most appropriate time point to uncover potential direct or early indirect Bckdk targets that may contribute to both the molecular defects seen at 4 hpf and the epiboly defects observed later at 6 hpf.

In addition, we agree with the reviewer that we should report the total number of proteins detected in our proteome analysis, as we did for the phospho-proteome. We have now added this information to the manuscript (lines 467 and 468).

Finally, the reviewer is correct in noting that we did not highlight the slight reduction in Bckdk protein levels. We now mention this explicitly, as it likely reflects depletion of the non-mitochondrial fraction of Bckdk, which we believe is primarily responsible for the

observed MZT effects. Because functional mitochondria and their protein content are maternally supplied, the remaining Bckdk protein upon *bckdk* mRNA KD at 4 hpf is likely of maternal origin and localized to mitochondria. We now discuss this interpretation in the revised manuscript (lines 600-603).

- Also in Suppl Table 2 please write add to the header of the column of “log2 fc” that this is KD/ctl – one can deduce that as the Cas13 is upregulated but would be nice to add it directly to the tables.

Thanks for the suggestion, we have done it.

- Lines 440 to 443: “Strikingly, the Bckdk depletion phenotype was rescued by the expression of a mutant of *phf10* mRNA containing an aspartic acid in position (T16D) mimicking the Phf10 phosphorylation by Bckdk in WT conditions (Fig. 5F).”

Did the author check here that both the knockdown and the overexpression worked successfully? This is not shown but assumed that the double (triple) perturbation worked. E.g. – it could be that expression of the D mutant suppresses somehow the downregulation of *bckdk* while the ectopic expression of the other variants does not.

We thank the reviewer for suggesting these important controls, which we have now performed. In all Phf10 rescue conditions, the level of knockdown was comparable (Extended Data Fig. 8H). In addition, we also assessed the expression of the different Phf10 variants by western blot using an anti-HA antibody in Bckdk depleted embryos (Extended Data Fig. 8I).

- Lines 445 and 446: “Interestingly, the overexpression of Ph10 WT version in the *bckdk* mRNA KD background exacerbated the epiboly defects (Fig. 5F).”

That is interesting as I would have expected the same result as the alanine mutant or maybe even a partial rescue as more substrate is present and a knockdown (such as of *bckdk*) is never complete. Could the authors maybe speculate?

We have now repeated and completed this experiment by measuring H3K27ac levels via western blot under different rescue conditions (Fig. 5G and Extended Data Fig. 8J and K). These results confirm that only the phosphomimetic version of Phf10 fully restores H3K27ac levels. In parallel, we also quantified developmental defects across these conditions. With the inclusion of additional replicates (we originally had only two), we found that the phenotype is not significantly exacerbated with the Phf10 WT version. We think that the previously observed effect was due to variability in a single noisy replicate. As a result, we have removed the corresponding sentence from the manuscript.

- Lines 562 to 573 in the discussion: “In addition to Smarca2 and Smarca4a, which are well-known to be phosphorylated during mitosis to remove them from chromatin we also detected phosphorylation of other SWI/SNF complex members including Arid1a, Arid1b, Arid2, Pbrm1, and Smarcb1, at 4 hpf (Supp. Table 2), suggesting protein phosphorylation could be an important post-translational regulator of the activity of this complex during MZT. Furthermore, our phospho-proteomic analysis detected phosphorylated proteins related to MZT such as Nanog, Brd3a, Brd3b, Brd4 and H1m. Together, these data suggest that, beyond the regulation of Phf10 mediated by Bckdk, the phosphorylation status of these chromatin and transcription factors may play a crucial role in mediating their function during the ZGA and MZT.”

What is not clear here is if you detected changes in phosphorylation of these proteins in the *bckdk* KD relative to control or you just detected those but unchanged. That would mean very different things. I think as written it implies that you measured changes relative to control.

We apologize for the confusion. We just detected those phosphopeptides but without changes in *Bckdk* KD vs control embryos. We have rephrased this section (lines 668-672) and clarified it.

- Figure 2B – I am missing at what time point hpf this experiment was done. I guess at 6hpf. Please indicate.

Yes, reviewer is right, we have added this information to the figure.

- Figure 2F: For medaka – I am missing the negative control experiment where a non-*bckdk* targeting control gRNA (for example against *gfp*) was used?

As in our previous zebrafish experiments, the control used here was Cas13d injected alone, which we established as a valid negative control in our earlier work (Kushawah G. et al., 2020). In that study, Cas13d alone produced no developmental or significant transcriptomic changes—similar to gRNA alone or gRNAs targeting a reporter gene. Nevertheless, in response to the reviewer's suggestion, we have now included a non-targeting gRNA as an additional negative control. As expected, this also resulted in no observable developmental defects, consistent with our previous findings for the Cas13d-alone condition. These new data have been incorporated into Figure 2G and are described in lines 294–295 of the revised manuscript and fit with other non-targeting controls added to our original screening (Fig. 1C and E, Extended Data Fig. 2 and 3)

- In 3C: I am missing probably something obvious, but why is the RNA seq for PZG defined by Baia-Amaral et al., not shown?

This is a great point. Our initial choice was based on the fact that different methods were used to define PZGs in RNA-seq (Lee et al., 2013) versus SLAM-seq (Baia-Amaral et al., 2024). However, we have performed the analysis using the PZG list defined by Baia-Amaral and found that these genes are also significantly downregulated upon *bckdk* mRNA knockdown—comparable to what we observed in the *phf10* mRNA knockdown condition (Fig. 5D and E).

We initially did not include this analysis in the screening because we felt it might not be a fair comparison, given that we were using RNA-seq data and preferred to rely on a PZG list derived using a similar approach. That said, our results consistently show a global reduction in the expression of PZGs upon *Bckdk* KD, regardless of the specific gene list used.

- In 4A: show maybe the p-value (KS-test) and D-values of the CDFs?

Thanks for the suggestion. We have added it to the figure since it's only mentioned in the text.

Reviewer #1 (Remarks to the Author):

The authors have compiled a major revision and included a considerable amount of new data. I initially had several concerns with this manuscript, including off-target effects of developmental delay, modest impacts on chromatin accessibility, a lack of evidence for impacts on H3K27ac, and an overall lack of negative controls in their study. The authors have now addressed all these concerns, and they have persuaded me that the Cas13 method has potential as a useful tool for zebrafish researchers interested in knocking down maternal mRNAs.

Reviewer 1 clearly acknowledges that all their concerns have been fully addressed but now they raised a new one which was not clearly pointed out in their first revision (see below). We respectfully think that this behavior suggests a lack of consistency that already seems to anticipate a potential bias opinion leading them to reject the manuscript rather than a fair analysis.

*I do have one remaining concern. Authors show in their revised manuscript through proteomics data that their Cas13-mediated knockdown of *bckdk* mRNA leads to a modest -0.2 Lg2FC reduction in overall *Bckdk* protein levels. The most rational explanation for this results is that their Cas13 method did not work as anticipated, and many of their outcomes are a results of non-specific impacts, which was my major initial concern. In this case, we havent learned anything new about ZGA or about *Bckdk* function during early zebrafish development. It would therefore seem that their evidence indicates the technique was not successful, leading me to question the basic premise of their study as well as nearly all of their conclusions.*

From their comment, we understand that Reviewer #1 appears to attribute all of our highly convergent findings from multiple and independent experiments to non-specific effects. However, we have a robust and completed set of data demonstrating that this new concern from the reviewer is inconsistent:

1.1) After selecting *Bckdk* for further analysis based on the developmental and initial molecular phenotype (RNA-seq, pure zygotic genes of ZGA were downregulated, **Fig. 1E**), **we used five independent gRNAs**, four designed in the coding sequence and one targeting the 3' UTR that efficiently decrease *bckdk* mRNA levels and recapitulated the early developmental phenotype previously observed (**Extended Data Table 1, Extended Data Fig. 5A-C & E**). If the effects were non-specific, five independent gRNAs would not be expected to consistently reproduce the same phenotype.

1.2) *bckdk* mRNA KD showed **no off-targets effects nor collateral activity** (**Extended Data Fig 2 and 4**) as the vast majority of KDs analyzed in this study with the exception of calmodulins depletion that presented some off-targets due to the sequence similarity between different calmodulins (something acknowledged and discussed in our manuscript)t. Beyond, we have demonstrated that collateral effects from CRISPR-RfxCas13d in zebrafish embryos are only observed when targeting extremely abundant RNAs that are ectopically injected such as reporter mRNAs (Moreno-Sanchez I. et al Nature Communications 2025).

1.3) We found a similar developmental and molecular phenotype when depleting *bckdk* mRNA in **Medaka (Fig. 2F-H and Extended Data Fig. 5M-O)**. It is hard to believe that a rational explanation for this is that all these non-controlled effects suggested by Reviewer #1 are going to be shared in two independent organisms where the knockdown of the same mRNA induces similar developmental a molecular phenotype.

1.4) We have **rescued the phenotype** caused by RfxCas13d with a gRNA targeting the 3' UTR of *bckdk* mRNA, by the co-injection of a mRNA encoding for Bckdk tagged with a HA containing a different, and gRNA resistant, 3' UTR (**Fig. 2A and B**). As a control, the phenotype was not rescued by an ectopic *gfp* mRNA (**Fig. 2A and B**). This experiment unequivocally demonstrated that the phenotype is dependent of *bckdk* mRNA depletion. Importantly, we went a step further in the revision and we demonstrated that a mRNA containing only **two mutations that generate a Bckdk protein lacking its kinase activity does not rescue the phenotype (Fig. 2C and Extended Data Fig. 5K-M)**. This result indicates that an active kinase encoded by the injected mRNA is responsible of the experimental rescue and it is not that just that *bckdk* mRNA might be able to rescue the phenotype. Altogether our data show that *bckdk* mRNA in zebrafish is encoding for an important and functional kinase to control early development. One more time, if the effect was non-specific, the rescue of the phenotype by only the wild type Bckdk (and not the kinase- dead Bckdk version) would not be expected.

Additionally, we have now proactively conducted an **additional experiment** directly addressing Reviewer #1's minor comment from the first revision that seems to be the trigger of this new point raised now, which we initially considered not strictly necessary for the acceptance of this work.

“Minor concern: The authors quantified mRNA levels after Cas13d knockdown; could they use an antibody for Bckdk or an affinity-tagged version?”

We performed a KD of HA-tagged *bckdk* mRNA by co-injecting Cas13d and gRNAs targeting *bckdk* mRNA. Protein levels were then assessed by western blot using an anti-HA antibody. The results unambiguously demonstrate that CRISPR-RfxCas13d-mediated KD of *bckdk*-HA mRNA leads to a strong and significant depletion of the corresponding protein at 4 hpf (p -value = 0.0007, unpaired t -test; see Reviewer Figure 1), as expected.

This suggest that the highly efficient mRNA depletion observed upon *bckdk* mRNA KD (91.08%) should be followed by a similarly robust reduction in the novo protein synthesis. The relatively modest but significant log2 fold change observed at the global proteome level (log2FC = -0.21, p -value = 0.03, empirical Bayes moderated t -test method and p -values adjusted with Benjamini-Hochberg method; **Extended Data Table 2**) reflects the high maternal contribution of Bckdk protein, particularly within the mitochondria—an aspect not affected by the CRISPR-RNA targeting strategy. While this may be seen as a limitation of the approach, in this specific context, it enables a more focused understanding of the cytosolic role of Bckdk during the maternal-to-zygotic transition (MZT) in teleosts. Importantly, the role of this post-translational regulator could not have been elucidated using traditional approaches such as MZ mutants, which would also disrupt the mitochondrial fraction. Our work highlights the utility of the CRISPR-RNA targeting strategy not only for dissecting maternal contributions during MZT but also for investigating specific cellular contexts.

We believe that this control—demonstrating that the degradation of *bckdk* mRNA is followed by a corresponding reduction in protein levels—is not essential to include in the main manuscript. However, if the editorial board or the reviewer considers it necessary for the acceptance of this work, we would be pleased to incorporate it.

Reviewer figure 1. Affinity tagged Bckdk version knockdown. **A)** 10 pg pf *bckdk-ha* mRNA was injected in zebrafish embryos together with 1 ng of RfxCas13d protein and 450 pg of gRNA 2 and 3 or RfxCas13d alone in one-cell stage zebrafish embryos. **B)** Barplots representing Bckdk-HA levels relative to total proteins as the averages \pm standard error of the mean from 3 independent biological replicates. Zebrafish embryos were collected at 4 hpf (n = 15-20 embryos/replicate, ***p<0.001 unpaired *t*-test). **C)** Western blot (up) of the experiments quantified in panel B and stain-free signal of the gel as loading control (below) are shown

*In an attempt to explain this results, authors speculate that Bckdk protein levels are high and stable in mitochondria, and their approach is unable to influence these levels. This logically implies that there are two separate pools for Bckdk, one which is not in the mitochondria, is more rapidly turned-over, and is critically important for ZGA (the pool which they can effectively target), and another stable pool in the mitochondria which has no role in ZGA. While this explanation seems possible, the authors provide no evidence to support this speculative view. Thus, the more simple explanation for their results seems more likely, that the Cas13 method did not work as intended, they were unable to knockdown Bckdk effectively, and many outcomes described in their study are indirect or non-specific outcomes. Since they are not claiming that *bckdk* mRNA has function, their entire study rests on their ability to deplete Bckdk protein, and so this issue remains a major sticking point for me.*

Reviewer #1 again seems to ignore part of our data and scientific literature that strongly suggest that Bckdk role during MZT is through its non-mitochondrial fraction.

2.1) The main function known for Bckdk in the mitochondria is to phosphorylate Bckdha to maintain BCAA levels and this has been extensively mentioned in the manuscript. Accordingly, we found that BCAA levels are similar (**Fig. 2 D and E**) and correspondingly Bckdha phosphorylation in a motive described as substrate of Bckdk (tyrighhS*tsddssa, Hornbeck PV et al., 2015 NAR) did not significantly change in Bckdk KD vs control embryos (**Fig. 5A**).

2.2) Moreover, our quantitative phospho-proteomic experiment showed that the 9 proteins

significantly less phosphorylated upon Bckdk KD are not predicted to be localized in the mitochondria (**Fig. 5A**) in contrast to a phospho-proteome upon Bckdk inhibition in rat liver where more than 55% of the identified targets localized in the mitochondria (White et al., 2018 Cell Metabolism). This result suggests the *bckdk* mRNA KD triggers a lack of function of this protein and that this occurs in the non-mitochondrial fraction.

2.3) Bckdk knockdown developmental and molecular phenotypes were rescued only with the mRNA encoding to the phospho-mimicking version of Phf10 and not by the wild-type nor the non-phosphorylatable mutant version (**Fig. 5 F and G**). If the effect was non-specific, the rescue of the Bckdk phenotype would not be expected by only the phospho-mimicking version of Phf10. But beyond, we have demonstrated that Bckdk and Phf10 interact during early development in zebrafish embryos indicating a clear connection between Bckdk and Phf10 (**Extended Data Figure 8L**), ruling out these indirect effects argue by Reviewer #1. Further, this interaction is unlikely to happen in the mitochondria, since Phf10 is obviously translated in the cytosol and then transported the nuclei where plays its role (We have mention this point in the discussion lines 668-669). To observe the interaction between these two proteins Bckdk should be localized in the cytosol and the lack of this protein likely causes the decreased of Phf10 phosphorylation with the resulting MZT alteration.

2.4) The reviewer argued that log₂ -0.2 depletion of Bckdk protein is a slight reduction, but they did not even look at the statistical analysis since **this reduction is significant** (*p-value*: 0.03, empirical Bayes moderated t-test method and *p-values* adjusted with Benjamini-Hochberg method; **Extended Data Table 2**). In our proteome only 81 out of the 4996 detected proteins (1.6%) showed a significant different level (*p-value*< 0.05), we have updated now this information to the manuscript in lines 601-610. Importantly, as reviewer #3 acknowledges in the revision, this significant reduction can generate an important alteration during MZT specially in a context where Bckdk (as protein) is highly maternally provided and abundant (see below). Indeed, according to the scientific literature (reviewed in our manuscript) and to the data we have generated and previously highlighted (points 2.1 and 2.2) most of the Bckdk protein appears to be mitochondrially located and its function unaltered. Indeed, the average of number of mitochondria is approximately 2×10^7 in one-cell stage zebrafish embryos that creates a particular context where the cytoplasm is full of mitochondria (Artuso L et al., 2012 BBA-Bioenergetics, Otten ABC et al., 2016 Cell Reports, Alfonso O et al., 2025 Nature Cell Biology). This high number of mitochondria, compared to mammalian oocytes, is likely attributable to the differing sizes and metabolic requirements of externally developing embryos and more importantly. Notably, most of the maternal mitochondria are mature and active during maternal stages in zebrafish embryos to promote the needed oxidative phosphorylation during early development. Then, it is reasonable to think that most of Bckdk protein that is maternally provided localizes in these functional mitochondria which correlates with the high maternal protein contribution of Bckdk. Notably, several reports based on proteomic data have shown that Bckdk is a highly abundant kinase during early developmental stages, including 4-cell and 32-cell stages, at 1 and 1.75 hpf (Shen W et al., 2022, Cell; Yan J et al., 2023, Journal of Proteome Research). For example, at 4-cell stage Bckdk is among the top 5% most detected protein and the top 15% within mitochondrial proteins (Yan J et al., 2023). This early stage mainly contains Bckdk maternal protein and not the product of maternal mRNA translation that will need more time to produce *de novo* Bckdk protein. In this scenario, the novo translated *bckdk* mRNA will provide new protein that, most likely, it will not be incorporated to the maternal mitochondria since, as earlier mentioned, they are fully mature and functional and includes active Bckdk protein as we showed in **Fig 2D and E and 5A**.

Notably, there is no generation of mitochondria until much later in zebrafish development once MZT is completed (Otten ABC et al., 2016 Cell Reports) where Bckdk from mRNA translation could be again incorporated to new produced mitochondria. Therefore, the simpler explanation is that Bckdk from the translation of its maternal RNA localizes in the non-mitochondrial fraction rather than in the mature and maternal mitochondria during zebrafish MZT that is already functional and that it is not affected when *bckdk* mRNA is depleted (**Fig. 2 D and E** and **Fig. 5A**). Consequently, it is reasonable to think that maternal *bckdk* mRNA translation in 4 h (when we measured our mRNA KD and protein levels through proteomics) will generate a smaller fraction of the total amount of Bckdk at this stage and that is exactly what we detected through quantitative proteomics: a small but significant reduction (\log_2 -0.2 depletion, *p-value*: 0.03) of Bckdk when efficiently eliminate maternal *bckdk* mRNA. Notably, the increased of Bckdk as detected protein from 4-cell stage to 1K-cell stage (3 hpf) is approximately 9% (Yan J et al., 2023, *Journal of Proteome Research*) that could be driven through the translation of *bckdk* mRNA and that correlates with the percentage of significant depletion that we observed in our proteome data upon *bckdk* mRNA KD (approximately 13%).

As we acknowledged in the response to the reviewers, we have tried two approaches to differentiate mitochondrial and non-mitochondrial Bckdk at 4 hpf but unfortunately and due to the extremely high number of mitochondria per embryo and cell (Otten ABC et al., 2016 Cell Reports, Alfonso O et al., 2025 Nature Cell Biology) we could not unequivocally and properly distinguish between these two fractions neither by sub-fractionation and western blot nor by immunofluorescence analysis. Nevertheless, this non-mitochondrial role of Bckdk is well supported by all the data we showed, and it is in agreement with the recently known activity of cytosolic Bckdk in biological contexts related to cell proliferation that are also associated to stem cell and pluripotency-like states as we discussed in our manuscript.

In summary, it is concerning that Reviewer #1 argues that the observed depletion and effects are likely non-specific simply because they think are “modest” **and ignoring that this depletion is statistically significant**. This is especially relevant in a context as zebrafish embryos that contains a high amount of maternally provided Bckdk and it is expected that the vast majority of this protein localizes in the functional maternal mitochondria where its role and level are unaltered in *bckdk* mRNA depleted condition. In such a scenario with high amount of maternal Bckdk, obtaining a significant reduction from this small and non-mitochondrial fraction from the translation of *bckdk* maternal mRNA is clearly meaningful.

Further, we have completely addressed all the points from reviewer #1 as they acknowledged. This included new experimental data even though some of them were fully demonstrated in our previous work (Kushawah G. et al., 2020 Dev Cell, Hernandez-Huertas L. et al., 2022 Star Protocols) and the clarification of certain points that they raised due to a lack of understanding and misinterpretation of the results as we think it is the case again. Indeed, their new issue comes from one of the original minor points where they suggested to test the level of Bckdk protein reduction using a tagged version of Bckdk injected as mRNA in zebrafish embryos that we have now done to further clarify their question (see above). Nevertheless, in our response of the first revision, we referred to all our data that they did previously ignore, bringing now this new issue in this second revision in an inconsistent manner. Indeed, it seems that this reviewer holds a persistent concern regarding the reliability of CRISPR-RfxCas13d technology and now without any solid argument to follow up our first response they have tried to argue again with something that is not actually supported by data and rather than a matter

of personal feeling or opinion.

In conclusion, how many coincidences need to occur to explain all our consistent results just by chance and as a product of unspecific consequences upon *bckdk* mRNA depletion? In other words, what are the chances that everything showed in our manuscript was off-targets and non-controlled effects or the product of a notable number of unrelated events that make all our data correlate and point to a clear and unique direction? The most reasonable answer with all data sets we have provided in the original and revised manuscript together with data from scientific literature is to think that the conclusions are robust since they have been demonstrated from independent and complementary experiments and approaches. We are convinced that the other two reviewers and independent experts in the field will see our arguments as reasonable. Therefore, and despite we appreciate the Editor work on contacting with other very prestigious journals, we hope that after our careful response the Editorial Team could also agree with our argument and have our manuscript evaluated by other independent reviewer and finally accepted for publication in Nature Structural and Molecular Biology.

Reviewer #2 (Remarks to the Author):

The authors have address all concerns satisfactorily.

Reviewer #3 (Remarks to the Author):

This is a great resubmission, which fully addresses my all my previous concerns (which were already mostly minor). I was already enthusiastic about the original submission and think that the authors resubmitted a substantially improved manuscript, which just raises my enthusiasm. I fully support the publication of the manuscript in "Journal X".

Dear Dr. Moreno-Mateos,

Thank you for submitting your manuscript which was previously reviewed at a different venue to The EMBO Journal. Your study has now been seen by an arbitrating referee, who finds that the remaining concerns by the original referees have been addressed and recommends publication of the manuscript. There remain only a few mainly editorial points that have to be addressed before I can extend formal acceptance of the manuscript:

- Please upload the manuscript as .docx file and remove the figures and "track changes"
- Please make sure to add all relevant funding information in the manuscript is also entered into our submission system.
- Please reduce the number of keywords on the abstract page to five (ideally choosing broad general terms).
- Please adjust the format of the reference list and of the in-text citations according to EMBO Journal format (alphabetical order, author name et al + year.../up to 10 author names in the reference list before et al / please refer to our Guide to Authors for additional information on EMBO J reference format).
- Please rename the Conflict of Interest section into "Disclosure and Competing Interests Statement", in accordance with our updated Guide to Authors (<https://www.embopress.org/competing-interests>)
- As we are switching from a free-text author contribution statement towards a more formal statement based on Contributor Role Taxonomy (CRediT) terms, please remove the present Author Contribution section and instead specify each author's contribution(s) directly in the Author Information page of our submission system during upload of the final manuscript. See <https://casrai.org/credit/> for more information.
- There is a reference to "data not shown" on page 50, "Tandem Mass Tag (TMT) Labeling" section. According to our policy, which does not permit references to "data not shown", please include this information in the Appendix. Please see also <https://www.embopress.org/page/journal/14602075/authorguide#unpublisheddata>.
- Please adjust the in-text callouts for individual figures and figure panels: e.g. missing callout for Fig. 2H
- Please provide either a "Yes" or a "Not Applicable" answer to each one of the questions in your Author Checklist (<https://www.embopress.org/pb-assets/embo-site/EMBO%20Press%20Author%20Checklist-1642513524327.xlsx>). In the last column of this checklist, only the sections of the manuscript where the relevant information can be found should be listed (the information per se should be included in the main manuscript file).
- Please upload all figures as individual, high-resolution Figure files; figure legends should remain in ms file below the References; nomenclature should be Figure EV1-EV9 instead of Extended Data Figure 1-8, and EV figures should also be upladd as individual, high-resolution Figure files with legends below the main figure legends in ms
- Please update all source file names, titles, legends and manuscript callouts to Dataset EV1-EV# instead of Extended Data Table 1-2, legends should be removed from ms and uploaded as a separate tab/sheet in each Excel file
- Please provide the Reagent and Tools Table. For more information, please check <https://www.embopress.org/page/journal/14602075/authorguide#structuredmethods> and download the template for Reagent Table
- Please provide suggestions for a short 'blurb' text prefacing and summing up the conceptual aspect of the study in two sentences (max. 250 characters), followed by 3-5 one-sentence 'bullet points' with brief factual statements of key results of the paper; they will form the basis of an editor-written 'Synopsis' accompanying the online version of the article. Please also provide an altered synopsis image, making sure that the aspect ratio conforms to our website's format - it should be exactly 550 pixels wide and between 300-600 pixels high.
- Please provide the specific URL for GSE268294 dataset is not provided in the data availability statement.
- Figure Legends (main + EV):
 1. Please note that the exact p values are not provided in the legends of figures 4C, 5F, G; EDF 1F, EDF 5A, E; EDF K, L; EDF 8H
 2. Please indicate the statistical test used for data analysis in the legends of figures 1E, F; 2F, 3F, EDF 2, EDF 3, EDF 5 O, EDF 6B, C, D, H

3. Please note that the error bars are not defined in the legend of figure 2C

- Please rename the "SUMMARY" to "Abstract"

- Sections need to be named and the order should be corrected: Title page - Abstract - Keywords - Introduction - Results - Discussion - Methods - Data Availability - Acknowledgements - Disclosure and Competing Interests Statement - References - Figure Legends - Table(s) - Expanded View Figure Legends.

With best regards,

Cornelius Schneider

Cornelius Schneider, PhD
Editor | The EMBO Journal
c.schneider@embojournal.org

Use the link below to submit your revision:

Referee #1:

Hernandez-Huertas et al. present an enhanced version of their CRISPR-RfxCas13d system, designed to efficiently knock down gene expression in early zebrafish embryos. Using this improved system, they conduct a proof-of-principle study to investigate ~50 kinases and phosphatases in the early stages of embryonic development in zebrafish. Downstream analysis focuses on the kinase Bckdk, which they show plays important roles in regulating MZT. The paper is significant not just for providing enhanced, validated tools for CRISPR-RfxCas13d screening in zebrafish, but because it provides extensive follow up of their top hit, revealing new insights into roles for Bckdk in early vertebrate development. Although CRISPR-RfxCas13d system is a relatively new method, it has been generally well accepted and successfully employed by others. As with any knockdown approach, Cas13 knockdown approaches have the potential to yield artifacts due to off target or nonspecific impacts. However, the authors do a commendable job of including extensive controls that demonstrate the effectiveness of these tools in their screen (including mock, RfxCas13d only, gfp, and rfp negative controls and a nanog positive control). Careful and appropriate stage match controls are used to demonstrate Bckdk impacts MZT, and is not just reflective of developmental delay. The fact that the Bckdk knockdown phenotype is rescued by gRNA resistant bckdk mRNA, but not a kinase dead version, and that it is rescued by a phosphor mimic Phf10 mRNA but not wildtype or non phosphorylatable versions lends strong support to mechanistic hypothesis that Bckdk acts in part through phosphorylation of phf10. Commendably the authors go on to demonstrate a conserved role for Bckdk in medaka, showing expanded applicability of their tools and providing further evidence for the new biological functions of Bckdk they have uncovered. Experimentally substantiated roles for Bckdk in two different organisms also provided further support for their screening approach.

Dear Cornelius,

Thanks for inviting us to resubmit our manuscript and allowing us to address some editorial requirements. Please see below our point-by-point response.

- Please upload the manuscript as .docx file and remove the figures and "track changes"

We have submitted the revised manuscript as .docx file without figures or track changes.

- Please make sure to add all relevant funding information in the manuscript is also entered into our submission system.

All relevant funding information has been included in the main manuscript (Acknowledgements section) and entered in the submission system.

- Please reduce the number of keywords on the abstract page to five (ideally choosing broad general terms).

We have reduced the number of keywords on the abstract page to five.

- Please adjust the format of the reference list and of the in-text citations according to EMBO Journal format (alphabetical order, author name et al + year.../up to 10 author names in the reference list before et al / please refer to our Guide to Authors for additional information on EMBO J reference format).

We have reformatted the reference list and all in-text citations according to EMBO Journal guidelines.

- Please rename the Conflict of Interest section into "Disclosure and Competing Interests Statement", in accordance with our updated Guide to Authors (<https://www.embopress.org/competing-interests>)

We have renamed the section accordingly.

- As we are switching from a free-text author contribution statement towards a more formal statement based on Contributor Role Taxonomy (CRediT) terms, please remove the present Author Contribution section and instead specify each author's contribution(s) directly in the Author Information page of our submission system during upload of the final manuscript. See <https://casrai.org/credit/> for more information.

We have removed the previous Author Contribution section and specified each author's contributions in the Author Information page of the submission system.

- There is a reference to "data not shown" on page 50, "Tandem Mass Tag (TMT) Labeling" section. According to our policy, which does not permit references to "data not shown", please include this information in the Appendix. Please see

also [https://www.embopress.org/page/journal/14602075/authorguide#unpublished data](https://www.embopress.org/page/journal/14602075/authorguide#unpublished_data).

In this sentence data not shown was a mistake since we only wanted to mention that the quality of proteomics was very high with TMT labelling levels were >99% for all detected peptides. Therefore, we have removed "data not shown".

- Please adjust the in-text callouts for individual figures and figure panels: e.g. missing callout for Fig. 2H

We apologize for this. All in-text callouts have been revised, and Fig. 2H is now correctly cited in line 320 of the manuscript.

- Please provide either a "Yes" or a "Not Applicable" answer to each one of the questions in your Author Checklist (<https://www.embopress.org/pb-assets/embosite/EMBO%20Press%20Author%20Checklist-1642513524327.xlsx>). In the last column of this checklist, only the sections of the manuscript where the relevant information can be found should be listed (the information per se should be included in the main manuscript file).

We have uploaded the complete form of the Author Checklist in the submission system.

- Please upload all figures as individual, high-resolution Figure files; figure legends should remain in ms file below the References; nomenclature should be Figure EV1-EV9 instead of Extended Data Figure 1-8, and EV figures should also be uploaded as individual, high-resolution Figure files with legends below the main figure legends in ms

We have uploaded all the figures and EV figures as individual high-resolution files (.ai). Figure and EV figure legends have now been included in the main manuscript below the References section. All nomenclature has been updated to EMBO Journal format.

- Please update all source file names, titles, legends and manuscript callouts to Dataset EV1-EV# instead of Extended Data Table 1-2, legends should be removed from ms and uploaded as a separate tab/sheet in each Excel file

We have updated all callouts to Dataset EV1-EV2.

- Please provide the Reagent and Tools Table. For more information, please check [https://www.embopress.org/page/journal/14602075/authorguide#structured methods](https://www.embopress.org/page/journal/14602075/authorguide#structured_methods) and download the template for Reagent Table

We have uploaded the Reagent and Tools Table to the submission system.

- Please provide suggestions for a short 'blurb' text prefacing and summing up the conceptual aspect of the study in two sentences (max. 250 characters), followed by 3-5 one-sentence 'bullet points' with brief factual statements of key results of the

paper; they will form the basis of an editor-written 'Synopsis' accompanying the online version of the article. Please also provide an altered synopsis image, making sure that the aspect ratio conforms to our website's format - it should be exactly 550 pixels wide and between 300-600 pixels high.

This is our proposed blurb text and bullet points.

Post-translational regulation during the maternal-to-zygotic transition (MZT) remains largely unexplored. Using an optimized CRISPR-RfxCas13d screening strategy, this study identifies *Bckdk* as a previously unrecognized kinase that governs the MZT in teleosts.

- CRISPR-RfxCas13d RNA-targeting screens uncover previously unrecognized kinases and phosphatases required for early zebrafish development.
- Depletion of *bckdk* mRNA causes widespread ZGA deregulation, inefficient clearance of maternal RNAs by miR-430, and a significant depletion of H3K27ac levels.
- *Bckdk* role during ZGA is conserved in medaka
- *Bckdk* phosphorylates Phf10/Baf45a, a chromatin remodelling factor, whose downregulation also alters ZGA.
- Restoring Phf10 phosphorylation levels rescues both developmental defects and H3K27ac depletion observed upon *bckdk* mRNA knockdown.

Additionally, we have updated a synopsis image to the submission system.

- Please provide the specific URL for GSE268294 dataset is not provided in the data availability statement.

We apologize for the omission. We have added the specific URL for GSE268294 in the Data Availability statement, and the dataset is publicly accessible. In addition, the mass spectrometry datasets are publicly available, and the corresponding links have been updated.

- Figure Legends (main + EV):

1. Please note that the exact p values are not provided in the legends of figures 4C, 5F, G; EDF 1F, EDF 5A, E; EDF K, L; EDF 8H

We included exact *p*-values in all figures and mentioned it accordingly in figure legends.

2. Please indicate the statistical test used for data analysis in the legends of figures 1E, F; 2F, 3F, EDF 2, EDF 3, EDF 5 O, EDF 6B, C, D, H.

We apologize for this. We have indicated the statistical tests used in the specified figure legends.

3. Please note that the error bars are not defined in the legend of figure 2C

We have defined the error bars in the legend of figure 2C

- Please rename the "SUMMARY" to "Abstract"

We have renamed the SUMMARY section to Abstract.

- Sections need to be named and the order should be corrected: Title page - Abstract - Keywords - Introduction - Results - Discussion - Methods - Data Availability - Acknowledgements - Disclosure and Competing Interests Statement - References - Figure Legends - Table(s) - Expanded View Figure Legends.

All sections have been named and reordered accordingly to The EMBO Journal requirements.

We hope that with these minor editorial requirements addressed you can find now our manuscript suitable for publication in The EMBO Journal.

Thank you very much for your cooperation and support.

Sincerely yours,

Miguel A. Moreno-Mateos PhD
Associate Professor and Group Leader
Andalusian Center of Developmental Biology
CSIC-UPO-JA, Seville, Spain

Ariel A. Bazzini
Assistant Investigator
Stowers Institute for Medical Research
Assistant Professor.
Dep. of Molecular & Integrative Physiology
KU Medical Center

Dear Dr. Moreno-Mateos,

I am pleased to inform you that your manuscript has been accepted for publication in the EMBO Journal.

Yours sincerely,

Cornelius Schneider, PhD
Editor
The EMBO Journal
c.schneider@embojournal.org
